# Detecting generalization deficits in large language and reasoning models by using natural variations in simple problems

**Marianna Nezhurina**[1,2,4*] **Lucia Cipolina-Kun**[1,2,3] **Mehdi Cherti**[1,2,4] **Jenia Jitsev**[1,2,4*]
[1] *LAION*    [2] *Juelich Supercomputing Center (JSC), Research Center Juelich (FZJ)*
[3] *School of Electrical and Electronic Engineering, University of Bristol*    [4] *Open-Ψ (Open-Sci) Collective*
[*] *Corresponding authors: {m.nezhurina,j.jitsev}@fz-juelich.de, contact@laion.ai*

**Reviewed on OpenReview:** *https://openreview.net/forum?id=frA7uYn2um*

## Abstract

Large language and reasoning models (LLMs, LRMs) are instances of foundation models exhibiting scaling laws that predict generalization improvement when increasing the pre-training scale. As such, they are supposed to possess strong generalization and therefore transfer robustly across various tasks and conditions in few-show or zero-shot manner. Such claims rely on various standardized benchmarks that should measure core functions like generalization and reasoning, where state-of-the-art (SOTA) models score high. We demonstrate remarkable zero-shot generalization deficit in most SOTA models which claim strong function, including reasoning models like DeepSeek R1 or o1-mini, trained at the largest scales, using a simple, short common sense math problem formulated in concise natural language, easily solvable by humans, which we term Alice in Wonderland, AIW, problem. The deficit manifests in strong performance fluctuations on natural variations in the simple problem template that do not change either problem structure or its difficulty at all. By testing models on further control problems with similar form, we rule out that deficit might be rooted in minor low-level issues like natural language or numbers parsing. In conventional LLMs, we observe strong overconfidence in the wrong solutions, expressed in form of plausible sounding explanation-like confabulations. Many models showing the deficit also collapse close to 0 accuracy on AIW problems, while still exhibiting high scores on various standardized benchmarks. We show how this illusion of strong function might be caused by leakage of test sets into training. For reasoning models, while observing clearly improved performance compared to LLMs, we still see strong fluctuations on problem variations that keep structure and difficulty unchanged. Our observations suggest that current LLMs and LRMs possess generalization deficits that can be detected by controlled structure and difficulty preserving variations in simple problems, in contrast to standardized benchmarks which contain problems of higher difficulty but fail to detect such clear failures[1].
.

## 1 Introduction

State-of-the-art (SOTA) large language and reasoning models (LLMs, LRMs) are important instances of foundation models. A central property attributed to foundation models is strong, robust zero-shot generalization, enabling them to solve problems not present during the training. Evidence of strong generalization rely mainly on high scores obtained in various standardized benchmarks. Such benchmarks contain sets of problems that reach up to advanced high school (MMLU Hendrycks et al. (2020), GSM8k Cobbe et al. (2021), MATH and MATH-500 Hendrycks et al. (2021)), olympiad (AIME24/25 AIME (2024)) or graduate (GPQA-Diamond Rein et al. (2024)) difficulty levels.

---

[1]Code for reproducing experiments in the paper and raw experiments data can be found at AIW repo

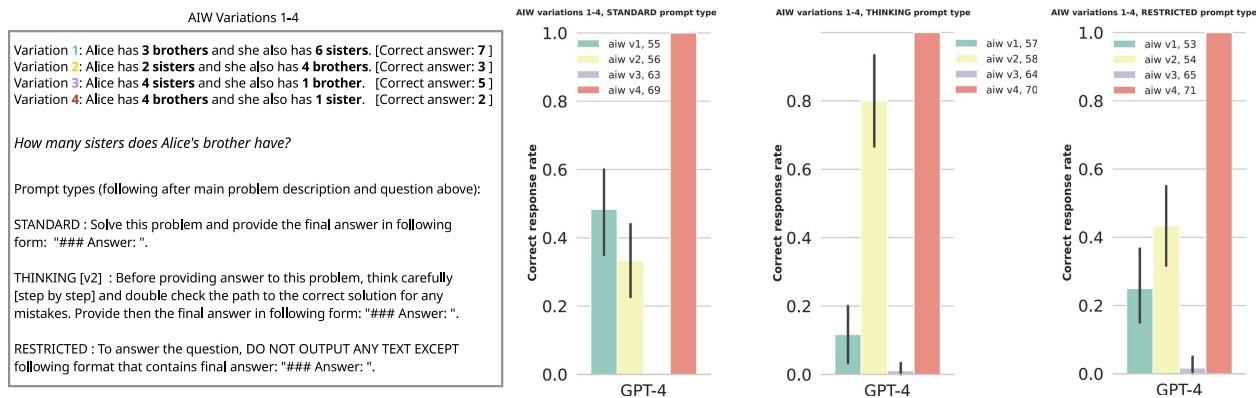

Figure 1: AIW problem is simple common sense math problem with short, concise formulation, serving as a minimalist setting to detect deficits in 0-shot generalization. Problem template is used to create AIW variations 1-4 (**left**) by instantiating different numbers $N, M$ of brothers and sisters, keeping problem structure and difficulty unchanged. We measure sensitivity of models to those variations, observing strong performance fluctuations. Here on example of GPT-4 (gpt-4-0613) tested with various prompt types STANDARD, THINKING, RESTRICTED, a color per each variation 1-4, executing 60 trials per each variation. Correct response rate varies strongly depending on the variation. E.g., it drops close to 0 for variation 3, while going up to 1 for variation 4. This observation is consistent for different prompt types - STANDARD, THINKING and RESTRICTED, showing that observed fluctuations are not due to prompt variation. Strong performance fluctuations on natural, structure preserving variations of such a simple problem points to severe lack of robustness and generalization deficits. Numbers in the legend are prompt IDs (Suppl. Tab. 3)

Recent works however provided evidence that high scores obtained on standardized benchmarks drop significantly if better controlling for contamination of training data by test sets Golchin & Surdeanu (2023); Li & Flanigan (2024). Further works questioned whether standardized benchmarks properly reflect model function and generalization by pointing out various function failures that were seemingly incompatible with claimed strong capabilities Mitchell (2023); Wu et al. (2023); Frieder et al. (2024). It thus still remains unclear whether scores on standardized benchmarks can be used as generalization error estimates. Instead of signaling strong generalization, high scores on benchmarks might simply reflect the degree of test set leakage into training. Should high scores on standardized benchmarks mislead about actual model generalization capability, it becomes further unclear how to reliably detect generalization deficits and debunk strong function claims in cases where those are based on memorization of test samples instead of proper generalization.

To shed light on this situation, we propose a distinct approach to detecting generalization deficits. We start from definition of a simple problem template that supports natural variations which keep problem structure and difficulty unchanged. We generate then a set of problem instances originating from same problem template using such variations. As each problem instance has same structure and difficulty, models that generalize well should exhibit none or only slight performance differences for each instance. In contrast, performance fluctuations on instances of same problem should reveal models with generalization deficits.

To demonstrate the approach, we create a short, common sense natural language math problem template. The problem (in following Alice in Wonderland, AIW problem) template has following form: ***"Alice has $N$ brothers and she also has $M$ sisters. How many sisters does Alice's brother have?"***. Instantiating natural numbers $N, M \leq 7$ allows natural variations that do not change problem structure and difficulty and thus should not affect ability to solve resulting problem instances. (Fig. 1).

When tested on instances of simple AIW problem, we observe SOTA LLMs exhibiting strong performance fluctuations, showing accuracies close to 100% on some, while dropping close to 0 on others, despite all instances being structure and difficulty preserving variations of the same simple problem (Sec. 3.1). The strong fluctuations remain across various prompt types, including chain of thoughts. To rule out that observed sensitivity to variations is rooted in inability to execute robustly various basic operations necessary to handle the problem, we test models on control problems containing same operations as required to solve the original

(Sec. 3.1.1). We observe models are able to solve all variations of control problems without performance fluctuations. This provides evidence that models can handle all the basic operations well, and observed sensitivity is due to inability to correctly infer same problem structure across instances, which is hallmark of failure to generalize.

Confronted with evidence of zero shot generalization breakdown all tested SOTA LLMs exhibit on such a simple problem, we investigate further AIW problem versions to see whether the same phenomenon of model sensitivity to problem structure preserving variations appears consistently. We use same technique of varying numbers in problem templates and observe same strong fluctuations of model performance also on other problem versions, obtaining further evidence that the generalization deficit is generic and not unique to original AIW formulation. We also observe SOTA LRMs like o3-mini OpenAI (2025a), o1-mini OpenAI (2024e), DeepSeek-R1 Guo et al. (2025) or various distilled reasoning models exhibiting strong fluctuations on AIW problem versions (Sec. 3.2). Already in models that use simple supervised finetuning (SFT) on small amounts of reasoning traces data unrelated to AIW (e.g, S1.1 32B Muennighoff et al. (2025)), we observe elevated average correct response rates and thus clearly improved performance compared to conventional LLMs. However, persisting strong fluctuations on problem structure and difficulty preserving variations reveal again lack of robustness and generalization deficit also for this model class.

We observe that many models score high on standardized benchmarks containing problems of higher difficulty, while exhibiting strong performance fluctuations on variations of simple AIW problems that are far below such levels. We hypothesize that this illusion of strong function conveyed by high benchmark scores is partly rooted in test set leakage and poisoning of training data, which distorts generalization measurement. We provide evidence for potential test set leakage by contrasting performance of models like Claude 3.5 Sonnet on older (before its appearance) and more recent (after its appearance) AIW problem versions. We accompany the evidence with experiment showing how Llama 3.1 8B goes from zero to strong performance across AIW variations if using AIW problem data for fine-tuning (Sec. 3.1.2).

In summary, our study makes following contributions: (i) presenting a novel approach to detect generalization deficits by measuring performance fluctuations across natural, difficulty and structure preserving variations of same simple problems (ii) showing SOTA LLMs and LRMs exhibit strong fluctuations on simple AIW problems (iii) designing control problems to ensure the observed issues are due to generalization deficit (iv) showing how discrepancy between high scores on standardized benchmarks and observed generalization deficits on AIW problems can be explained by test set leakage, warning against the illusion of strong function mediated by the benchmark scores.

Unlike evaluations that vary problem difficulty and execution length (see eg Shojaee et al. (2025) for Tower-of-Hanoi problem based setup), introducing additional confounds, our approach preserves problem difficulty, allowing to attribute failures on problem instances clear to generalization deficits. We also study far simpler problems - most AIW versions are elementary-school level, so failures are especially diagnostic.

Our study highlights that novel evaluation procedures and benchmarks are required to properly reflect models' generalization. Strong performance fluctuations exhibited by LLMs and LRMs on variations of simple AIW problems reveal generalization deficits that are incompatible with claims of robust problem solving (ranging up to olympiad or graduate difficulty level) as conveyed by high scores on standardized benchmarks. Using structure and difficulty preserving variations of simple problem templates offers thus an effective way to test for model sensitivity and generalization failures that are not detected by standardized benchmarks.

## 2 Methods & Experiment Setup

### 2.1 Simple common sense reasoning problems with difficulty & structure preserving variations

**AIW Problem.** To measure models' sensitivity to problem irrelevant variations and thus probe the zero-shot generalization, we use following problem template: ***"Alice has $N$ brothers and she also has $M$ sisters. How many sisters does Alice's brother have?"***. The problem has a simple common sense solution which assumes all sisters and brothers in the problem setting share the same parents. The correct response $C$ - number of sisters - is easily obtained by calculating $C = M + 1$ (Alice and her sisters), which gives the number

of sisters Alice's brother has. To create problem variations, we choose to vary natural numbers $N, M \leq 7$, restricting number variations to a simple common sense setting. This way we obtain AIW variations 1-4 (see Suppl. Tab. 3) that all pose same problem using variations irrelevant for problem solving, leaving its difficulty and structure unchanged. See Suppl. Sec. B for the resulting variations. We use 3 prompt types, RESTRICTED, STANDARD and THINKING, to ensure we measure models across various prompt formulations and check the observations hold independent of employed prompt type (see also Sec. 2.2 and Suppl. Sec. B).

### 2.1.1 Control AIW Light problems

To control for models failures not coming from inability to execute basic problem specific operations, we make various versions of AIW problem - AIW Light Family, AIW Light Arithmetic Siblings and AIW Light Arithmetic Total Girls - that are designed to test various operations. Staying similar to AIW original, control problems test basic family relations structure handling, identifying relevant entities, executing arithmetic operations and in general, natural language and numbers parsing required for handling the original AIW problem.

The AIW Light problems keep problem template close to the original, changing only the final question part such that the posed modified question tests particular operations. The shared common template part is taken from the original AIW problem: ***"Alice has $N$ brothers and she also has $M$ sisters.*** The variations 1-4 are again created in the same way like in AIW original by varying natural numbers of brothers and sisters, while ensuring that the natural numbers for final correct answers in AIW original and AIW Light are matched across variations 1-4 (see also Suppl. Sec. B.1).

**AIW Light Family**. AIW Light Family continues common template with following question: ***"How many brothers does Alice's sister have?"***. To solve the problem, reporting already given number of brothers is sufficient - the correct answer is $C = N$. This requires only basic grasping of relational family structure (understanding entity "Alice's sister", binding female attribute to Alice and realizing Alice and her sisters share same brothers). It does not require execution of any arithmetic or set operations, in contrast to AIW original. Should the issues with solving AIW original be rooted in handling basic family structure, we should see models also failing here. For variations 1-4, see Suppl. Tab. 5.

**AIW Light Arithmetic Siblings**. AIW Light Arithmetic Siblings continues common template with following question: ***"How many siblings does Alice have?"***. To solve the problem, summing up already given numbers of brothers and sisters is sufficient - the correct answer is $C = N + M$. This requires basic grasping of relational family structure (realizing Alice's siblings are her sisters and brothers) and selection and execution of elementary arithmetic sum operation. In contrast to AIW original, it does not require execution of set operations nor binding sex attribute to Alice to properly assign her to correct sets. Should the issues with solving AIW original be rooted in selection and execution of elementary arithmetic operations in family frame, we should see models also failing here. For variations 1-4, see Suppl. Tab. 4

**AIW Light Arithmetic Total Girls**. AIW Light Arithmetic Total Girls continues common template with following question: ***"How many girls are there in total?"***. To solve the problem, it is necessary to bind female attribute to Alice via the pronoun "she", to assign correct female attributes to the sisters and to execute the correct arithmetic sum operation adding all the obtained girls - the correct answer is $C = M + 1$ (also coinciding with correct answer for AIW problem). This requires basic grasping of family structure (realizing who are the girls in the family) and selection and execution of elementary arithmetic sum operation. In contrast to AIW original, it does not require execution of set operations to properly assign Alice to sisters set. Should the issues with solving AIW original be rooted in binding correct sex attributes or counting total members of particular sex in family frame given its structure, we should see models also failing here. For variations 1-4, see Suppl. Tab. 6.

### 2.1.2 AIW problem versions with similar structure

To see whether observations are consistent for other problems of related kind, we construct further two problem versions, AIW Extended (AIW Ext) and AIW Friends that have similar problem structure and difficulty as AIW original. For both problems, the correct answer still has the form $C = M + 1$, where $M$

depends on natural number variation within the problem template. We employ both problem versions as further test for generalization, as it allows us to have both variations within a problem version (natural numbers variations) and also check model sensitivity to changes across problem templates, while still keeping highly similar problem structure and similar difficulty across problems (in contrast to AIW Light control problems which are designed to test low level operations involved, such that their templates pose problems of lower difficulty than AIW original).

**AIW extended (AIW Ext.)** In this problem version, two entities, Alice and Bob are employed as sister and brother, otherwise keeping the problem template and structure close to the AIW original. See Suppl. Sec. B.2 for the full problem formulation with variations 1-6.

**AIW Friends** In this problem version, the family frame setting is abandoned. Instead, we use male and female friends in problem formulation. This way, while we vary the problem formulation, the problem structure and difficulty are still highly similar to AIW original (brothers and sisters are male and female siblings). We use an additional statement to ensure there is no common sense ambiguity in this problem version. See Suppl. Sec. B.2 for the full problem formulation with variations 1-6.

### 2.1.3 Harder AIW problem versions

To test what happens if the simple AIW problems are further extended towards a substantially harder difficulty level, providing further challenge for the tested models, we construct problem versions of higher difficulty increasing number of relations and entities, while still keeping same rather simple relational logic structure. We formulate the problems such that despite their increased difficulty, the correct answer still has the form $C = M + 1$, with $M$ depending on given natural number instantiation within the problem template. While the solution to AIW+ and AIW Colleague Circles problem is harder to obtain than the solution to AIW original with its much simpler structure, the problem solving capabilities required to cope with these problems are far below from olympiad or graduate difficulty level, on which many recent LLMs and LRMs are claimed to perform successfully.

**AIW+ problem**. We constructed an AIW+ problem that features additional hierarchy and distractors when describing family structure that includes various entities like brothers, sisters, cousins, uncles and aunts. It has also a substantially longer template than AIW original. See Suppl. Sec B.2.1 for full formulation and its variations.

**AIW Colleague Circles**. This problem deals with work colleagues setting, and the difficulty is increased by introducing circles of male and female colleagues with various relations that has to be handled to properly answer the question. The template is also longer than AIW original. See Suppl. Sec B.2.1 for full formulation and its variations.

## 2.2 Prompt types and response parsing

It is well known that so-called prompt engineering can heavily influence the model behavior and model response quality Arora et al. (2022); Wei et al. (2022); White et al. (2023). To check that our observations reflect model sensitivity to controlled problem structure preserving variations in same manner independent of particular prompt type, we employed 3 various prompt types to provide model's input: STANDARD (prompt with instruction to format final answer output as a natural number), THINKING (prompt that in addition encourages thinking in spirit of CoT) and RESTRICTED (prompt with instruction to output nothing else but final answer as a natural number). THINKING v2 prompt type is a minor variation of THINKING type that just adds "step by step" after already existing "think carefully" phrasing (control experiments show that THINKING and THINKING v2 are equivalent in terms of observed performance, so we use both interchangeably). STANDARD, THINKING and THINKING v2 prompt types allow models to generate any output length and thus not restricting compute before delivering the final answer. RESTRICTED is used as control with restricted output to measure baseline of model performance when compute for producing final answer is limited (Suppl. Tab. 3)

**Parsing model responses.** To perform evaluations of model performance, it is necessary to parse and extract the final answer from the responses provided by the models. Each input to the model is combination

of a AIW problem variation, followed by one of prompt types as described before. To keep the parsing procedure simple, we add to each problem prompt following output format instruction: *"provide the final answer in following form: "### Answer: ""*. We observed that all models we have chosen to test were able to follow such an instruction, providing a response that could be easily parsed. We also ran control experiments without such formatting instruction in the problem formulation, ensuring that behavior does not depend on it.

## 2.3 Selecting models for evaluation and conducting experiments

We are interested in testing current state-of-the-art models that claim strong function, especially in generalization and reasoning, backed up by high scores shown on standardized benchmarks that are assumed to measure generalization and reasoning capabilities to solve problems. We therefore select models widely known and used in the ML community that also appear in the top rankings of the popular LLM leaderboards, like openLLM leaderboard by HuggingFace or ELO leaderboard by LMsys. We provide the overview of the selected models in Suppl. Tab. 2 and list in Suppl. Tab. 13 the corresponding standardized benchmarks where they obtain strong scores.

We expose selected SOTA LLMs and LRMs, including advanced models at largest scales (see Suppl. Tab. 2) to AIW problem variations 1-4 (Suppl. Tab. 3), AIW Light control problems (Suppl. Tab. 4, 5, 6) and further AIW versions (Suppl. Sec. B.2), using different prompt types as described above. For each combination of model, AIW problem variation and prompt type, at least 30 trials are collected to compute correct response rates, Suppl. Fig. 27. For details on correct response rates estimation procedure, see Suppl. Sec. A.2

We use hosting platforms that offer API access or local deployment via vLLM Kwon et al. (2024) for testing the models, and automatize the procedure by scripting the routines necessary to prompt models with our prompts set. The routines are simple and can be used by anybody with access to the APIs (we used liteLLM and TogetherAI for our experiments) or to locally hosted models to reproduce and verify our results. We protocol all the data from interactions with the models to enable community checking. We release all the collected raw response data, correct response rates estimates and routines used to conduct experiments as open-source for reproducibility and further usage.

## 3 Results

### 3.1 Breakdown of SOTA LLMs on the simple AIW problem

**AIW reveals severe generalization and reasoning deficits in SOTA LLMs**. Following the procedures described in Sec 2, we expose the selected models that claim strong function and reasoning capabilities (Suppl. Tab. 2) and measure their correct response rate performance across and for each AIW variations 1-4 using various prompt types, executing $> 30$ trials for each combination (see also Suppl. Tab. 3 and Suppl. Fig. 27). The results suggest that confronted with the AIW problem, models suffer a severe function breakdown. This breakdown has two main manifestations:

**1. Low correct response rates.** Despite evident problem's simplicity, the majority of models stay well below correct response rate of $p = 0.4$. We summarize the main results in the Fig. 2. Many models are not able to deliver a single correct response (Suppl. Tab. 13, Suppl. Fig. 6). The only major exceptions from the observation of very low correct response rates are the largest scale closed models GPT-4o/4 and Claude 3 Opus. These models obtain correct response rates well above $p = 0.4$, leaving the remaining large and smaller scales open-weights (e.g., Mistral/Mixtral, Llama 2/3, Qwen 2.5, Command R+, and Dbrx Instruct) and closed-weights models (e.g., Gemini Pro, Mistral Large) far behind. Many of the models that claim high scores on standardized benchmarks, show very low correct response rates close to 0, eg. Llama-3-8B, Mixtral-8x22B, DBRX instruct, or exhibit even complete breakdown on AIW with correct response rate of zero across all variations, eg Command R+ or Qwen1.5-72B (Suppl. Tab. 13)

The results presented in the Fig. 2 show estimates for correct response rates averaged across STANDARD, THINKING and THINKING v2 prompt types (Suppl. Tab. 3, prompt IDs provided for reproducibility; Suppl. Fig. 6 with more models scoring 0). RESTRICTED prompt type was used as further control, providing

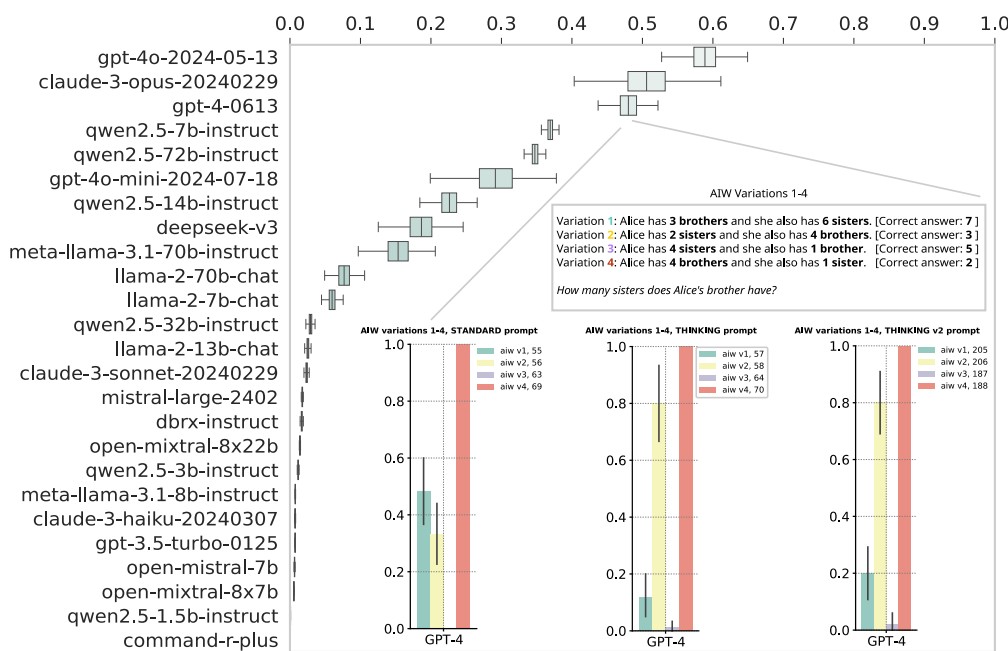

Figure 2: Severe deficits of conventional LLMs on simple AIW problem. (**main**) Models correct response rate, average over STANDARD, THINKING, THINKING v2 prompt types and AIW variations 1-4. (**inlay**) Strong fluctuations on AIW variations 1-4, despite problem structure and difficulty remaining entirely unchanged across variations. Overall correct response rate averaged across variations does not reveal these fluctuations (shown on example of GPT-4). Numbers in the legend are prompt IDs (Suppl. Tab. 3)

baseline for performance with restricted compute by forcing models into short outputs (see Suppl. Sec. C and Suppl. Fig. 7). The 3 models crossing $p = 0.4$ as a better performing group are the GPT-4o ($p = 0.589$), Claude 3 Opus ($p = 0.507$) and GPT-4 ($p = 0.4803$). Other open-weights and closed models are lagging far behind. For better performers, when inspecting the responses with correct final answers, we see also correct reasoning backing up the final answers. In the poor performers, among the few responses with a correct final answer we encounter responses where final answer turns out to be an accident of executing entirely wrong reasoning. In such cases, various mistakes lead coincidentally to the final output number corresponding to the right answer. Such responses are encountered in models with low correct performance rates ($p < 0.4$) (see Suppl. Sec. D for response examples, and for raw data), and we correct via manual inspection the status of correct response for such cases.

**2. Strong performance fluctuations across irrelevant AIW problem variations.** Importantly, we observe strong fluctuation of correct response rates across AIW variations 1-4 as introduced in Sec. 2. Such fluctuations strongly affect better performers with higher average correct response rates like GPT-4/4o and Claude 3 Opus (as poor performers have often correct response rates across all variations close to 0, so that no room for fluctuations exist on that very low performance level). As shown in the Fig. 2 (inlay) for AIW original formulation and Fig. 3 for various AIW versions (**A**) original, (**B**) AIW Ext and (**C**) AIW Friends, the correct response rates can fluctuate between being close to 1 to being close to 0, depending on AIW variation. Remarkable is that such fluctuations appear despite AIW variations being all instances of the very same simple problem, as changes in numbers used across AIW variations do not change either the problem structure or its difficulty at all. This lack of robustness in simple problem setting hints on severe deficits in generalization. The strong fluctuations across variations appear independent of employed prompt types (Fig. 2 (inlay), Fig. 1), while correct response rate averaged across all variations also varies across prompt types, showing in addition expected prompt type dependency (Suppl. Fig. 7, 8)

### 3.1.1 Control experiments using AIW Light problems

As outlined in Sec. 2.1, we execute control experiments using AIW Light problems (Fig. 3 **E** - **F**) to rule out that observed deficits are caused by low level issues like natural language/numbers parsing related to

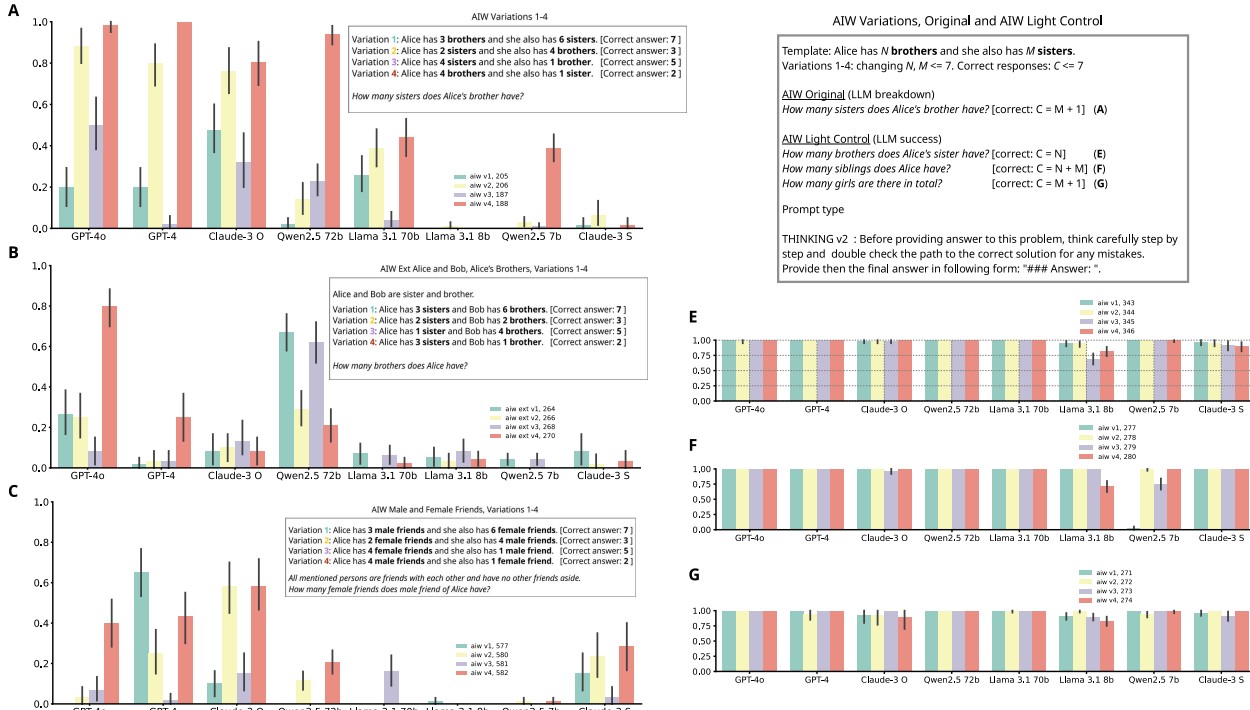

Figure 3: **Strong fluctuations** on problem variations (a color per each variation 1-4) observed on AIW original (**A**) and two further versions, AIW Extended (AIW Ext (**B**) and AIW Friends (**C**), using THINKING v2 prompt type. On all AIW versions, better performers (GPT-4/4o, Claude 3 Opus) show strong fluctuations, getting higher correct response rates ($p > 0.3$) on some variations, while dropping strongly close to zero on others, despite variations 1-4 leaving problem structure and difficulty unchanged, only instantiating different numbers $N, M$ into problem templates. AIW Ext (**B**) and AIW Friends (**C**) show overall altered correct response rates (eg, for GPT-4/4o, Claude 3 Opus) compared to AIW original (**A**)). Strong fluctuations are common phenomenon across the problem versions, hinting on generic generalization deficits that are independent of specific problem formulation. Differences in performance between the problem versions provide additional evidence for lack of model robustness, as problem structure and difficulty is highly similar between the versions. **Control experiments**. We rule out low-level issues (tokenization/language parsing) and lack of specific knowledge by AIW Light control experiments (**E**) Family (**F**) Total Siblings and (**G**) Total Girls. Across all AIW Light control problems, strong fluctuations disappear. Models collapsing on AIW variations close to 0 obtain high correct response rates on control. This proves that handling family relations, binding sex attributes, parsing numbers, handling elementary arithmetics, etc, are all intact and not the cause for breakdown on AIW variations.

tokenization or by inability to deal with specific problem setting, eg handling family relations. In all following experiments, for each AIW Light variation, 60 trials were executed to estimate correct response rate and its variance. We observe that models that undergo collapse close to 0 on AIW original variations are able to perform well on variations of all AIW Light problems, such that strong fluctuations observed on AIW original mostly disappear. Strong performance on control problems show that models can handle family structure and bind attributes in AIW Light Family (Fig. 3 **E**), and select and execute necessary arithmetic operation to compute quantities following correct inference of family structure in AIW Total Siblings (Fig. 3 **F**) and AIW Total Girls (Fig. 3 **G**). This demonstrates that handling various basic operations necessary to solve AIW original problem is not the issue.

### 3.1.2 Debunking strong function illusion

After public release of AIW original problem (May 2024), number of models claiming strong function were released, including Claude 3.5 Sonnet (June 2024), also confirmed by standardized benchmarks. We show

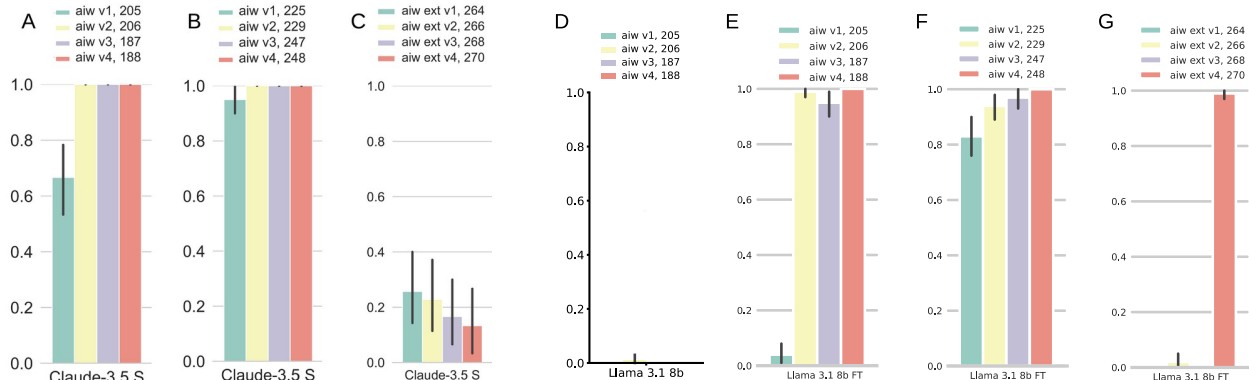

Figure 4: Debunking strong function illusion. Claude 3.5 Sonnet Rise **(A)**, **(B)** and Fall **(C)**, and its possible explanation via test set contamination using Llama 3.1 8B fine tuning experiment. While correct response rates go up close to 1 on variations 1-4 for **(A)** AIW original and also for **(B)** AIW Bob version, strong breakdown is observed on AIW Ext **(C)**. Strong fluctuation occurs despite AIW Ext being highly similar to AIW original, revealing severe generalization deficit. Possible explanation lies in data contamination that can create illusion of strong function, as shown by the experiment with fine-tuning Llama 3.1 8B. In its original state, the performance on AIW original variations 1-4 is close to 0 **(D)**. After fine-tuning on generated samples containing AIW original variations 1-4, mixed with Alpaca instruction data, correct response rates shoot high up for AIW original **(E)** and AIW Bob version **(F)**. On AIW Ext **(G)** however performance collapses, again revealing generalization deficit and resembling observations for Claude 3.5 Sonnet.

here how such claims can be misleading by testing the model on AIW original and slightly modified AIW versions (Sec. 2.1.2). In Fig. 4 we see that while the model handles well both AIW original (Fig. 4 **A**) and a version where Alice is replaced with Bob (Fig. 4 **B**), the performance drops strongly on AIW Ext featuring both Alice and Bob (Fig. 4 **C**). AIW Ext and original AIW have highly similar problem structure, however AIW Ext was not revealed before Claude 3.5 Sonnet release. We hypothesize thus that strikingly different behavior on AIW Ext roots in test set leakage of AIW original problems into pre- or posttraining data.

We provide supporting evidence that test set leakage might indeed cause observed illusion of strong function by fine-tuning open weights Llama 3.1 8B on the data containing various problem instances generated from AIW original problem template. As evident in Fig. 4 **E** - **F** , similar picture emerges - we observe strong performance improvement of the fine-tuned Llama 3.1 8B on AIW original. It scores close to 1 on most of AIW variations, while still poorly performing on AIW Ext (Fig. 4 **G**). Models like Claude 3.5 Sonnet report high scores on standardized benchmarks, however strong function claims cannot be derived from those, as we see from strong collapse and performance fluctuations on variations of rather simple problems as AIW Ext.

### 3.1.3 Further relevant observations.

1. *Standardized benchmarks failure.* We observe failure of various standardized benchmarks (GSM8k, MATH, MMLU, ARC, etc) to properly reflect generalization and basic reasoning skills of SOTA LLMs by noting significant disparity between the low models' performance on the AIW problem and the high benchmark scores (see Suppl. Sec. C.5) *2. Dominance of wrong responses* We measure distribution of natural numbers responses on output, showing that for AIW variations with low correct response rate, peaks are on wrong answers, excluding majority voting methods as a fix (Suppl. Sec. C.4). *3. Confabulations and overconfident tone* We observe that wrong responses are often accompanied by persuasive explanation-like confabulations and overconfident tone about correctness of the wrong solutions provided by the models, which can further mislead model users (Suppl. Sec. E).

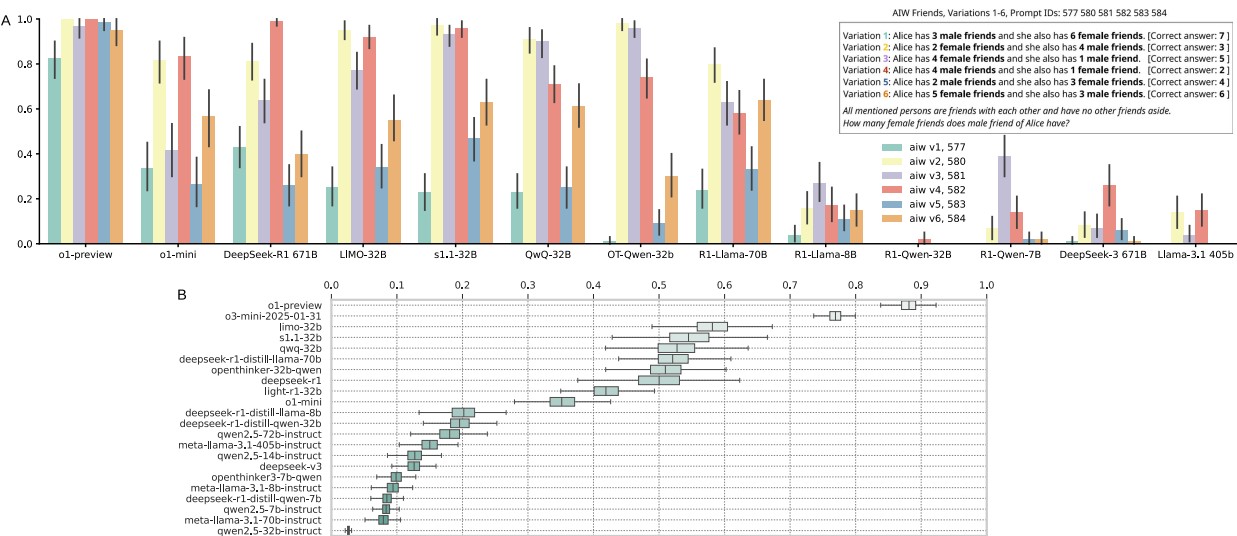

Figure 5: Reasoning models show severe deficits in generalization, while outperforming conventional LLMs. **(A)** Strong fluctuations of correct response rates on variations of AIW Friends problem (a color per each variation 1-6) exhibited by reasoning models. The fluctuations affect to same extent SFT only (eg S1.1 32B, LIMO-32B, OpenThinker-32B) and SFT+RL (eg QwQ 32B, DeepSeek-R1 671B) reasoning models. Conventional LLMs (DeepSeek-v3 671B, Llama 3.1 405B) are strongly outperformed by reasoning models. For reference, o1-preview is shown, which has much smaller fluctuations and higher overall correct response rates, where data contamination cannot be excluded as cause for the observed performance. Numbers in the legend are prompt IDs, see AIW repo **(B)** Albeit suffering from strong fluctuations, larger-scale reasoning models set themselves apart from the conventional language models. Shown are correct response rates averaged across all variations 1-6 of AIW Friends, AIW Plus, and AIW Circles Colleagues problems. Larger reasoning models on a 32B scale populate the mid level correct response rate range 0.3 - 0.6 (the only exception being R1-Qwen-32B), while conventional LLMs at largest scale stay well below 0.2. As a reference, we show closed reasoning models o3-mini and o1-preview that show only weak fluctuations and settle in the upper performance region above 0.7. Test data contamination cannot be excluded for models with closed data.

## 3.2 Severe generalization deficit in advanced reasoning models revealed by AIW variants

Recent reasoning models have shown strong increase compared to conventional LLMs in scores on standardized benchmarks related to problem solving at olympiad or graduate difficulty levels (AIME24 AIME (2024), MATH500 Hendrycks et al. (2021), GPQA Diamond Rein et al. (2024)). We take various types of reasoning models created from a strong base using either real or synthetic reasoning traces data: those using both conventional supervised fine-tuning (SFT) and reinforcement learning (RL) (DeepSeek R1 Guo et al. (2025) created from DeepSeek v3; QwQ 32B and Light-R1-32B created from Qwen 2.5 32B Instruct) and those employing SFT only, distilled using reasoning traces, either synthetically generated from DeepSeek R1 (S1.1 32B Muennighoff et al. (2025), OpenThinker 32B Team (2025a;b), based on Qwen 2.5 32B Instruct), from DeepSeek R1 Zero (R1-Distilled-Qwen-32b and R1-Distilled-Llama-70b Guo et al. (2025)) or curated from mix of real and synthetic reasoning data (LIMO-32B Ye et al. (2025)).

We measure the performance of these models using AIW problems, to test the strong claims behind the high scores achieved on reasoning benchmarks (see also Sec. 3.1.2) and also check how these models compare to conventional LLMs. To reduce confounds of test data leakage as described in Sec. 3.1.2, Fig. 4 and Suppl. Sec. C.2, Suppl. Fig. 14, we exclude AIW original and AIW Ext from the experiments, as reasoning models released very recently might have been exposed to data from those or very similar problems due to their public availability. We take thus AIW Friends, AIW Plus and AIW Circles Colleagues as problem test set (see Suppl. Sec. B.2 for AIW variants overview)

As evident from Fig. 5, reasoning models also exhibit strong fluctuations across AIW variations (shown for AIW Friends variations 1-6, THINKING v2 prompt, Fig. 5 **(A)**, o1-preview being a frontier closed model exception where data contamination can't be entirely ruled out). While average correct response rates for reasoning models increase across variations with larger model scale, strong fluctuations clearly persist. Despite reasoning models clearly outperforming conventional LLMs in average correct response rates (Fig. 5 **(B)**), we thus still observe severe zero-shot generalization deficits on variations of simple AIW problems, also for larger scale reasoning models (see also Suppl. Sec. C.1). Standardized reasoning benchmarks cannot detect the observed generalization deficit: models showing strong performance fluctuations show high benchmark scores. Models scoring high on MATH-500, AIME24 and GPQA-D can also have very low AIW Friends scores, Tab. 1. For similar observations on further AIW versions, AIW Plus and AIW Circles Colleagues, see Suppl. Sec. C.1, Suppl. Fig. 12, 13.

| Model | AIME24 | MATH-500 | GPQA-D | AIW Friends |
|---|---|---|---|---|
| s1.1-32B | 0.647 | 0.890 | 0.601 | 0.698 |
| DeepSeek-R1 | 0.798 | 0.973 | 0.715 | 0.588 |
| o1-mini | 0.636 | 0.900 | 0.600 | 0.529 |
| DS-R1-Distilled-Llama-70B | 0.700 | 0.945 | 0.652 | 0.537 |
| DS-R1-Distilled-Llama-8B | 0.417 | 0.891 | 0.490 | 0.150 |
| DS-R1-Distilled-Qwen-7B | 0.555 | 0.928 | 0.491 | 0.107 |
| OpenThoughts3-7B | 0.690 | 0.900 | 0.537 | 0.070 |
| DS-R1-Distilled-Qwen-32B | 0.726 | 0.943 | 0.621 | 0.003 |

Table 1: High scores on reasoning benchmarks containing advanced high school, olympiad or graduate level problems (MATH-500, AIME24, GPQA-D) do not guarantee high scores on much simpler AIW Friends problem. Lack of model robustness to AIW Friends variations are also not reflected in benchmark scores. Standardised reasoning benchmarks are thus not suitable for detecting problem solving and generalization deficits.

## 4 Related work & limitations

**Measuring LLMs' capabilities.** Since the seminal breakthroughs in language modelling Devlin et al. (2018); Raffel et al. (2020); Brown et al. (2020), measuring LLM capabilities became indispensable for evaluations and model comparison. To measure how well a language model performs on reasoning, there exists a plethora of different standardized reasoning benchmarks such as ARC Clark et al. (2018), PIQA Bisk et al. (2020), GSM8K Cobbe et al. (2021), HellaSwag Zellers et al. (2019), MMLU Hendrycks et al. (2020), WinoGrande Sakaguchi et al. (2019) and more recent (AIME24 AIME (2024) and GPQA Diamond Rein et al. (2024)). Multiple works aim on improving reasoning performance of LLMs as measured by those standardized benchmarks in various ways Wei et al. (2022); Yao et al. (2024); Zhou et al. (2022); Wang et al. (2022); Pfau et al. (2024); Guo et al. (2025).

**Stress-testing LLMs' weaknesses.** Paralleling impressive progress shown by LLM research, cautious voices have been raising concern about discrepancy between claimed capabilities as measured by standardized benchmarks and true LLM reasoning skills by presenting carefully selected evidence for model failures Mitchell (2023). In response, the research community has been undertaking attempts to create more challenging benchmarks like HELM Liang et al. (2023) or BIG-bench Srivastava et al. (2023). These benchmarks also aimed at properly testing generalization capabilities beyond memorization, in line with recent works that pointed out high test dataset contamination due to large-scale pre-training on web-scale data Golchin & Surdeanu (2023); Li & Flanigan (2024).

Multiple studies Wu et al. (2023); Dziri et al. (2024); Lewis & Mitchell (2024); Berglund et al. (2023); Moskvichev et al. (2023); Huang et al. (2023) have shown breakdowns of language models reasoning capabilities in different scenarios and also lack of robustness to variation of problem formulation Zong et al. (2024); Zheng et al. (2024). Other works were looking into particular reasoning failures like deficits in causality inference

Jin et al. (2023b;a). These works operate often with formalized, rather complex problems that does not have simple common sense character expressible in natural language. Similar in spirit to our work, simple math word problems were used in Patel et al. (2021) to show model breakdowns, but models on current frontiers that claim strong generalization and advanced capabilities were not tested. Recently, Mirzadeh et al. (2024) made use of similar approach to create variations from templates of GSM8K problems. This work however does not provide for strong models any conclusive evidence of function breakdown on simple problems, in fact showing evidence that models like GPT-4o or Llama 3 8B can handle those well (see Suppl. Sec. C.6, Suppl. Fig. 29). In contrast, we are able to measure lack of model robustness and strong fluctuations also for frontiers LLMs and reasoning models pretrained at largest scales. We control for problem difficulty to keep it unchanged, also ensuring problem execution length not becoming a confound for failures, so that we can clearly attribute those to generalization deficits, in contrast to Shojaee et al. (2025). A key limitation of our current approach is the lack of sufficient diversity in AIW problem variations. This can be addressed in future work by systematic procedural instance generation for broader evaluation across more diverse problem types.

## 5    Discussion & Conclusion

In our work, using a simple AIW problem (Sec. 2) that can be easily solved by adults and arguably even children, we observe a striking breakdown of SOTA LLMs performance when confronted with variations of the AIW problem template that do not change problem structure or its difficulty (Suppl. Tab. 3). The breakdown is manifested in (i) low average correct response rates (Fig. 2) and (ii) strong performance fluctuation across structure and difficulty preserving natural variations of the same problem, which reveals severe zero-shot generalization deficit of the models in this case (Fig. 3). While clearly outperforming conventional LLMs, recent reasoning models like DeepSeek-R1 also show strong fluctuations when facing variations of different AIW versions (Fig. 5). By executing control experiments, we show that models do not lack capability to perform basic operations necessary to solve AIW problem (Sec. 3.1.1, Fig. 3 **E-F**). This provides further support for the hypothesis that observed performance fluctuations origin from failure to properly infer common structure from instances of the same problem and thus signal generalization deficit.

Our study also points to failure of standardized benchmarks to properly measure model generalization and detect generalization deficits as revealed by testing on AIW problem variations (Tab. 1, Suppl. Sec. C.5, Fig. 22, Tab. 13). Standardized benchmarks assigning high scores to SOTA LLMs and LRMs fail to signal clear generalization deficits made evident by breakdown on natural variations of simple AIW problem. We note that despite observed breakdown with low average correct response rates and strong fluctuations, the reasoning can be functional and also conventional LLMs, especially at larger scale (eg. GPT-4, Claude 3 Opus), show instances of fully correct reasoning (see Suppl. Fig. 32, 33). As our results show, this reasoning capability is however fragile and cannot be accessed robustly, as reflected by strong performance fluctuations on natural variations of problems as simple as AIW.

Detecting generalization deficits via AIW problem testing helps to avoid illusion of robust strong function. It reveals discrepancy between high scores on problems of graduate or olympiad difficulty contained in standardized benchmarks and inability to handle robustly simple AIW problems. As a hint on origin of this apparent contradiction we show that for highly similar, but different AIW versions, models can exhibit strongly different performance (Fig. 4 **(A-C)**). Fine-tuning experiment reveals that exposing model to AIW problem examples during training can strongly push performance and abolish fluctuations across variations of the given problem. However, strong fluctuations on a different, similar AIW problem version that is not shown during training remain (Fig. 4 **(D-G)**). This again provides evidence for generalization deficits as root cause and test set leakage being one possible explanation for the strong function illusion.

The evidence from our study serves thus as a warning against attributing to LLMs and LRMs strong generalization and robust reasoning on problems of higher difficulty by relying solely on high scores the models achieve on standardized problem solving benchmarks. While we see in the recent reasoning models a promising direction to enable stronger model function than conventional LLMs offer, clear generalization deficits still persist in LRMs also on problems as simple as AIW Friends, as reflected in strong performance fluctuations (Fig. 5). While already simple SFT on reasoning traces data can thus strongly boost LLMs, the generalization deficit however remains, also when using SFT+RL. To properly measure generalization

and guide further progress, more sensitive benchmarks are required. AIW problem and its variations offer a measurement technique that can reveal lack of robustness and generalization deficits that current benchmarks do not detect. Variations built into problem templates can serve as method for creating new benchmarks that are, in contrast to current common benchmarks, no longer static and can probe robustness to variations and also better counter test set leakage (Fig. 4), providing more reliable measurement and testing of model generalization and reasoning. New benchmarks should follow Karl Popper's principle of falsifiability Popper (1934), aiming to falsify strong function hypothesis by spotting deficits, and thus showing possible directions for model improvement, while also offering protection from overblown claims about model's capabilities.

## Acknowledgments

We would like to express gratitude to all the people who are working on making code, models and data publicly available, advancing community based research and making research more reproducible. Specifically, we would like to thank all the members of the LAION Discord server[2] community and Open-$\Psi$ (Open-Sci) Collective[3] for providing fruitful ground for scientific exchange and open-source development.

LCK acknowledges the Helmholtz Information & Data Science Academy (HIDA) for providing financial support enabling a short-term research stay at Juelich Supercomputing Center (JSC), Research Center Juelich (FZJ) to conduct research on foundation models and their evaluation.

MN, MC and JJ acknowledge co-funding by EU from Digital Europe Programme under grant no. 101195233 (openEuroLLM), co-funding from EuroHPC Joint Undertaking programme under grant no. 101182737 (MINERVA), funding by EU Horizon under grant no. 101214398 (ELLIOT), as well as funding by the Federal Ministry of Research, Technology and Space of Germany (BMFTR) under grant no. 01IS24085C (OPENHAFM), under the grant 16HPC117K (MINERVA) and under the grant no. 01IS22094B (WestAI - AI Service Center West). We also acknowledge compute resources provided via WestAI compute grant "Measuring and enhancing advanced reasoning capabilities of foundation models via local model deployment" (westai0007, westai0066) at Juelich Supercomputing Center (JSC).

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

# Supplementary.

## A   Additional details on performed experiments

Here we give further details on the procedures around the executed experiments.

### A.1   Models selected for experiments

To provide overview over origin of core tested models used for the AIW experiments, we list those in Suppl Tab. 2. All tested models use same default inference hyperparameters, $T = 0.1$, *top-p* $= 1.0$ (we executed control experiments to check that various settings do not change the main pattern in the observed behavior). The output of standard LLMs was limited to 2048 tokens, and as evident from Suppl Fig. 28, most observed responses stayed well below this limit. When testing recent reasoning models, the limit was raised to 32k or 43k, depending on model type.

### A.2   Evaluating model responses

The formatting instruction makes it possible to extract for each prompting trial whether a model has provided a correct answer to the AIW problem posed in the input. We can interpret then any number $n$ of collected responses as executing $n$ trials given a particular prompt for a given model ($n$ - number of Bernoulli trials), observing in each $i-$th trial a Bernoulli variable $X_i = \{0, 1\}$. We interpret the number of correct responses $X = \sum_i X_i$ as random variable following a Beta-Binomial distribution with unknown probability $p$ of correct response that we also treat as random variable that comes from a Beta distribution, i.e. $p \sim Beta(\alpha, \beta)$, where $\alpha$ and $\beta$ are parameters of the Beta distribution. To obtain plots showing correct response ratios, we would like to estimate Beta distribution underlying $p$, and for that, we first estimate the mean of $p$ and its variance from the collected observations. To estimate $\hat{p}$, we use the formula for estimating the mean of $p$ for a binomial distribution: $\hat{p} = X/n$ (i.e. as a proportion of successes). We can report the estimate $\hat{p}$ as the estimate of the correct response rate of a given model and also, compare the correct response rates of various tested models. Moreover, we can estimate the variance of the probability of a correct response by using the following formula:

$$\text{var}\left(\frac{1}{n}\sum_{i=1}^{n} X_i\right) = \frac{1}{n^2}\sum_{i=1}^{n}\text{var}(X_i) = \frac{n\text{var}(X_i)}{n^2} = \frac{\text{var}(X_i)}{n} = \frac{p(1-p)}{n} \tag{1}$$

The estimates of the variance and the standard deviation of $p$ can be thus obtained by using $\hat{p}$ as $\frac{\hat{p}(1-\hat{p})}{n}$ and $\sqrt{\frac{\hat{p}(1-\hat{p})}{n}}$ respectively. Using the estimated variance and mean of $p$, we can use the following relations for the variance: $\left(\sigma^2 = \frac{n\alpha\beta(\alpha+\beta+n)}{(\alpha+\beta)^2(\alpha+\beta+1)}\right)$ and the mean $\left(\mu = \frac{\alpha}{\alpha+\beta}\right)$ in order to obtain $\alpha$ and $\beta$ parameters for the Beta distribution. To simulate data for the plots, we draw $N$ random samples corresponding to correct and incorrect responses using the estimated distribution of $p$ and obtain the plots showing performance on the task for various models of interest as a full distribution of the respective $p$.

## B   Problem versions, prompt types and variations

**AIW Problem.** To measure models' sensitivity to problem irrelevant variations and thus probe the zero-shot generalization, we use problem templates, as described in Sec. 2.1. We create controlled problem variations by varying natural numbers $N, M \leq 7$, obtaining AIW variations 1-4 (see Suppl. Tab. 3) where problem structure and difficulty remains unchanged, which should ensure variations being irrelevant for problem handling. The resulting variations are as following:

> Variation 1. Alice has 3 brothers and she also has 6 sisters. [Correct answer: 7]
> Variation 2. Alice has 2 sisters and she also has 4 brothers. [Correct answer: 3]

Table 2: Names, origin and versioning of core test models used in the experiments.

| Name | Origin | Released | Open Weights | Sources |
|---|---|---|---|---|
| o3-mini | OpenAI | 31.01.2025 | No | OpenAI (2025a;b) |
| o1-2024-12-17 | OpenAI | 17.12.2024 | No | OpenAI (2024c;d) |
| o1-preview | OpenAI | 12.09.2024 | No | OpenAI (2024g;h) |
| o1-mini | OpenAI | 12.09.2024 | No | OpenAI (2024e;f) |
| GPT-4o-2024-05-13 | OpenAI | 13.05.2024 | No | Achiam et al. (2023); OpenAI (2024a;b) |
| GPT-4-turbo-2024-04-09 | OpenAI | 09.04.2024 | No | Achiam et al. (2023); OpenAI (a) |
| GPT-4-0125-preview | OpenAI | 25.01.2024 | No | Achiam et al. (2023); OpenAI (a) |
| GPT-4-0613 | OpenAI | 13.06.2023 | No | Achiam et al. (2023); OpenAI (a) |
| GPT-3.5-turbo-0125 | OpenAI | 24.01.2024 | No | OpenAI (2022; b;c) |
| Claude-3-5-sonnet-20240620 | Anthropic | 21.06.2024 | No | Anthropic (2024b) |
| Claude-3-opus-20240229 | Anthropic | 04.03.2024 | No | Anthropic (2024a;c) |
| Claude-3-sonnet-20240229 | Anthropic | 04.03.2024 | No | Anthropic (2024a;c) |
| Claude-3-haiku-20240307 | Anthropic | 04.03.2024 | No | Anthropic (2024a;c) |
| Gemini 1.0 Pro | Google | 06.12.2023 | No | Pichai & Hassabis (2023); Team et al. (2023) |
| Gemini 1.5 Pro | Google | 16.02.2024 | No | Pichai & Hassabis (2024); Reid et al. (2024) |
| gemma-2b/7b-it | Google | 05.04.2024 (v1.1) | Yes | Google (2024a;b) |
| Mistral-large-2402 | Mistral AI | 26.02.2024 | No | Mistral-AI-Team (2024c;d) |
| Mistral-medium-2312 | Mistral AI | 23.12.2023 | No | Mistral-AI-Team (2024c;d) |
| Mistral-small-2402 | Mistral AI | 26.02.2024 | No | Mistral-AI-Team (2024c;d) |
| open-mixtral-8x22b-instruct-v0.1 | Mistral AI | 17.04.2024 | Yes | Mistral-AI-Team (2024c;a) |
| open-mixtral-8x7b-instruct-v0.1 | Mistral AI | 11.12.2023 | Yes | Mistral-AI-Team (2024c;b) |
| open-mistral-7b-instruct-v0.2 | Mistral AI | 11.12.2023 | Yes | Jiang et al. (2023); Mistral-AI-Team (2024c; 2023) |
| Command R+ | Cohere | 04.04.2024 | Yes | Cohere (2024a;b) |
| Dbrx Instruct | Mosaic | 27.03.2024 | Yes | Mosaic |
| Llama 2 7B, 13B, 70B Chat | Meta | 18.07.2023 | Yes | Meta (2023); Touvron et al. (2023b) |
| Llama 3 8B, 70B Chat | Meta | 18.04.2024 | Yes | Meta (2024a;c) |
| Llama 3.1 8B - 405B Instruct | Meta | 23.07.2024 | Yes | Meta (2024b;c) |
| Qwen 1.5 1.8B - 72B Chat | Alibaba | 04.02.2024 | Yes | Bai et al. (2023); Alibaba (2024a) |
| Qwen 2 05B - 72B Instruct | Alibaba | 07.06.2024 | Yes | Alibaba (2024c) |
| Qwen 2.5 0.5B - 72B Instruct | Alibaba | 19.09.2024 | Yes | Alibaba (2024b); Qwen et al. (2025) |
| DeepSeek R1 671B | DeepSeek | 20.01.2025 | Yes | Guo et al. (2025) |
| DeepSeek v3 671B | DeepSeek | 26.12.2024 | Yes | Liu et al. (2024) |

Variation 3. Alice has 4 sisters and she also has 1 brother. [Correct answer: 5]
Variation 4. Alice has 4 brothers and she also has 1 sister. [Correct answer: 2]

Question: How many sisters does Alice's brother have?

**Prompt types.** For testing the model dependence on input prompt type when solving AIW and AIW Light problems to see whether our observations hold independent of prompting, we used three main prompt types - STANDARD (original prompt with answer formatting instructions), THINKING (prompt that encourages thinking with answer formatting instructions) and RESTRICTED (prompt that instructs model to output only formatted answer and nothing else). THINKING v2 prompt type is a minor variation of THINKING type that just adds "step by step" after already existing "think carefully" phrasing (control experiments show that THINKING and THINKING v2 are equivalent in terms of observed performance, so we use both interchangeably, Suppl. Fig. 8b).

For each trial, models receive thus an input that has a form *<instantiated-template> <prompt-type>*, where *<instantiated-template>* is template with substituted numbers instantiating one of problem variations 1-4 (1-6 for some of experiments in Appendix) containing the question, followed by *<prompt-type>* with task and output instructions corresponding to one of prompt types as described above. See Suppl. Fig. **??** for an illustratory example showing GPT-4 evaluation.

All AIW problem versions employed in this work use the same construction as described above, differing either in the main part or in the question of their template. We make the correct answers match across variations of different problem versions to ensure the source of variation is confined to the problem input only and no further source of variation enters through different numbers in the outcomes.

### B.1 Control AIW Light problems

To control for models struggling either with basic family relations structure handling or with executing arithmetic operations in frame of the posed AIW problem, we make various versions of AIW problem - AIW Light Family, AIW Light Arithmetic Siblings and AIW Light Arithmetic Total Girls, relying on main structure of AIW original template while adapting the question accordingly to operations to be tested, as described in Sec. 2.1

**AIW Light Arithmetic Siblings**. The problem version has following variations 1-4:

> Variation 1. Alice has 3 brothers and she also has 4 sisters. [Correct answer: 7]
> Variation 2. Alice has 2 sisters and she also has 1 brother. [Correct answer: 3]
> Variation 3. Alice has 4 sisters and she also has 1 brother. [Correct answer: 5]
> Variation 4. Alice has 1 brother and she also has 1 sister. [Correct answer: 2]

> Question: *How many siblings does Alice have?*

**AIW Light Family**. The problem version has following variations 1-4:

> Variation 1. Alice has 7 brothers and she also has 3 sisters. [Correct answer: 7]
> Variation 2. Alice has 4 sisters and she also has 3 brothers. [Correct answer: 3]
> Variation 3. Alice has 2 sisters and she also has 5 brothers. [Correct answer: 5]
> Variation 4. Alice has 2 brothers and she also has 3 sisters. [Correct answer: 2]

> Question: *How many brothers does Alice's sister have?*

**AIW Light Arithmetic Total Girls**. The problem version has following variations 1-4:

> Variation 1. Alice has 6 sisters and she also has 3 brothers. [Correct answer: 7]
> Variation 2. Alice has 2 sisters and she also has 4 brothers. [Correct answer: 3]
> Variation 3. Alice has 4 sisters and she also has 1 brother. [Correct answer: 5]
> Variation 4. Alice has 1 sister and she also has 4 brothers. [Correct answer: 2]

> Question: *How many girls are there in total?*

See Suppl. Tab. 4, 5, 6 for examples with full prompt versions for each problem and its variations[4].

## B.2 Further AIW problem versions

**AIW Extended.** As described in Sec. 2.1.2, AIW Extended (AIW Ext) employs Alice and Bob as two entities, keeping the template structure close to the AIW original The resulting variations are as following:

> Alice and Bob are sister and brother.
>
> Variation 1. Alice has 3 sisters and Bob has 6 brothers. [Correct answer: 7]
> Variation 2. Alice has 2 sisters and Bob has 2 brothers. [Correct answer: 3]
> Variation 3. Alice has 1 sister and Bob has 4 brothers. [Correct answer: 5]
> Variation 4. Alice has 3 sisters and Bob has 1 brother. [Correct answer: 2]
>
> Question: How many brothers does Alice have?

In further experiments, we also use additional variations 5-6 as following:

> Variation 5. Alice has 2 sisters and Bob has 3 brothers. [Correct answer: 4]
> Variation 6. Alice has 3 sisters and Bob has 5 brothers. [Correct answer: 6]
>
> Question: How many sisters does Alice's brother have?

**AIW Friends.** As mentioned in Sec. 2.1.2, in this problem version, we abandon the family frame setting. Instead, we use male and female friends in problem formulation. Note the problem structure is still related to AIW original (as brothers and sisters are male and female siblings). We use an additional condition to ensure there is no common sense ambiguity in this problem version. The resulting variations are as following:

> Variation 1. Alice has 3 male friends and she also has 6 female friends. [Correct answer: 7]
> Variation 2. Alice has 2 female friends and she also has 4 male friends. [Correct answer: 3]
> Variation 3. Alice has 4 female friends and she also has 1 male friend. [Correct answer: 5]
> Variation 4. Alice has 4 male friends and she also has 1 female friend. [Correct answer: 2]
>
> All mentioned persons are friends with each other and have no other friends aside.
> Question: How many female friends does male friend of Alice have?

In further experiments, we also use additional variations 5-6 as following:

> Variation 5. Alice has 2 male friends and she also has 3 female friends. [Correct answer: 4]
> Variation 6. Alice has 5 female friends and she also has 3 male friends. [Correct answer: 6]
>
> Question: How many female friends does male friend of Alice have?

The two problem versions AIW Ext and AIW Friends have similar problem structure and difficulty to AIW original. They thus can be also seen as control to ensure our observations are consistent among various problem formulations that have similar problem difficulty (in contrast to AIW Light control problems which are designed to test low level operations involved, such that their templates pose problems of lower difficulty than AIW original). See Suppl. Tab. 9, 10 for examples with full prompt versions for each problem and its variations.

### B.2.1 Harder problem versions

To test what happens if the simple AIW problem is further extended towards a substantially harder difficulty level, providing further challenge for the tested models, we construct further problem versions of higher

---

[4]All prompts and their IDs available at `https://github.com/LAION-AI/AIW/blob/main/prompts/prompts.json`

difficulty while keeping same relational logic structure appeal and conduct experiments as described in following.

**AIW+ problem**. We constructed an AIW+ problem that features additional hierarchy and distractors when describing relational family structure (see Suppl. Sec **??** for full formulation). AIW+ problem template has following form: *"Alice has $M$ sisters and $N$ brothers in total. Her mother has 2 brothers. She also has 1 sister who does not have children and who has $X$ nephews and nieces in total. Alice's father has 2 sisters. He also has a brother who has $Y$ nephews and nieces in total, and who also has $K$ [sons/daughters]. How many cousins does Alice's sister have?"*.

The solution to AIW+ problem is harder to obtain than the solution to AIW original with its much simpler structure. Solving AIW+ requires taking different paternal sides, that of mother and father, and calculating the number of cousins, taking care of subtracting Alice and her siblings, and summing up the total number of cousins from both sides. Still, this problem is arguably far from olympiad or university graduate level, as it requires just using provided numbers and careful execution of elementary arithmetic operations on straightforward path to solution. The correct solution is given by $C = (X-(M+N+1))+(Y-(M+N+1)+K)$. We follow again our approach to create variations by instantiating numbers $N, M, X, Y, K$ in problem template to obtain problem instances of same problem structure and difficulty (see also Suppl. Tab. 11).

We show a full example of correct solution with instantiated numbers for AIW+ variation 1 ($N = M = 1, X = 6, Y = 5, K = 2$), which corresponds to problem instance as following:

> Variation 1. Alice has 1 sister and 1 brother in total. Her mother has 2 brothers. She also has 1 sister who does not have children and who has 6 nephews and nieces in total. Alice's father has 2 sisters. He also has a brother who has 5 nephews and nieces in total, and who also has 2 sons. [Correct answer: 7]

> Question: How many cousins does Alice's sister have?

Here, we have on the mother side: 6 (total nephews and nieces) - 3 (Alice and her siblings) = 3 cousins; on the father side: 5 (total nephews and nieces) + 2 (sons of the father's brother) - 3 (Alice and her siblings) = 4 cousins; summing up $3 + 4 = 7$ cousins which Alice and any of her siblings share.

**AIW Colleague Circles**. Evidencing strong performance of o1-preview on AIW+, we design a further problem with harder difficulty level than AIW original to test whether we can observe same breakdown patterns on the strongest model. AIW like problems can be understood as problems on graphs, featuring entities, properties and relationships that define sets. To increase problem difficulty, we depart from simple connectivity that was characteristic of the AIW original and introduce circles of colleagues where all-to-all connectivity defines a circles, while some entities have connections to outside, make those entities hubs connecting circles.

AIW Colleague Circles problem template has following form: *"Alice has 3 male colleagues and she also has $M$ female colleagues in total. All these mentioned persons in the circle around Alice are colleagues of each other. Bob has 2 female colleagues and 1 male colleague in total. All these mentioned persons in the circle around Bob are colleagues of each other. The people in the circle around Bob do not have other colleagues aside - with the only exception of Matilda. She is colleague of Bob, being part of Bob's circle, and she is also colleague of Alice, being part of Alice's circle. All the mentioned persons have no colleagues beyond the already described group of people. How many female colleagues does Matilda have?"*.

Again, the solution to AIW Colleague Circles problem is harder to obtain than the solution to AIW original with its much simpler structure. Same as AIW+, the problem is though still far from olympiad or graduate level, which advanced SOTA LLMs often claim to master robustly. The correct solution is given by $C = M+1$. We follow again our approach to create variations by instantiating number $M$ in problem template to obtain problem instances of same problem structure and difficulty (note we use only alteration of a single number here to create variations 1-6, corresponding to instantiating $M = 6, 2, 4, 1, 3, 5$ with corresponding correct answers $C = 7, 3, 5, 2, 4, 6$, see also Suppl. Tab. 12).

Table 3: AIW main variations 1-4, prompt types and correct answers overview.

| Var. | Prompt | Type/Answer | ID |
|---|---|---|---|
| 1 | Alice has 3 brothers and she also has 6 sisters. How many sisters does Alice's brother have? Solve this problem and provide the final answer in following form: "### Answer: ". | STANDARD / 7 | 55 |
| 1 | Alice has 3 brothers and she also has 6 sisters. How many sisters does Alice's brother have? Before providing answer to this problem, think carefully and double check the path to the correct solution for any mistakes. Provide then the final answer in following form: "### Answer: ". | THINKING / 7 | 57 |
| 1 | Alice has 3 brothers and she also has 6 sisters. How many sisters does Alice's brother have? To answer the question, DO NOT OUTPUT ANY TEXT EXCEPT following format that contains final answer: "### Answer: ". | RESTRICTED / 7 | 53 |
| 2 | Alice has 2 sisters and she also has 4 brothers. How many sisters does Alice's brother have? Solve this problem and provide the final answer in following form: "### Answer: ". | STANDARD / 3 | 56 |
| 2 | Alice has 2 sisters and she also has 4 brothers. How many sisters does Alice's brother have? Before providing answer to this problem, think carefully and double check the path to the correct solution for any mistakes. Provide then the final answer in following form: "### Answer: ". | THINKING / 3 | 58 |
| 2 | Alice has 2 sisters and she also has 4 brothers. How many sisters does Alice's brother have? To answer the question, DO NOT OUTPUT ANY TEXT EXCEPT following format that contains final answer: "### Answer: ". | RESTRICTED / 3 | 54 |
| 3 | Alice has 4 sisters and she also has 1 brother. How many sisters does Alice's brother have? Solve this problem and provide the final answer in following form: "### Answer: ". | STANDARD / 5 | 63 |
| 3 | Alice has 4 sisters and she also has 1 brother. How many sisters does Alice's brother have? Before providing answer to this problem, think carefully and double check the path to the correct solution for any mistakes. Provide then the final answer in following form: "### Answer: ". | THINKING / 5 | 64 |
| 3 | Alice has 4 sisters and she also has 1 brother. How many sisters does Alice's brother have? To answer the question, DO NOT OUTPUT ANY TEXT EXCEPT following format that contains final answer: "### Answer: ". | RESTRICTED / 5 | 65 |
| 4 | Alice has 4 brothers and she also has 1 sister. How many sisters does Alice's brother have? Solve this problem and provide the final answer in following form: "### Answer: ". | STANDARD / 2 | 69 |
| 4 | Alice has 4 brothers and she also has 1 sister. How many sisters does Alice's brother have? Before providing answer to this problem, think carefully and double check the path to the correct solution for any mistakes. Provide then the final answer in following form: "### Answer: ". | THINKING / 2 | 70 |
| 4 | Alice has 4 brothers and she also has 1 sister. How many sisters does Alice's brother have? To answer the question, DO NOT OUTPUT ANY TEXT EXCEPT following format that contains final answer: "### Answer: ". | RESTRICTED / 2 | 71 |
| 4 | Alice has 4 brothers and she also has 1 sister. How many sisters does Alice's brother have? Solve the problem by taking care not to make any mistakes. Express your level of confidence in the provided solution as precisely as possible. | CONFIDENCE / 2 | 11 |
| 3 | Alice has 4 sisters and she also has 1 brother. How many sisters does Alice's brother have? To solve the problem, approach it as a very intelligent, accurate and precise scientist capable of strong and sound reasoning. Provide the solution to the problem by thinking step by step, | SCIENTIST / 5 | 40 |

Table 4: AIW Light Arithmetic Siblings variations 1-4

| Var. | Prompt | Type/Answer | ID |
|---|---|---|---|
| 1 | Alice has 3 brothers and she also has 4 sisters. How many siblings does Alice have? Before providing answer to this problem, think carefully step by step and double check the path to the correct solution for any mistakes. Provide then the final answer in following form: "### Answer:". | THINKING v2 / 7 | 277 |
| 2 | Alice has 2 sisters and she also has 1 brother. How many siblings does Alice have? Before providing answer to this problem, think carefully step by step and double check the path to the correct solution for any mistakes. Provide then the final answer in following form: "### Answer:". | THINKING v2 / 3 | 278 |
| 3 | Alice has 4 sisters and she also has 1 brother. How many siblings does Alice have? Before providing answer to this problem, think carefully step by step and double check the path to the correct solution for any mistakes. Provide then the final answer in following form: "### Answer: ". | THINKING v2 / 5 | 279 |
| 4 | Alice has 1 brother and she also has 1 sister. How many siblings does Alice have? Before providing answer to this problem, think carefully step by step and double check the path to the correct solution for any mistakes. Provide then the final answer in following form: "### Answer:". | THINKING v2 / 2 | 280 |

Table 5: AIW Light Family variations 1-4

| Var. | Prompt | Type/Answer | ID |
|---|---|---|---|
| 1 | Alice has 7 brothers and she also has 3 sisters. How many brothers does Alice's sister have? Before providing answer to this problem, think carefully step by step and double check the path to the correct solution for any mistakes. Provide then the final answer in following form: "### Answer:". | THINKING v2 / 7 | 271 |
| 2 | Alice has 4 sisters and she also has 3 brothers. How many brothers does Alice's sister have? Before providing answer to this problem, think carefully step by step and double check the path to the correct solution for any mistakes. Provide then the final answer in following form: "### Answer:". | THINKING v2 / 3 | 272 |
| 3 | Alice has 2 sisters and she also has 5 brothers. How many brothers does Alice's sister have? Before providing answer to this problem, think carefully step by step and double check the path to the correct solution for any mistakes. Provide then the final answer in following form: "### Answer: ". | THINKING v2 / 5 | 273 |
| 4 | Alice has 2 brothers and she also has 3 sisters. How many brothers does Alice's sister have? Before providing answer to this problem, think carefully step by step and double check the path to the correct solution for any mistakes. Provide then the final answer in following form: "### Answer:". | THINKING v2 / 2 | 274 |

Table 6: AIW Light Arithmetic Total Girls variations 1-4

| Var. | Prompt | Type/Answer | ID |
|------|--------|-------------|-----|
| 1 | Alice has 6 sisters and she also has 3 brothers. How many girls are there in total? Before providing answer to this problem, think carefully step by step and double check the path to the correct solution for any mistakes. Provide then the final answer in following form: "**### Answer:**". | THINKING v2 / 7 | 343 |
| 2 | Alice has 2 sisters and she also has 4 brothers. How many girls are there in total? Before providing answer to this problem, think carefully step by step and double check the path to the correct solution for any mistakes. Provide then the final answer in following form: "**### Answer:**". | THINKING v2 / 3 | 344 |
| 3 | Alice has 4 sisters and she also has 1 brother. How many girls are there in total? Before providing answer to this problem, think carefully step by step and double check the path to the correct solution for any mistakes. Provide then the final answer in following form: "**### Answer: **". | THINKING v2 / 5 | 345 |
| 4 | Alice has 1 sister and she also has 4 brothers. How many girls are there in total? Before providing answer to this problem, think carefully step by step and double check the path to the correct solution for any mistakes. Provide then the final answer in following form: "**### Answer:**". | THINKING v2 / 2 | 346 |

## C   Model performance and behavior on AIW problem versions

Here we report further details on model evaluation, performance and behavior as observed on AIW problems. For executing experiments, we either use local model deployment via vLLM Kwon et al. (2024), or API based liteLLM Berri.AI (2024) and TogetherAI TogetherAI (2024).

For the full overview of average correct response rate including models that score zero, see Suppl. Fig. 6. For the statistics on number of trials conducted for each model and each prompt type, see Suppl. Fig. 27. For the statistics on the average output length across models and prompt types, see Suppl. Fig. 28. For models' behavior on RESTRICT prompt types, see Suppl. Fig. 7. For control comparison of THINKING v2 prompt type to THINKING and STANDARD, see see Suppl. Fig. 8.

### C.1   Persisting fluctuations on AIW versions by recent advanced language and reasoning models

Following Sec. 3.1, 3.2, we demonstrate here on further AIW versions that strong fluctuations also affect models released very recently, including large scale LLMs like DeepSeek-v3 and Llama 3.1 405 or advanced reasoning models like DeepSeek R1 and o1-mini that go beyond standard LLMs.

On AIW Ext (Suppl. Fig. 9), we observe o1-mini exhibiting strong fluctuations, while o1-preview handles the problem robustly. This also falsifies strong claims put forward by o1-mini to match or even outperform o1-preview which were made relying on standardized benchmarks. Large scale standard LLMs like DeepSeek 3 670B, Llama 3.1 405 and Qwen 2.5 72B also exhibit strong fluctuations and have lower correct response rates compared to o1-mini. Claude 3.5 Sonnet collapses strongly with low correct response rates.

Exposing models to AIW+, we observe further, even stronger collapse of performance also for those advanced models that were showing significant correct response rates for AIW problem (Fig. 10) For instance, for GPT-4/4o and Claude 3 Opus overall correct response rate averaged across variations stays below $p < 0.2$ (Fig. 10 (**A**)). Large-scale open weight SOTA models Llama 3.1 405B, DeepSeek-v3 671B and Qwen 2.5 72B are settled around $p = 0.1$ and below. Recent Claude 3.5 Sonnet is an outlier scoring higher up close to $p = 0.4$, without showing strong fluctuations we usually observe (Fig. 10 (**B**); see however Sec. on Claude 3.5 Sonnet breakdown - its performance on some versions of AIW might be due to exposure to AIW tasks in post training, as it appeared after first version of our public AIW release). To show that problem can be successfully handled, we also test here o1-preview that comes from the recent generation of reasoning models (which we treat as an exception; LLMs are conventionally understood as models pre-trained in purely

Table 7: AIW Alice Female Power Boost and AIW Original, variations 1-4, THINKING v2 prompt

| Var. | Prompt | Type/Answer | ID |
|---|---|---|---|
| 1 | *Alice is female* and has 3 brothers and she also has 6 sisters. How many sisters does Alice's brother have? Before providing answer to this problem, think carefully step by step and double check the path to the correct solution for any mistakes. Provide then the final answer in following form: "### Answer: ". | FEMALE BOOST / 7 | 193 |
| 1 | Alice has 3 brothers and she also has 6 sisters. How many sisters does Alice's brother have? Before providing answer to this problem, think carefully step by step and double check the path to the correct solution for any mistakes. Provide then the final answer in following form: "### Answer: ". | NO BOOST / 7 | 205 |
| 2 | *Alice is female* and has 2 sisters and she also has 4 brothers. How many sisters does Alice's brother have? Before providing answer to this problem, think carefully step by step and double check the path to the correct solution for any mistakes. Provide then the final answer in following form: "### Answer: ". | FEMALE BOOST / 3 | 197 |
| 2 | Alice has 2 sisters and she also has 4 brothers. How many sisters does Alice's brother have? Before providing answer to this problem, think carefully step by step and double check the path to the correct solution for any mistakes. Provide then the final answer in following form: "### Answer: ". | NO BOOST / 3 | 206 |
| 3 | *Alice is female* and has 4 sisters and she also has 1 brother. How many sisters does Alice's brother have? Before providing answer to this problem, think carefully step by step and double check the path to the correct solution for any mistakes. Provide then the final answer in following form: "### Answer: ". | FEMALE BOOST / 5 | 189 |
| 3 | Alice has 4 sisters and she also has 1 brother. How many sisters does Alice's brother have? Before providing answer to this problem, think carefully step by step and double check the path to the correct solution for any mistakes. Provide then the final answer in following form: "### Answer: ". | NO BOOST / 5 | 187 |
| 4 | *Alice is female* and has 4 brothers and she also has 1 sister. How many sisters does Alice's brother have? Before providing answer to this problem, think carefully step by step and double check the path to the correct solution for any mistakes. Provide then the final answer in following form: "### Answer: ". | FEMALE BOOST / 2 | 190 |
| 4 | Alice has 4 brothers and she also has 1 sister. How many sisters does Alice's brother have? Before providing answer to this problem, think carefully step by step and double check the path to the correct solution for any mistakes. Provide then the final answer in following form: "### Answer: ". | NO BOOST / 2 | 188 |

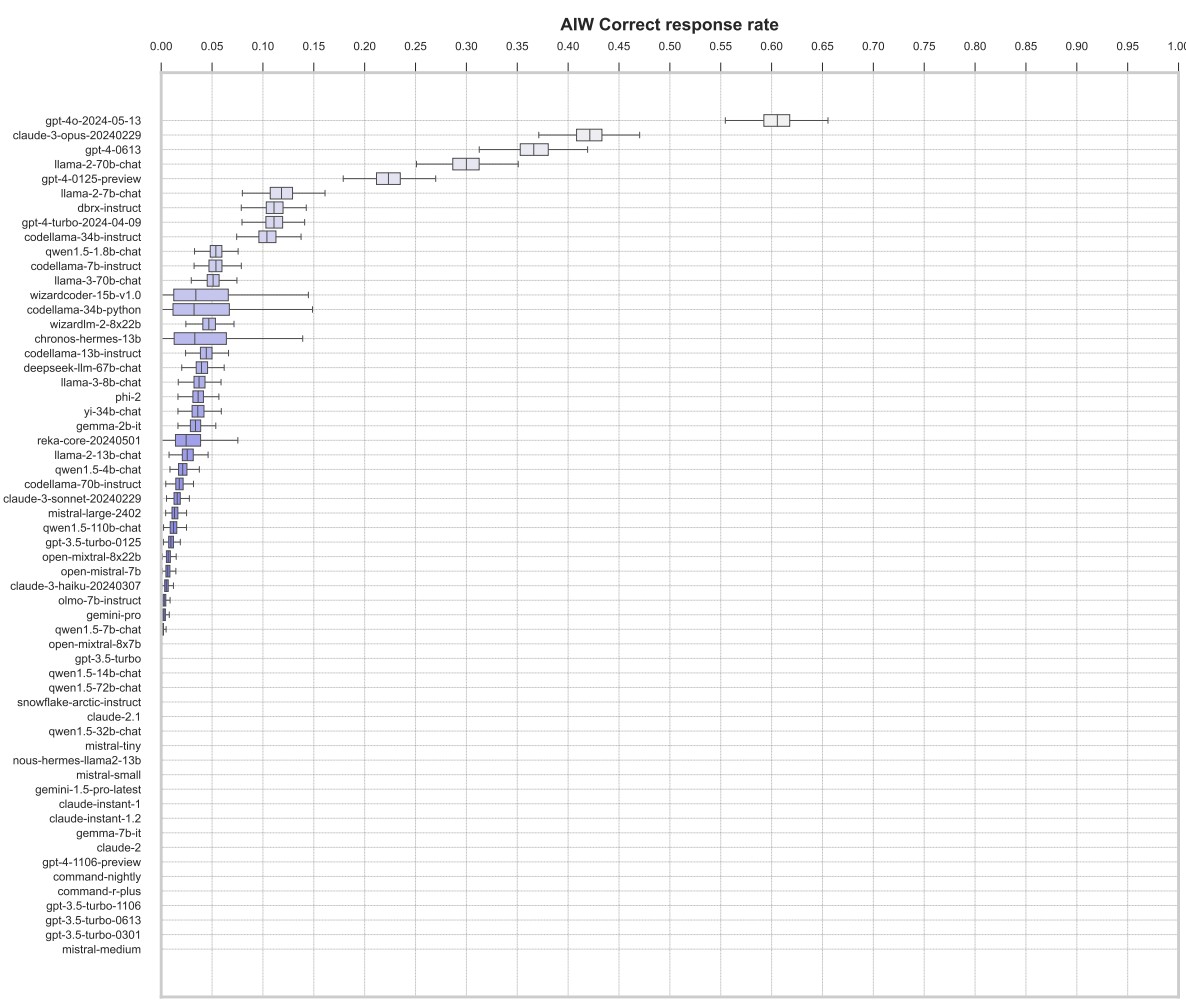

Figure 6: Collapse of most SOTA LLMs on AIW problem. AIW correct response rate across AIW variations averaged across all prompt types THINKING, STANDARD and RESTRICTED. Only 5 models manage to show rates above $p = 0.2$: GPT-4o, Claude 3 Opus, GPT-4-0613, Llama 2 70B Chat and GPT-4-0125-preview (GPT4-Turbo). Llama 2 70B Chat is the only open-weights model in this set. The rest either shows poor performance below $p = 0.15$, or even collapses entirely to 0. Among those models collapsing to 0 are many which claim strong function via high scores obtained on standardized benchmarks, eg larger scale GPT-3.5, Mixtral 8x7B and 8x22B, Command R Plus, Qwen 1.5 72B Chat and smaller scale Gemma-7b-it, Mistral Small and Mistral Medium.

Table 8: AIW Exaggerated Numbers variations 1-4, Thinking v2 prompt type.

| Var. | Prompt | Type/Answer | ID |
|------|--------|-------------|-----|
| 1 | Alice has 63 brothers and she also has 66 sisters. How many sisters does Alice's brother have? Before providing answer to this problem, think carefully step by step and double check the path to the correct solution for any mistakes. Provide then the final answer in following form: "**###** Answer:". | THINKING v2 / 67 | 653 |
| 2 | Alice has 62 sisters and she also has 64 brothers. How many sisters does Alice's brother have? Before providing answer to this problem, think carefully step by step and double check the path to the correct solution for any mistakes. Provide then the final answer in following form: "**###** Answer:". | THINKING v2 / 63 | 654 |
| 3 | Alice has 64 sisters and she also has 61 brother. How many sisters does Alice's brother have? Before providing answer to this problem, think carefully step by step and double check the path to the correct solution for any mistakes. Provide then the final answer in following form: "**###** Answer:". | THINKING v2 / 65 | 655 |
| 4 | Alice has 64 brothers and she also has 61 sister. How many sisters does Alice's brother have? Before providing answer to this problem, think carefully step by step and double check the path to the correct solution for any mistakes. Provide then the final answer in following form: "**###** Answer:". | THINKING v2 / 62 | 656 |

autoregressive manner. It is still unknown for o1 class of models whether RL on unknown amounts of synthetic data presumably of math and logic type is executed during pre-training or is rather a part of post-training). o1-preview is a clear exception and has robust performance close to 1 across all AIW+ variations. Remarkably, o1-mini coming presumably from the same model class does not show same robustness - its performance is comparable to standard LLM generation far below o1-preview, settled close to 0 (Fig. 10 (**A**)) and exhibiting fluctuations as usually observed in our study (Fig. 10 (**B**)).

On AIW Colleague Circle problem, we again observe the already familiar breakdown pattern for all tested models - including o1-preview (Fig. 11). While we saw o1-preview solving all the posed AIW versions so far robustly with high correct response rates close to 1 across all variations, here we observe rates below (p=0.8) and more importantly, strong fluctuations, eg. $p < 0.6$ on variation 1 vs $p > 0.8$ on variation 4 (Fig. 11 (**B**)). With only one single natural number varied in the problem template, this provides evidence that also larger scale models of o1 class are not robust to problem structure and difficulty preserving variations, hinting on generalization deficits in rather simple setting (as compared to claimed capabilities to robustly solve problems in graduate and olympiad level math setting, which are far above the level here). Interestingly, o1-mini does not undergo such a strong collapse as on AIW+ (Fig. 10), which might be further evidence for AIW Colleague Circles difficulty being rather moderate, as we observe o1-mini usually lagging far behind o1-preview on AIW problem versions tested here.

We provide also results on testing further reasoning models on both AIW+ and AIW Collegue Circles, complemeting Fig. 5 from main text.

As evident from Fig. 5, 12, and 13, despite their high scores on standardized reasoning benchmarks like MATH500, AIME24/25 and GPQA-diamond, most reasoning models still suffer from strong fluctuations across AIW problem variations. Notable exception is again o1-preview, although also this model exhibit significant fluctuations and decreased average correct response rates on AIW Circles Colleagues. This shows that claims of robust problem solving of olympiad or graduate level as signalled by strong performance on reasoning benchmarks are not sustainable, as AIW problems where models show lack of robustness are far below these levels. On the other hand, reasoning models indeed strongly improve in robustness and average correct response rates compared to tested conventional SOTA LLMs as evident from Fig. 5. Remarkable

Table 9: AIW Ext variations 1-4, Thinking v2 prompt type.

| Var. | Prompt | Type/Answer | ID |
|------|--------|-------------|-----|
| 1 | Alice and Bob are sister and brother. Alice has 3 sisters and Bob has 6 brothers. How many brothers does Alice have? Before providing answer to this problem, think carefully step by step and double check the path to the correct solution for any mistakes. Provide then the final answer in following form: "**###** Answer: ". | THINKING v2 / 7 | 264 |
| 2 | Alice and Bob are sister and brother. Alice has 2 sisters and Bob has 2 brothers. How many brothers does Alice have? Before providing answer to this problem, think carefully step by step and double check the path to the correct solution for any mistakes. Provide then the final answer in following form: "**###** Answer: ". | THINKING v2 / 3 | 266 |
| 3 | Alice and Bob are sister and brother. Alice has 1 sister and Bob has 4 brothers. How many brothers does Alice have? Before providing answer to this problem, think carefully step by step and double check the path to the correct solution for any mistakes. Provide then the final answer in following form: "**###** Answer: ". | THINKING v2 / 5 | 268 |
| 4 | Alice and Bob are sister and brother. Alice has 3 sisters and Bob has 1 brother. How many brothers does Alice have? Before providing answer to this problem, think carefully step by step and double check the path to the correct solution for any mistakes. Provide then the final answer in following form: "**###** Answer:". | THINKING v2 / 2 | 270 |
| 5 | Alice and Bob are sister and brother. Alice has 2 sisters and Bob has 3 brothers. How many brothers does Alice have? Before providing answer to this problem, think carefully step by step and double check the path to the correct solution for any mistakes. Provide then the final answer in following form: "**###** Answer:". | THINKING v2 / 4 | 455 |
| 6 | Alice and Bob are sister and brother. Alice has 3 sisters and Bob has 5 brothers. How many brothers does Alice have? Before providing answer to this problem, think carefully step by step and double check the path to the correct solution for any mistakes. Provide then the final answer in following form: "**###** Answer:". | THINKING v2 / 6 | 456 |

Table 10: AIW Friends variations 1-4

| Var. | Prompt | Type/Answer | ID |
|---|---|---|---|
| 1 | Alice has 3 male friends and she also has 6 female friends. All mentioned persons are friends with each other and have no other friends aside. How many female friends does male friend of Alice have? Before providing answer to this problem, think carefully step by step and double check the path to the correct solution for any mistakes. Provide then the final answer in following form: "### Answer: ". | THINKING v2 / 7 | 577 |
| 2 | Alice has 2 female friends and she also has 4 male friends. All mentioned persons are friends with each other and have no other friends aside. How many female friends does male friend of Alice have? Before providing answer to this problem, think carefully step by step and double check the path to the correct solution for any mistakes. Provide then the final answer in following form: "### Answer: ". | THINKING v2 / 3 | 580 |
| 3 | Alice has 4 female friends and she also has 1 male friend. All mentioned persons are friends with each other and have no other friends aside. How many female friends does male friend of Alice have? Before providing answer to this problem, think carefully step by step and double check the path to the correct solution for any mistakes. Provide then the final answer in following form: "### Answer: ". | THINKING v2 / 5 | 581 |
| 4 | Alice has 4 male friends and she also has 1 female friend. All mentioned persons are friends with each other and have no other friends aside. How many female friends does male friend of Alice have? Before providing answer to this problem, think carefully step by step and double check the path to the correct solution for any mistakes. Provide then the final answer in following form: "### Answer:". | THINKING v2 / 2 | 582 |
| 5 | Alice has 2 male friends and she also has 3 female friends. All mentioned persons are friends with each other and have no other friends aside. How many female friends does male friend of Alice have? Before providing answer to this problem, think carefully step by step and double check the path to the correct solution for any mistakes. Provide then the final answer in following form: "### Answer:". | THINKING v2 / 4 | 583 |
| 6 | Alice has 5 female friends and she also has 3 male friends. All mentioned persons are friends with each other and have no other friends aside. How many female friends does male friend of Alice have? Before providing answer to this problem, think carefully step by step and double check the path to the correct solution for any mistakes. Provide then the final answer in following form: "### Answer:". | THINKING v2 / 6 | 584 |

Table 11: AIW Plus variations 1-6.

| Var. | Prompt | Type/Answer | ID |
|------|--------|-------------|-----|
| 1 | Alice has 1 sister and 1 brother in total. Her mother has 2 brothers. She also has 1 sister who does not have children and who has 6 nephews and nieces in total. Alice's father has 2 sisters. He also has a brother who has 5 nephews and nieces in total, and who also has 2 sons. How many cousins does Alice's sister have? Before providing answer to this problem, think carefully step by step and double check the path to the correct solution for any mistakes. Provide then the final answer in following form: "**###** Answer: ". | THINKING V2/7 | 559 |
| 2 | Alice has 2 sisters and 1 brother in total. Her mother has 2 brothers. She also has 1 sister who does not have children and who has 6 nephews and nieces in total. Alice's father has 2 sisters. He also has a brother who has 4 nephews and nieces in total, and who also has 1 son. How many cousins does Alice's sister have? Before providing answer to this problem, think carefully step by step and double check the path to the correct solution for any mistakes. Provide then the final answer in following form: "**###** Answer: ". | THINKING V2/3 | 560 |
| 3 | Alice has 2 sisters and 1 brother in total. Her mother has 2 brothers. She also has 1 sister who does not have children and who has 7 nephews and nieces in total. Alice's father has 2 sisters. He also has a brother who has 5 nephews and nieces in total, and who also has 1 son. How many cousins does Alice's sister have? Before providing answer to this problem, think carefully step by step and double check the path to the correct solution for any mistakes. Provide then the final answer in following form: "**###** Answer: ". | THINKING V2/5 | 561 |
| 4 | Alice has 1 sister and 3 brothers in total. Her mother has 2 brothers. She also has 1 sister who does not have children and who has 6 nephews and nieces in total. Alice's father has 2 sisters. He also has a brother who has 5 nephews and nieces in total, and who also has 1 daughter. How many cousins does Alice's sister have? Before providing answer to this problem, think carefully step by step and double check the path to the correct solution for any mistakes. Provide then the final answer in following form: "**###** Answer: ". | THINKING V2/2 | 562 |
| 5 | Alice has 2 sisters and 1 brother in total. Her mother has 2 brothers. She also has 1 sister who does not have children and who has 6 nephews and nieces in total. Alice's father has 2 sisters. He also has a brother who has 5 nephews and nieces in total, and who also has 1 son. How many cousins does Alice's sister have? Before providing answer to this problem, think carefully step by step and double check the path to the correct solution for any mistakes. Provide then the final answer in following form: "**###** Answer: ". | THINKING V2/4 | 563 |
| 6 | Alice has 1 sister and 1 brother in total. Her mother has 2 brothers. She also has 1 sister who does not have children and who has 6 nephews and nieces in total. Alice's father has 2 sisters. He also has a brother who has 5 nephews and nieces in total, and who also has 1 daughter. How many cousins does Alice's sister have? Before providing answer to this problem, think carefully step by step and double check the path to the correct solution for any mistakes. Provide then the final answer in following form: "**###** Answer: ". | THINKING V2/6 | 564 |

Table 12: AIW Circles Colleagues variations (omitting variations 2-5 for better readability)

| Var. | Prompt | Type/Answer | ID |
|---|---|---|---|
| 1 | Alice has 3 male colleagues and she also has 6 female colleagues in total. All these mentioned persons in the circle around Alice are colleagues of each other. Bob has 2 female colleagues and 1 male colleague in total. All these mentioned persons in the circle around Bob are colleagues of each other. The people in the circle around Bob do not have other colleagues aside - with the only exception of Matilda. She is colleague of Bob, being part of Bob's circle, and she is also colleague of Alice, being part of Alice's circle. All the mentioned persons have no colleagues beyond the already described group of people. How many female colleagues does Matilda have? Before providing answer to this problem, think carefully step by step and double check the path to the correct solution for any mistakes. Provide then the final answer in following form: "### Answer: ". | THINKING V2/7 | 637 |
| 2 | ... | THINKING V2/3 | 638 |
| 3 | ... | THINKING V2/5 | 639 |
| 4 | ... | THINKING V2/2 | 640 |
| 5 | ... | THINKING V2/4 | 641 |
| 6 | Alice has 3 male colleagues and she also has 5 female colleagues in total. All these mentioned persons in the circle around Alice are colleagues of each other. Bob has 2 female colleagues and 1 male colleague in total. All these mentioned persons in the circle around Bob are colleagues of each other. The people in the circle around Bob do not have other colleagues aside - with the only exception of Matilda. She is colleague of Bob, being part of Bob's circle, and she is also colleague of Alice, being part of Alice's circle. All the mentioned persons have no colleagues beyond the already described group of people. How many female colleagues does Matilda have? Before providing answer to this problem, think carefully step by step and double check the path to the correct solution for any mistakes. Provide then the final answer in following form: "### Answer: ". | THINKING V2/6 | 642 |

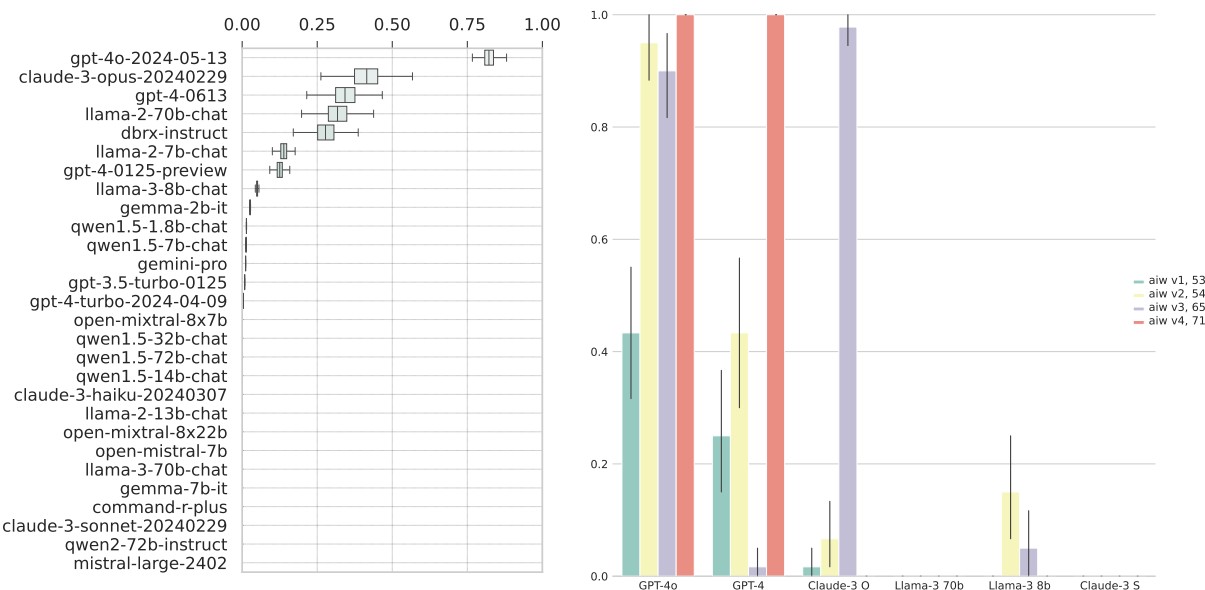

(a) Correct response rates for RESTRICTED prompt type, averaged across AIW var 1-4

(b) Strong fluctuations across AIW variations 1-4, RESTRICTED prompt type

Figure 7: Correct response rates on RESTRICTED prompt type. The prompt type enforcing to output only final answer without any further text was used as further control. **(a)** Correct response rates averaged over variations 1-4 resemble behavior with STANDARD and THINKING types, while looking at fluctuations across variations 1-4 in **(b)** reveals stronger models' lack of robustness compared to other prompt types (see for comparison Fig. 3). We thus used THINKING and THINKING v2 prompt types across main experiments not to put models into disadvantage on AIW testing. See also Suppl. Fig. 8

is that this improvement in robustness is also achieved by distilled reasoning models that use only SFT on reasoning traces in single stage post-training. Models like R1-Distilled-Llama-70B Guo et al. (2025) (distilled on 800k closed data), OpenThinker-32B Team (2025a;b) (distilled on OpenThoughts-114k, 3 epochs), S1.1-32B Muennighoff et al. (2025) (distilled on 1k DeepSeek-R1 data), LIMO-32B Ye et al. (2025) (distilled on 0.8k samples mix from real and DeepSeek-R1 generated data), having either Llama 3.3 or Qwen 2.5 as a base, are increasing their performance strongly compared to their base instruct models, reaching up to levels comparable with DeepSeek R1 671B, which uses both SFT and RL during multi stage training, also outperforming closed reasoning models like o1-mini. We observe remarkable similarity in the distribution shape of correct response rates between larger scale SFT only distilled models (S1.1 32B, OpenThinker 32B, R1-Llama-70B) and DeepSeek-R1 or o1-mini and o3-mini that use RL (Suppl. Fig. 13).

We thus can obtain further confirmation that AIW problem versions, despite being simple - far below olympiad or graduate level problems that advanced models claim to be able to handle via their performance on standardized benchmarks - can reveal lack of model robustness and generalization deficits, also when used for testing most recent model generation, including advanced reasoning models of o1 type. This shows potential for model ranking and construction of benchmarks that are capable to detect model weaknesses that remain undiscovered when testing via standardized benchmarks, which use more complex problems without though measuring performance on controlled variations of those.

## C.2 Comparing models of various scale

We compare open weights model families Qwen 2.5 (Fig. 15 and Llama 3.1 (Fig. 14), which offer a broad range of pre-training scales (Qwen 2.5 1.5B-72B and Llama 3.1 8B-405B model scales). From results obtained from testing on AIW original and AIW Ext, we observe expected dependence of performance on pre-training scale. In line with previous observations, small scale models show strong breakdown with correct response

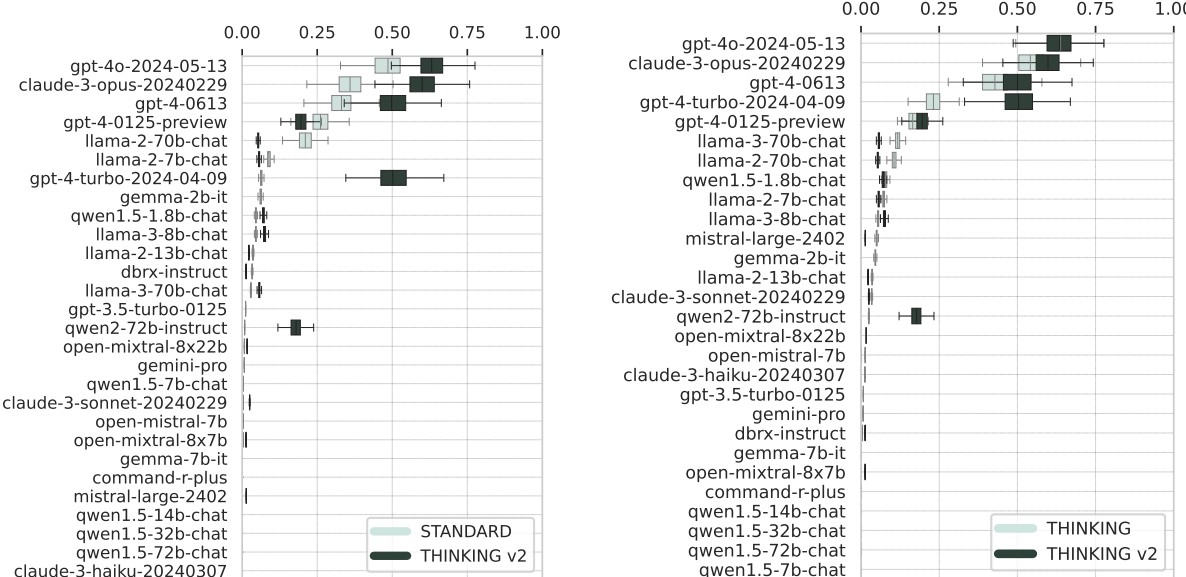

(a) Correct response rates THINKING v2 vs. STAN-DARD prompt type, averaged across AIW var 1-4

(b) Correct response rates THINKING v2 vs. THINKING prompt type, averaged across AIW var 1-4

Figure 8: Control comparison of correct response rates averaged across AIW variations 1-4. **(a)** THINKING v2 vs. STANDARD, **(b)** THINKING v2 vs. THINKING prompt types. THINKING provides better average correct response rates for tested models. We thus used THINKING prompt types for main and control experiments to ensure tested models are not disadvantaged on AIW problem. THINKING and THINKING v2 show highly similar behavior across tested models **(b)** and can be used interchangeably (THINKING v2 only difference to THINKING is the explicit phrasing "step by step", Suppl Tab. 7)

rates close to 0 across variations. With larger scale, models exhibit higher correct response rates, while showing strong performance fluctuations across problem structure & difficulty preserving variations. We see again strong performance fluctuation also across problem versions, for instance for Llama 3.1 405B having much higher correct response rate for AIW original compared to AIW Ext (Fig. 14), although both problems have highly similar structure. Similar to our observations on Claude 3.5 Sonnet (Fig. **??**), this might be again the case of training data containing AIW original instances, with strong diminished performance on AIW Ext pointing to generalization deficits.

For smaller scale models below 8B, the breakdown on AIW original and AIW Ext is so strong that correct response rates do no reflect differences in scale, most models at that small scales being close to 0. This is in line with our observations that the few models capable of showing significant non-zero correct response rate for the AIW problem are residing on the larger scales. Observing the performance on the AIW problem across various models, we see evidence that in general, smaller scale models (known to have been overtrained on large token budgets of $> 2T$ tokens) that score high on standardized reasoning benchmarks, creating illusion of having similar capabilities as larger scale models that have similar scores, suffer severe collapse on the AIW problem. No small scale model can even remotely approach the performance on AIW problems shown by better performers residing at larger scales. Better performing larger scale models, while exhibiting strong fluctuations on problem irrelevant variations, do show significant correct response rates across AIW problem variations, while smaller scale models drop across variations close to zero, staying far below $p = 0.1$ on average. This is valid for both conventional language and reasoning models.

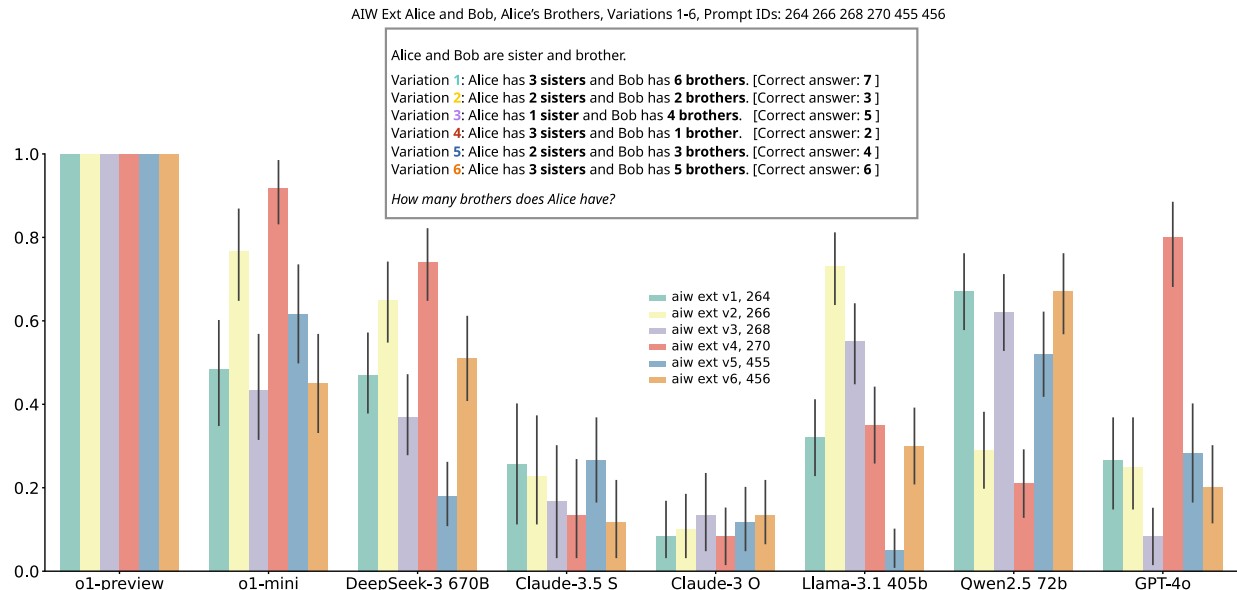

Figure 9: Strong fluctuations of correct response rates on variations of AIW Ext (a color per each variation 1-6) are also exhibited by recent reasoning models like o1-mini. Recent large-scale standard LLMs like DeepSeek 3 670B, Llama 3.1 405 and Qwen 2.5 72B show also strong fluctuations, with lower correct response rates compared to o1-mini. Claude 3.5 Sonnet and Claude 3 Opus collapse strongly with low correct response rates. Despite its simplicity, AIW Ext can thus reveal lack of model robustness and generalization deficits also in most recent SOTA LLMs that claim very strong function via standardized benchmarks. o1-preview is an exception that is able to handle the problem robustly, without any fluctuations.

### C.3 Debunking strong function claims

Here we show examples of debunking strong function claims put forward based on standardized benchmarks by testing and comparing models on AIW problems.

**NuminaMath-7B & claim of olympiad level problem solving.** We provide an example of debunking overblown claims in a case of NuminaMath-7B that was ranked 1st at the AIMO competition in July 2024, solving 29/50 private set problems of olympiad math level. The claim was widely put forward that the model is capable of solving high school olympiad math problems. AIW has arguably average elementary school level and does not require any advanced math knowledge. We tested NuminaMath-7B on AIW and observed a strong collapse of this model on AIW problem, with correct response rates close to 0 across AIW variations 1-4 (Fig. **??**). Using AIW Light control problems, we can also see that NuminaMath-7B can handle all the low level operations (elementary arithmetic, attribute binding, etc) and knowledge required to deal with family structure, ruling out that those are the issues. Using the AIW problem setting, we thus can contradict the strong claim of being capable to deal with olympiad level high school math

**o1-mini & claim of matching larger scale with smaller ones.** o1-mini was announced together with o1 and o1-preview as a smaller scale member of the new class of reasoning models. o1-mini was reported to obtain very strong scores on standardized benchmarks. Based on this, claims of strong function were put forward, specifically in comparison to the larger scale o1-preview. In original openAI announcement (Sep. 2024), o1-mini was reported to outperform larger-scale o1-preview on high school AIME math competition and coding tasks. This led to speculations that o1-mini can match or even outperform o1-preview as robust problem solver, a claim that is often put forward for models on smaller scales trained with substantial compute or obtained via distillation or other compression techniques from their larger scale counterparts.

We tested o1-mini and o1-preview using various AIW problem templates introduced here and the problem structure and difficulty preserving variations. As evident from conducted experiments, o1-mini shows signs of breakdown starting with AIW original, that become more severe on AIW Ext, where fluctuations become

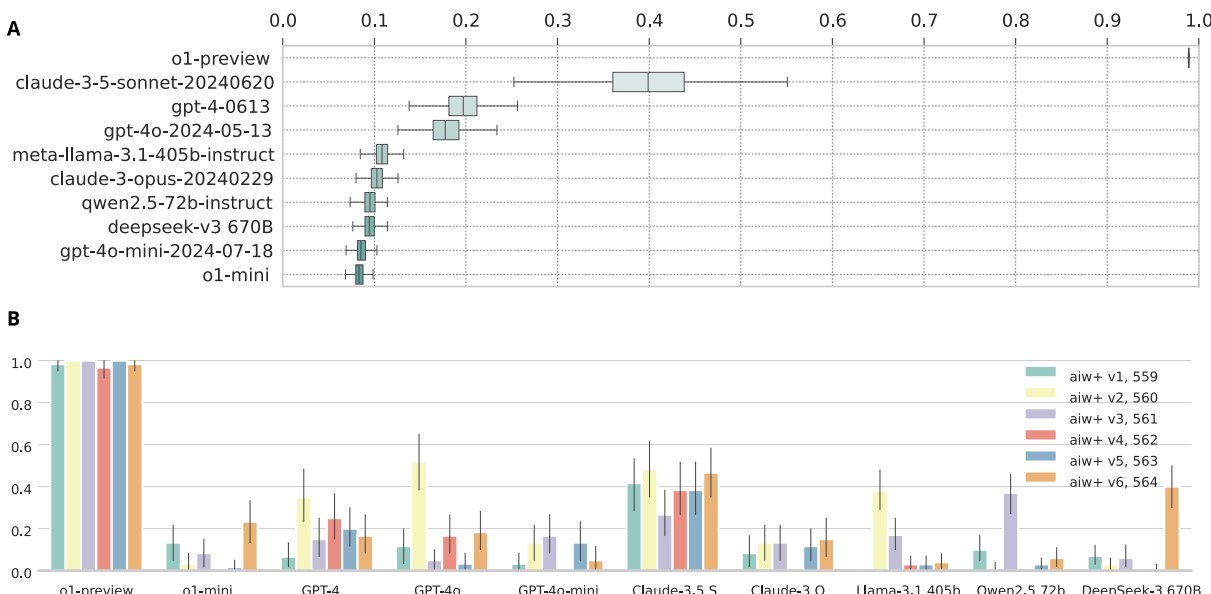

Figure 10: AIW+ correct response rates averaged over variations 1-6 (**A**) and fluctuations across variations (**B**). Most tested models undergo further collapse compared to AIW original. o1 preview as clear exception shows robust ability to solve AIW+ across all variations without fluctuations. o1-mini on contrary collapses close to 0, also showing fluctuations (eg on variation 6 vs others). AIW+ was made intentionally harder than simple AIW. However, models claiming strong function should be able to solve it, as it does not involve any higher level logic or math.

stronger, culminating with collapse on AIW+, where correct performance rate on most variations stays well below $p = 0.2$. o1-preview on contrary demonstrates robust problem solving with high correct response rates close to 1 without performance fluctuations across variations on all problem versions (see Fig. 11 for AIW Colleague Circles problem where o1-preview also exhibits breakdown pattern - while still clearly outperforming o1-mini) This is in line with our observations reported here wrt. to smaller scale models and also specifically for o1-mini in comparison to other models (Fig. 10, 11) , where o1-mini stays far beyond o1-preview, and importantly, showing much higher sensitivity to problem structure preserving variations, pointing to severe generalization deficits, in contrast to o1-preview.

Thus, AIW problem templates and the problem structure preserving variations offer a measurement technique that can reveal lack of robustness and model weaknesses in generalization and problem solving that remain undiscovered by standardized benchmarks. We think that conducted experiments can also serve as a vivid warning that many of the claims put forward for strong model functions cannot be trusted. Such claims often rely on benchmarks that overlook clear function deficits, and simple AIW problem templates with their variations offers a tool for systematic, reproducible stress testing and debunking of such claims.

### C.4 Frequency distribution of natural numbers on output and dominance of wrong responses.

To shed more light on modes of correct or wrong responses provided by the models when confronted with AIW problem variations, we show here frequency distribution for natural numbers on the output for AIW variations with higher and lower correct response rates.

As evident from the plots, in higher performance AIW variations (Suppl. Fig. 19), dominants peaks are often positioned on correct answer C=M+1, while for lower performance AIW variations (Suppl. Fig. 18), dominant peaks fall on wrong answer M. Further, for weaker models, distribution broadens, covering more numbers (eg in Llama 3 8b), while for better performers, responses concentrate on M and M+1, peaking on correct or wrong answer on depending on AIW variation. Remarkably, for lower performance AIW variations (Suppl. Fig. 18), performance cannot be rescued by major voting or by similar ensemble like strategies, as

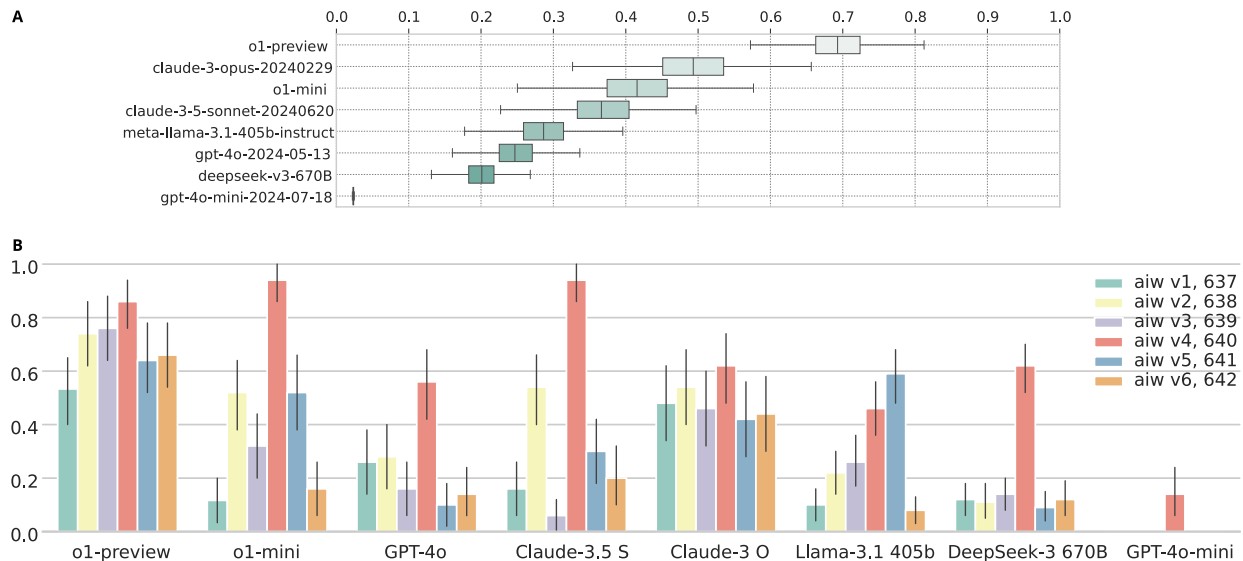

Figure 11: AIW Colleagues Circles correct response rates averaged over variations 1-6 (**A**) and fluctuations across variations (**B**). Same breakdown pattern is evident in the tested models - either overall low correct response rates or strong performance fluctuations across problem structure and difficulty preserving variations as observed on AIW original. Also o1 preview exhibits such fluctuations (eg. variation 4 vs variation 1, 5 or 6), providing evidence that also strongest models from o1 class are not robust, pointing to generalization deficits. o1-mini, which collapsed strongly on AIW+, exhibits here fluctuations much stronger than o1 preview, again providing evidence that o1-mini is much weaker than o1 preview, contrary to claims relying on standardized benchmarks. Strong fluctuations affect most models that obtain significant correct response rates on at least one of the variations, eg. GPT-4o, Claude 3.5 Sonnet, Llama 3.1 405B, DeepSeek v3 671B. GPT-4o-mini collapses entirely. AIW Colleague Circles is harder than simple AIW, but it does not involve any high difficulty logic or math. Models claiming strong function on level of graduate or olympiad math tasks should be able to solve it without fluctuations with higher correct response rates across variations.

peaks on wrong response numbers dominate clearly peaks on numbers for correct responses, which would still correspond to committing wrong answer when performing majority voting.

For the AIW Light problem versions used in control experiments, we observe as expected clear dominant peaks on the numbers corresponding for correct responses across all tested models (Suppl. Fig. 20, 21), as AIW Light problems are successfully solved across all their variations.

We note that distribution characteristics, eg concentration on numbers around the correct answer, height of the peaks, can be a further signature that reflects model's capability to handle the problem. More capable models retain dominant peaks on number corresponding to correct answer with smaller peaks on neighboring numbers, while weak models have large peaks on numbers corresponding to wrong answers or in general broad distribution across all natural numbers below 10. Computing scores from distribution shape can thus also enable model ranking.

## C.5   Standardized benchmarks failure

We observe failure of standardized reasoning benchmarks to properly reflect generalization and basic reasoning skills of SOTA LLMs by noting significant disparity between the model's performance on the AIW problem and the scores on conventional standardized benchmarks. All of the tested models report high scores on various standardized benchmarks that claim to test problem solving via reasoning, Our observations of SOTA models breaking down on the simple AIW problem hint that the benchmarks do not reflect deficits in generalization and basic reasoning of those models properly. We visualize this failure by plotting scores tested models obtain on wide-spread and accepted standardized benchmarks like MMLU versus the performance

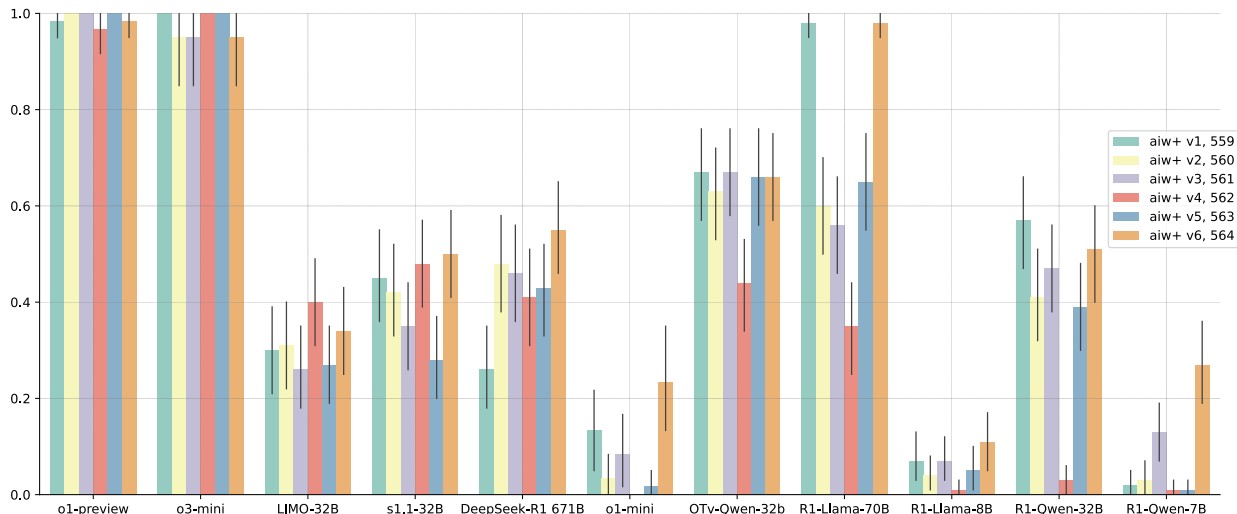

Figure 12: Strong fluctuations of correct response rates on variations of AIW Plus problem (a color per each variation 1-6) exhibited by recent reasoning models. With exception of o1-preview and o3-mini, most reasoning models that show strong performance on standardized reasoning benchmarks reveal inability to cope with the problem robustly when facing slight variations in problem template. Eg, correct response rate drops for R1-Llama 70B from close to 1.0 on variation 1 or 6 below 0.4 on variation 4, despite differences between the those being just instantiated numbers. o1-mini undergoes strong overall collapse. Distilled reasoning models (S1.1 32B, LIMO 32B, OpenThinker-Qwen-32B) perform on par with DeepSeek-R1 or outmatch it, despite using for distillation SFT only. Distilled models at larger scales (32B, 70B) perform significantly better than smaller scale 7B/8B models.

we observe on our proposed AIW problem. As strikingly evident from Fig. 22, there is a strong mismatch between high scores on MMLU reported by the models and the correct response rates they obtain on AIW. This mismatch and lack of differentiation makes it impossible for a given model to predict from its score on MMLU whether it will suffer breakdown on a simple problem like AIW, making the score unreliable for measuring core capabilities. Also model ranking fails, as models with similarly high MMLU scores claiming similar function level can have dramatic difference on simple AIW problem. For instance, models like Llama-3-70B, Mistral Large or GPT-4-Turbo come close with their MMLU score to GPT-4/Claude 3 Opus, hinting comparable capabilities, while settled in the crowd of high MMLU - low AIW score region (left upper part of Fig. 22) with most other models that achieve very low AIW performance close to 0. This also demonstrates that MMLU, while containing problems of arguably higher difficulty, does not properly reflect deficits in basic model function, as revealed by much simpler AIW problem.

This observation also holds on further standardized reasoning benchmarks like MATH, ARC-c, GSM8K and Hellaswag (Suppl Tab. 13). Also there we see strong mismatch between high benchmark scores reported by many models and the low correct response rates they obtain on AIW (in some cases 0 for models with high standardized benchmark scores), in Figures 24, 26, 23, 25.

We see thus that standardized benchmarks fail to properly reflect true model capabilities to generalize and reason - the majority of the tested models score high on standardized benchmarks, suggesting strong function, while showing extreme low correct response rates on simple AIW problem. Many of the models with high scores on standardized benchmarks cannot solve AIW problem a single time (e.g. Command R+ is unable to solve a single AIW problem instance, see Suppl Tab. 13). This discrepancy refutes the claim of standardized benchmarks to measure correctly current models' core functionality.

Table 13: Performance of tested models on MMLU, Hellaswag, ARC-c, GSM8k and AIW. Correct response rate averaged across AIW variations 1-4, across STANDARD, THINKING and RESTRICTED prompt types.

| Model | MMLU | Hellaswag | ARC-c | GSM8k | AIW Correct average resp. rate |
|---|---|---|---|---|---|
| gpt-4o-2024-05-13 | 0.89 | 95.3 | 95.9 | 95.00 | 0.65 |
| claude-3-opus-20240229 | 0.87 | 95.40 | 96.40 | 95.00 | 0.43 |
| gpt-4-0613 | 0.86 | 95.30 | 96.30 | 92.00 | 0.37 |
| gpt-4o-mini | 0.82 | 89.1 | 87.5 | 83.9 | 0.29 |
| llama-2-70b-chat | 0.64 | 85.90 | 64.60 | 56.80 | 0.30 |
| llama-2-7b-chat | 0.55 | 77.10 | 43.20 | 25.40 | 0.13 |
| dbrx-instruct | 0.74 | 88.85 | 67.83 | 67.32 | 0.11 |
| gpt-4-turbo-2024-04-09 | 0.80 | - | - | - | 0.10 |
| llama-3-8b-chat | 0.67 | 78.55 | 60.75 | 79.60 | 0.05 |
| llama-3-70b-chat | 0.80 | 85.69 | 71.42 | 93.00 | 0.05 |
| qwen1.5-1.8b-chat | 0.46 | 46.25 | 36.69 | 38.40 | 0.05 |
| gemma-2b-it | 0.38 | 71.40 | 42.10 | 17.70 | 0.04 |
| llama-2-13b-chat | 0.66 | 80.70 | 48.80 | 77.40 | 0.03 |
| qwen1.5-4b-chat | 0.56 | 51.70 | 40.44 | 57.00 | 0.02 |
| claude-3-sonnet-20240229 | 0.79 | 89.00 | 93.20 | 92.30 | 0.01 |
| mistral-large-2402 | 0.81 | 89.20 | 94.20 | 81.00 | 0.01 |
| gpt-3.5-turbo-0125 | 0.70 | 85.50 | 85.20 | 57.10 | 0.01 |
| gemini-pro | 0.72 | 84.70 | - | 77.90 | 0.01 |
| open-mixtral-8x22b | 0.78 | 89.08 | 72.70 | 82.03 | 0.01 |
| open-mistral-7b | 0.64 | 84.88 | 63.14 | 40.03 | 0.01 |
| qwen1.5-7b-chat | 0.62 | 59.38 | 52.30 | 62.50 | 0.01 |
| claude-3-haiku-20240307 | 0.75 | 85.90 | 89.20 | 88.90 | 0.00 |
| open-mixtral-8x7b | 0.72 | 87.55 | 70.22 | 61.11 | 0.00 |
| command-r-plus | 0.76 | 88.56 | 70.99 | 70.74 | 0.00 |
| qwen1.5-14b-chat | 0.69 | 63.32 | 54.27 | 70.10 | 0.00 |
| gemma-7b-it | 0.54 | 81.20 | 53.20 | 46.40 | 0.00 |
| qwen1.5-72b-chat | 0.77 | 68.37 | 65.36 | 79.50 | 0.00 |
| qwen1.5-32b-chat | 0.75 | 66.84 | 62.97 | 77.40 | 0.00 |

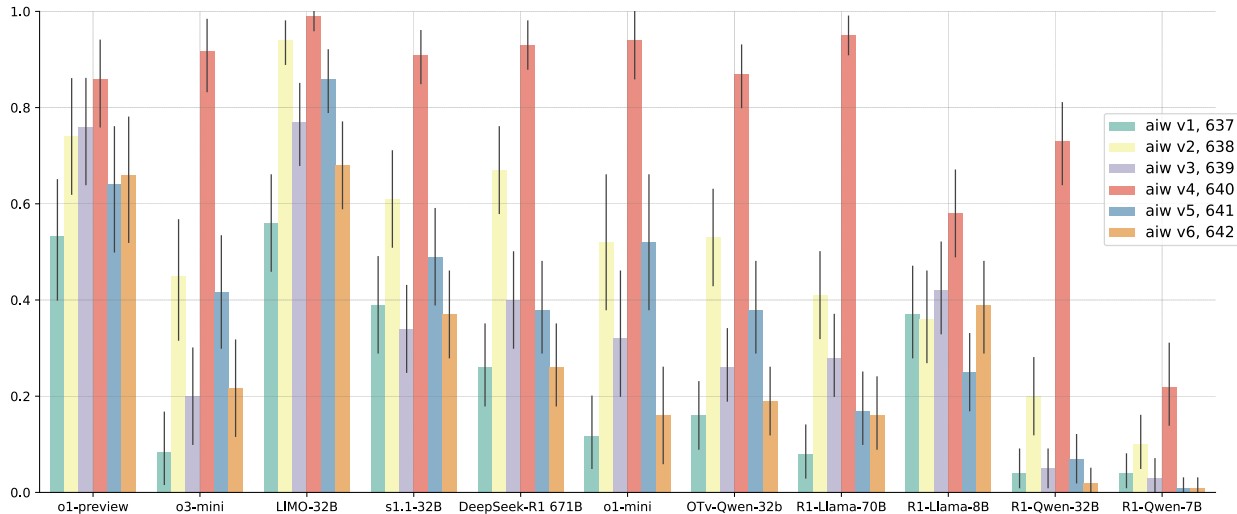

Figure 13: Strong fluctuations of correct response rates on variations of AIW Circles Colleagues problem (a color per each variation 1-6) exhibited by recent reasoning models, including top frontier models like o1-preview and o3-mini (medium). Fluctuations reveal that all reasoning models with strong performance on standardized reasoning benchmarks are unable to cope with the problem robustly, being sensitive to slight variations in problem template. Eg, correct response rate drops for o3-mini-medium from above 0.9 on variation 4 to below 0.1 on variation 1, despite differences between those being just instantiated numbers. Distilled reasoning models (S1.1 32B, LIMO 32B, OpenThinker-Qwen-32B) perform on par with DeepSeek-R1 or outmatch it, despite using for distillation SFT only. Distilled models at larger scales (32B, 70B) perform significantly better than smaller scale 7B/8B models (with exception of R1-Llama-8B vs R1-Llama-70B). Larger scale distilled models, DeepSeek-R1, o3-mini and o1-mini show remarkable similarity in the distribution of correct response rates across problem variations.

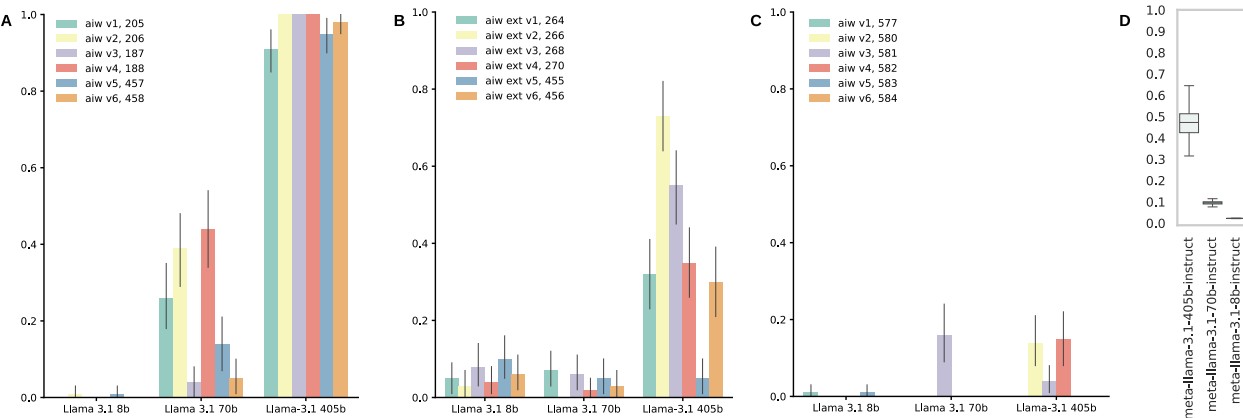

Figure 14: Model comparison and ranking, example of various scales of Llama 3.1 family. Models were tested on (**A**) AIW, (**B**) AIW Ext and (**C**) AIW Friends. Effect of scale is evident when comparing distribution of correct response rates across problem variations and correct response rate averaged across all problem versions in (**D**), hinting advantage of larger scale pre-training. Lack of model robustness is evident from strong collapse models suffer on AIW Friends, which is structurally similar to AIW and AIW ext.

## C.6 Comparing AIW testing with GSM Symbolic

Recently, Mirzadeh et al. (2024) made use of similar approach to create variations from templates of GSM8K problems. While also measuring fluctuations across problem structure preserving variations. this work does

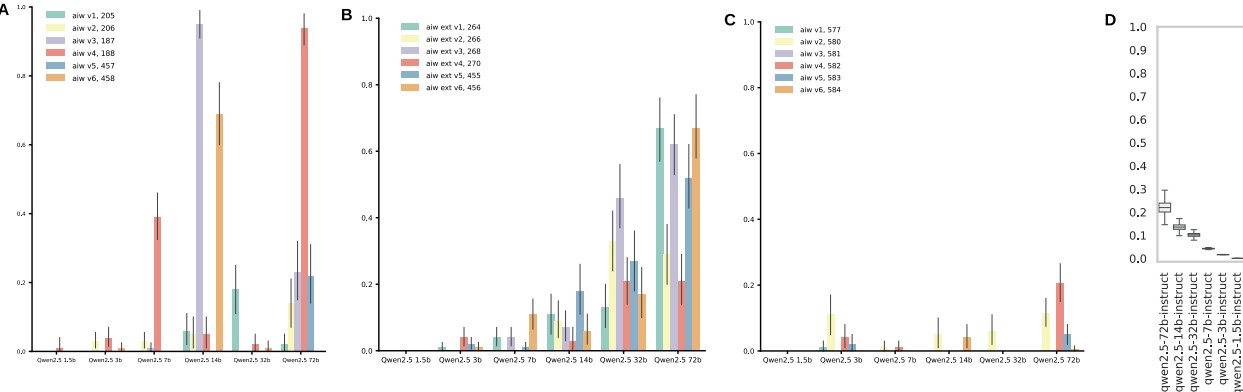

Figure 15: Model comparison and ranking, example of various scales of Qwen 2.5 model family. Models were tested on (**A**) AIW, (**B**) AIW Ext and (**C**) AIW Friends. Effect of scale is evident when comparing distribution of correct response rates across problem variations and correct response rate averaged across all problem versions in (**D**), hinting advantage of larger scale pre-training. Lack of model robustness is evident from strong collapse models suffer on AIW Friends, which is structurally similar to AIW and AIW ext.

not provide evidence that stronger models undergo function breakdown or exhibit generalization deficits on simple problems. The evidence obtained in GSM Symbolic suggests in fact that models like GPT-4o or Llama 3 8B can handle such problems well without exhibiting strong fluctuations or low correct response rates across variations (see A. Fig. 29). In contrast, we measure strong fluctuations also for advanced LLMs pretrained at largest scales (GPT-4/4o, Claude 3 Opus, Claude 3.5 Sonnet) using variations in very simple AIW template, which clearly shows lack of model robustness and generalization deficits also in those supposedly strong models. We hypothesize that inability to observe such strong fluctuations in GSM Symbolic might also be due to training set contamination, as GSM8k is a well known publicly available benchmark where test set might either accidentally or intentionally leak into training, or be used for synthetic data generation close to the test set mixed into training.

## D  Examples of correct and failed responses

We provide all collected model responses we obtained during this study in the collected_responses folder in the AIW repo. Here we also showcase some correct and incorrect answers as an example (see Suppl. Figs. 30, 33, 31, 32, 34).

## E  Confabulations and overconfident tone accompanying wrong answers

**Overconfident tone.** In ideal scenario, if LLM cannot correctly solve the AIW problem, it should at least be capable of expressing high uncertainty about the provided incorrect solution to the user. We used CONFIDENCE prompt type (see Suppl Tab. 3) for AIW problem to see how confident tested models are in their wrong solutions.

From our experiments we can see that LLMs most of the time express high certainty even if their answers are completely wrong, thus mediating strong confidence (see Fig. 35). The models also use highly persuasive tone to argue for the expressed certainty and correctness of the provided wrong solutions, using words like "highly confident", "definitive answer", or "accurate and unambiguous". We see also strong overconfidence expressed in multi-turn interactions with models, where user is insisting on solution provided being incorrect, and observe there high resistance of models to revise their decisions, which was already referred to as "stubbornness" in other works Zhang et al. (2024) (see Suppl Sec. **??** and also data provided in the AIW repo)

**Confabulations.** In our experiments we observe frequent tendency of those tested models that show strong reasoning collapse and produce frequent wrong answers for AIW problem to generate at the same

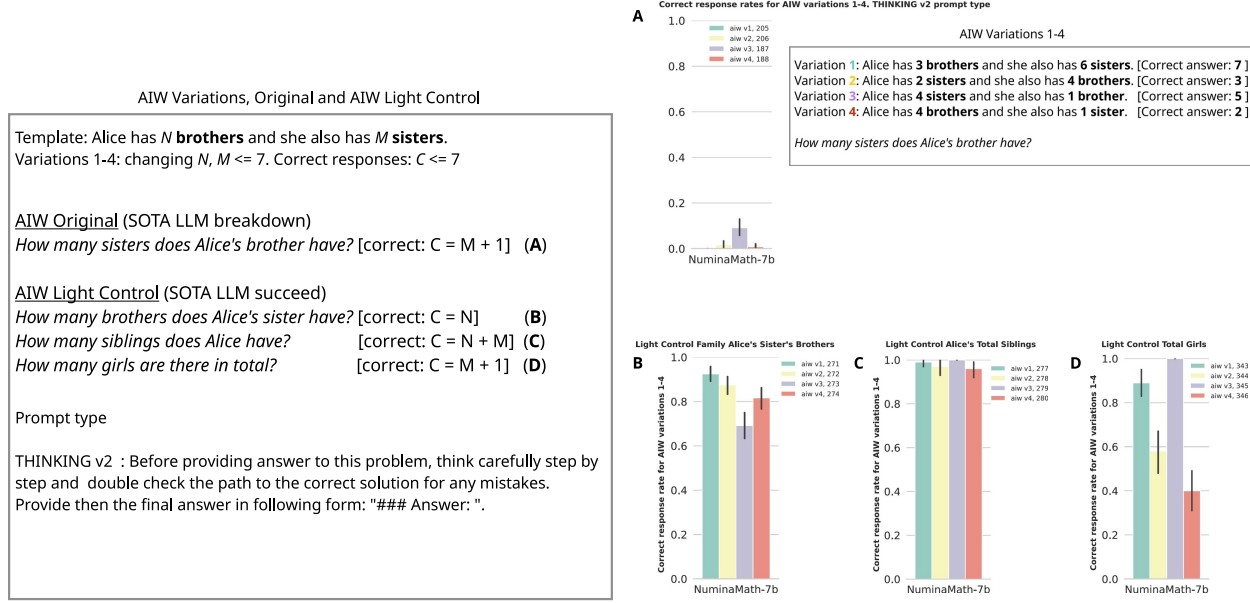

Figure 16: Testing NuminaMath-7B, which claimed olympiad high school level math problem solving via 1st rank in AIMO competition, by using AIW original and AIW Light control problems. (**A**) Very low correct response rates across AIW problem variations 1-4 (THINKING v2 prompt type). NuminaMath-7B suffers strong collapse on simple AIW problem that has average elementary school level. This reveals clear deficits in generalization and even basic reasoning, refuting the claim of strong function on special domain of math problems. For each AIW variation, 100 trials were executed to estimate correct response rate and its variance. AIW Light experiments test various operations and knowledge required for solving AIW (**B**) Asking for Alice's sister's brothers number (requires understanding entity "Alice's sister", binding female attribute to Alice and realizing Alice and her sisters share same brothers) (**C**) Asking for Alice's siblings number (requires understanding entity "siblings", accessing numbers of Alice's brothers and sisters, executing addition operation) (**D**) Asking for total girls number (requires binding female attribute to Alice via pronoun "she" and to her sisters, selecting and executing the correct arithmetic sum operation to count all the obtained girls). Across all AIW Light control problems, NuminaMath-7B obtains correct response rates much higher than for AIW original, some being close to 1. This proves that handling language, basic family structure, parsing numbers, and handling elementary arithmetics like counting are all intact and not the cause for failures in AIW (**A**). In (**D**), strong fluctuations despite only differences being instantiated numbers across variations of the same simple problem hint again on severe generalization deficits. The reason for the collapse on the AIW original problem is thus failure in inferring the problem structure, pointing to generalization and reasoning deficits, which is in contrast to claims made for NuminaMath-7B as a strong high school olympiad level math problem solver, based on AIMO competition benchmark, which did not reveal such flaws.

time persuasive sounding pseudo-explanations to back up their incorrect answers. We term here such pseudo-explanations confabulations, and present a selection of those as examples.

Such confabulations can contain mathematical calculations or other logic-like expressions and operations that make little or absolutely no sense given the problem to be solved, see examples for Olmo-7B, Fig. 36 and Command R+, Fig. 38.

Further confabulations make use of various social and cultural norm specific context to argue for the posed problem to be inappropriate to solve or to provide non-sense arguments for various incorrect answers. There are many such examples that we have observed, we present here only a small selection.

CodeLlama-70B-instruct for instance seems to be specifically prone to claim ethical or moral reasons for not addressing the problem correctly, in the presented example inventing out of nowhere a person with

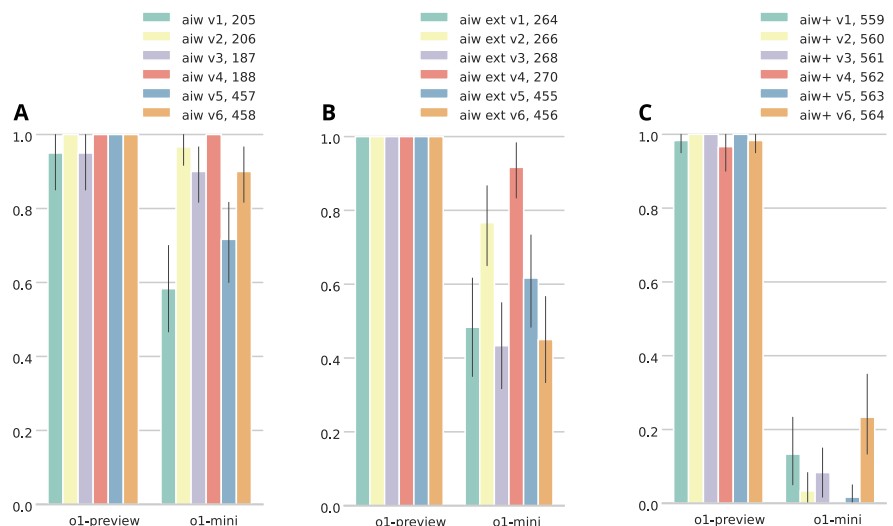

Figure 17: Testing o1-mini on the subject of the claims to match or outperform its larger scale counterpart, o1-preview. Comparing o1-preview and o1-mini using variations of AIW original (**A**), AIW Ext (**B**) and the harder AIW+. o1-mini shows progressive signs of breakdown, with fluctuations already visible in (**A**), becoming more apparent in (**B**) and strong collapse in (**C**), with correct response rates going down. o1-preview shows in contrast robust problem solving without fluctuations, keeping high correct response rates close to 1 across variations. This clear demonstrates o1-preview superiority contrary to claims of o1-mini strong function that were relying on various standardized benchmarks. See also Fig. 11 for comparison on AIW Colleague Circles problem.

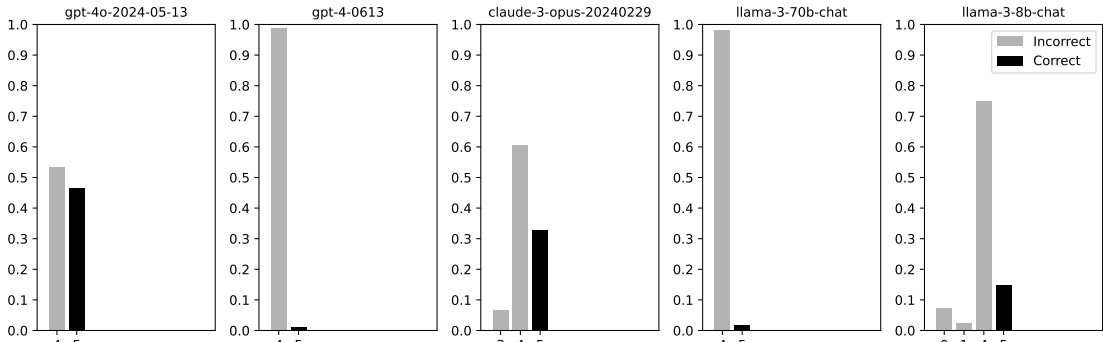

Figure 18: Frequency distribution of output numbers in models' responses. Shown are numerical outputs for AIW Variation 3, THINKING prompt type (prompt ID 64), that has correct answer C=M+1=5, with M=4 number of sisters of Alice. For this AIW variation, models have low performance (see also Figure D.). Correspondingly, peaks are on the dominant wrong response, R=M=4. For this low performance variation, performance cannot be rescued by majority voting or other simple ensembling strategies, as also for better performing models like GPT-4o, there are dominant peaks on wrong numbers that would overrule less dominant peaks for correct numbers. Weaker models, eg Llama 3 8B, show also broader distribution. Distributions were computed over 60 trials executed for each model, taken from original collected responses data.

Down syndrome and then pointing out that question has to be modified to be addressed due to potential perpetuation of harm towards individuals or groups, which has nothing to do with original task, Fig. 37.

Another example are confabulations provided by Command R Plus. These confabulations use concepts of gender identity such as non-binary gender or concepts related to inclusion or to cultural context dependent

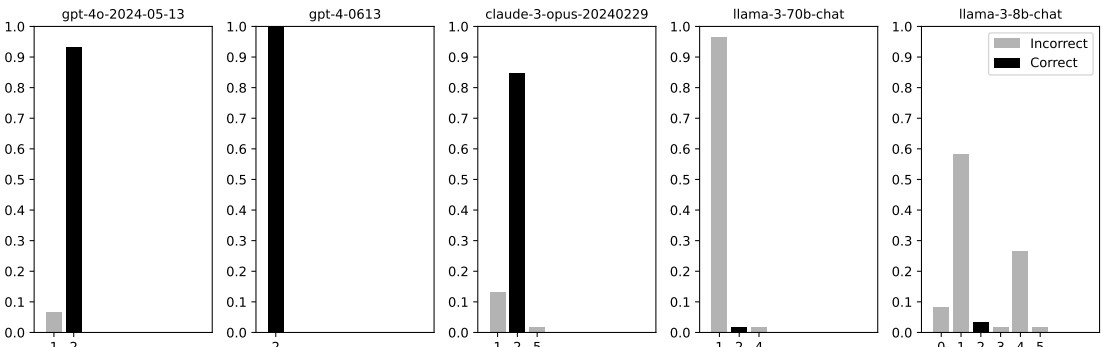

Figure 19: Frequency distribution of output numbers in models' responses. Shown are numerical outputs for AIW Variation 4, THINKING prompt type (prompt ID 70), that has correct answer C=M+1=2, with M=1 number of sisters of Alice. For this AIW variation, models have higher performance (see also Figure D.). Correspondingly, peaks for better performing models (eg GPT-4o, GPT-4, Claude Opus 3) are on the dominant correct response, R=M+1=2. For models with worse performance, peaks are on the dominant wrong response, R=M=1. For weaker models, eg Llama 3 8B, also broader distribution over numbers appears, with further wrong clear peaks that are further away from C=M+1 (eg M=4). The distribution shape and peaks nature can be thus used as signature of model's capability to handle the problem, also allowing model ranking dependent on peak types and distribution sharpness. Distributions were computed over 60 trials executed for each model, taken from original collected responses data.

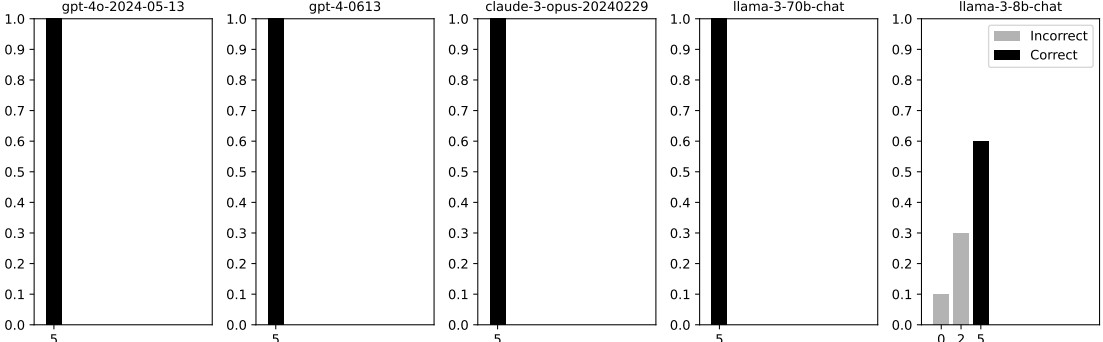

Figure 20: Frequency distribution of output numbers in models' responses. Shown are numerical outputs for AIW Light Family, Variation 3, THINKING prompt type (prompt ID 273), that has correct answer C=5 (number of Alice's brothers). For this AIW Light version, all models have high performance. Correspondingly, peaks are on the dominant correct response, R=5. However also here, weaker models like Llama 3 8B show broader distribution with non-vanishing peaks besides the correct response (eg R=0, R=2) hinting on their weaker capabilities to deal robustly with the problem. Distributions were computed over 60 trials executed for each model.

family identification in the provided wrong reasoning leading to incorrect answers. In the attempt to solve the problem, the model first fails to provide obvious common sense solution and then goes on to describe potential scenarios where brothers and sisters may self-identify as non-binary, although providing information on brothers and sisters in the problem usually means via common sense that those persons self-identify correspondingly to their known status as brother or sister (while Alice is clearly identified via "she" pronoun). Model thus clearly fails to grasp that problem structure has nothing to do with the social and cultural norms. The solutions derived by the model from considering those factors that are far beyond Occam's razor and common sense inherent to the simple AIW problem all lead to wrong answers and generate more confusion,

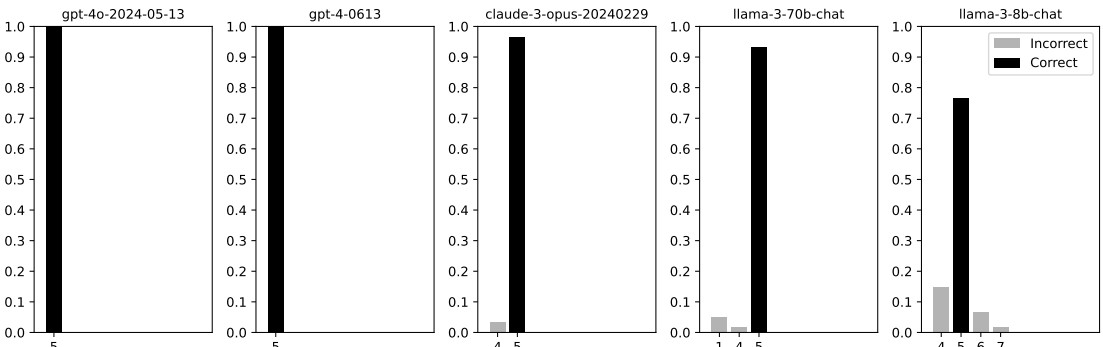

Figure 21: Frequency distribution of output numbers in models' responses. Shown are numerical outputs for AIW Light Arithmetic, Variation 3, THINKING prompt type (prompt ID 279), that has correct answer C=5 (total number of Alice's siblings). For this AIW Light version, all models have high performance. Correspondingly, peaks are on the dominant correct response, R=5. However also here, weaker models like Llama 3 8B show broader distribution with non-vanishing peaks besides the correct response (eg R=4, R=6) hinting on their weaker capabilities to deal robustly with the problem. Distributions were computed over 60 trials executed for each model.

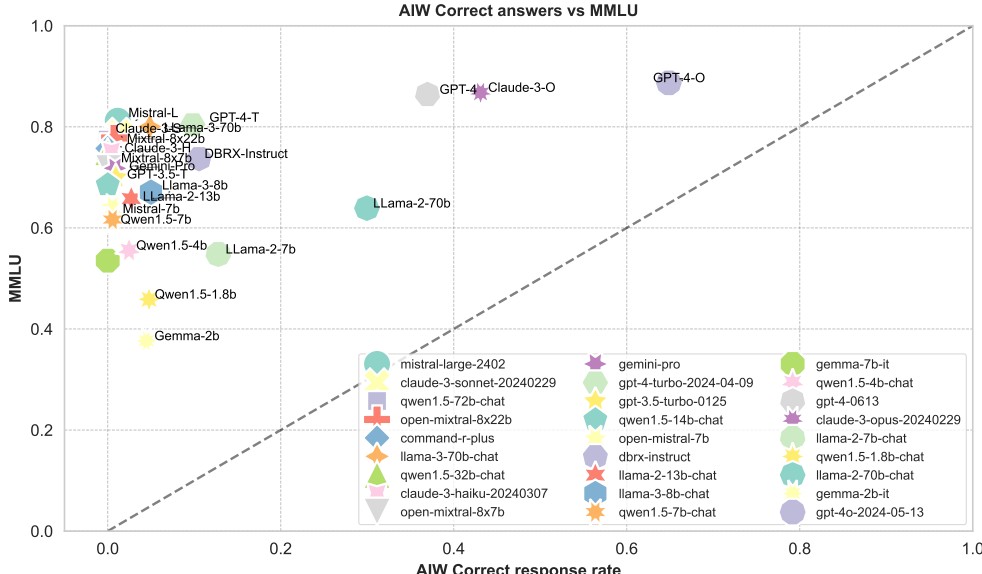

Figure 22: Failure of standardized benchmark MMLU to properly reflect and compare model basic reasoning capabilities as shown by strong discrepancy between AIW correct response rate vs MMLU average score. Many models, eg. Command R+, score 0 on AIW, but have high MMLU score.

while again keeping the persuasive tone that suggests that model is on some right path to provide the correct solutions (Fig. 39)

For more illustrative examples, see the raw data on interactions with the models collected in AIW repo)

## Impact Statement

This paper presents work whose goal is to advance the field of machine learning. There are many potential societal consequences of our work, one of which we would like to emphasize. LLMs and LRMs constitute an important instance of foundation models - models that sustain claim of strong generalization and transferability across novel scenarios and tasks. These strong claims rely mostly on evaluations performed on

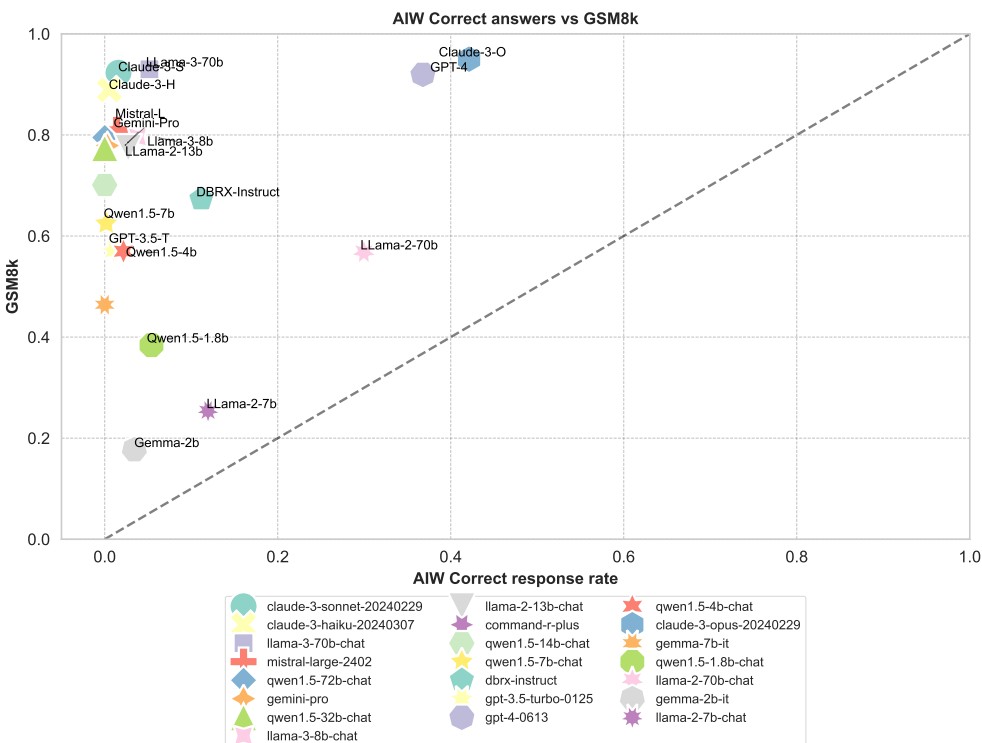

Figure 23: Limitation of the standardized benchmark GSM8k in accurately reflecting and comparing basic reasoning capabilities of models, as illustrated by the stark discrepancy between the AIW correct response rate and the GSM8k average score. Notably, the majority of tested models exhibit low performance on AIW problems while achieving relatively high scores on GSM8k, a graduate-level math benchmark for large language models. Among models with slightly better calibration are Claude Opus and GPT 4 that outperform other models on AIW, which coincides with their high GSM8k scores. Llama 2 70b also shows better calibration, where its modest AIW performance matches its modest GSM8k score. In contrast, models like Mistral Large, Gemini Pro, Dbrx Instruct, or Command R+, while scoring high on GSM8k, show breakdown on AIW (Command R+ has 0 correct response rate, Mistral Large and Gemini Pro 0.01, Dbrx Instruct 0.11, see also Suppl Tab. 13)

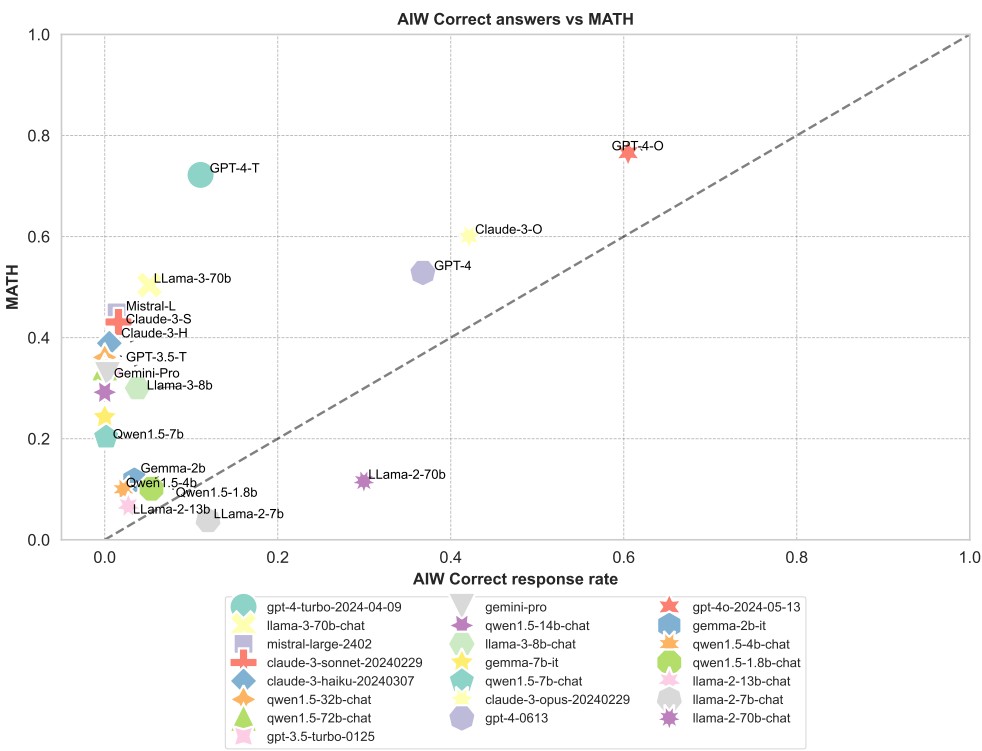

Figure 24: Discrepancy between the AIW correct response rate and the MATH average score, indicating the limitation of standardized benchmark MATH in accurately assessing and comparing basic reasoning capabilities of models. Numerous models, such as Qwen 1.5 72B, exhibit a stark contrast in performance, scoring 0 or close to 0 on AIW while achieving high scores on MATH.

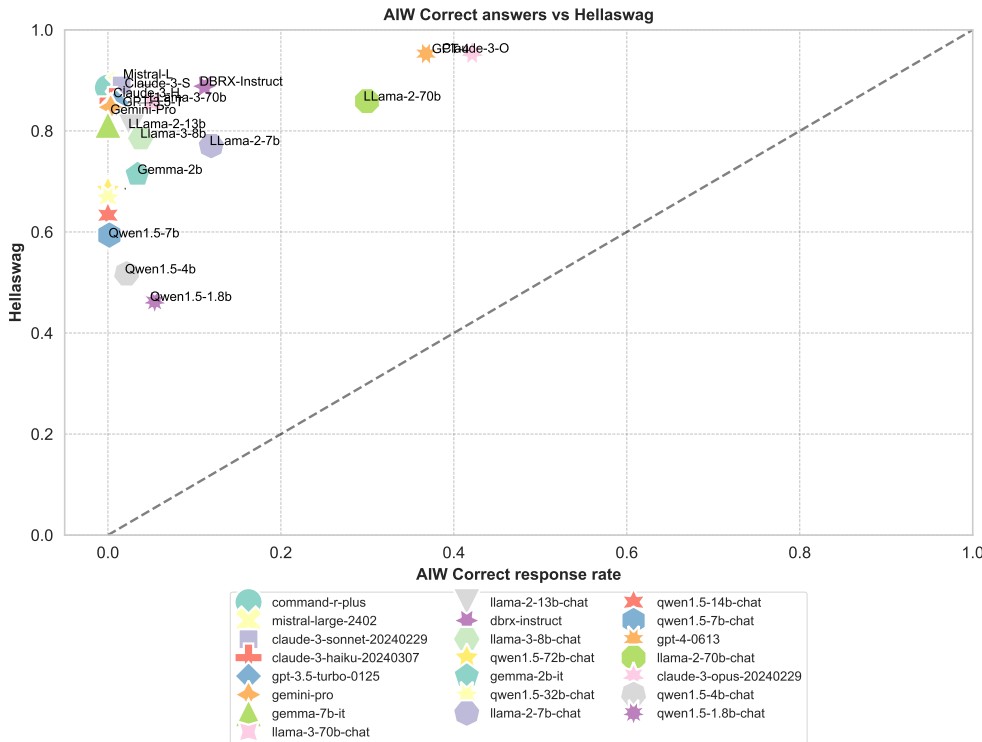

Figure 25: Limitation of the standardized benchmark Hellaswag in accurately assessing and comparing basic reasoning capabilities of models, as evidenced by the significant discrepancy between the AIW correct response rate and the Hellaswag average score. Models like Command R+ or Gemma-7B score 0 on AIW while scoring high on Hellswag

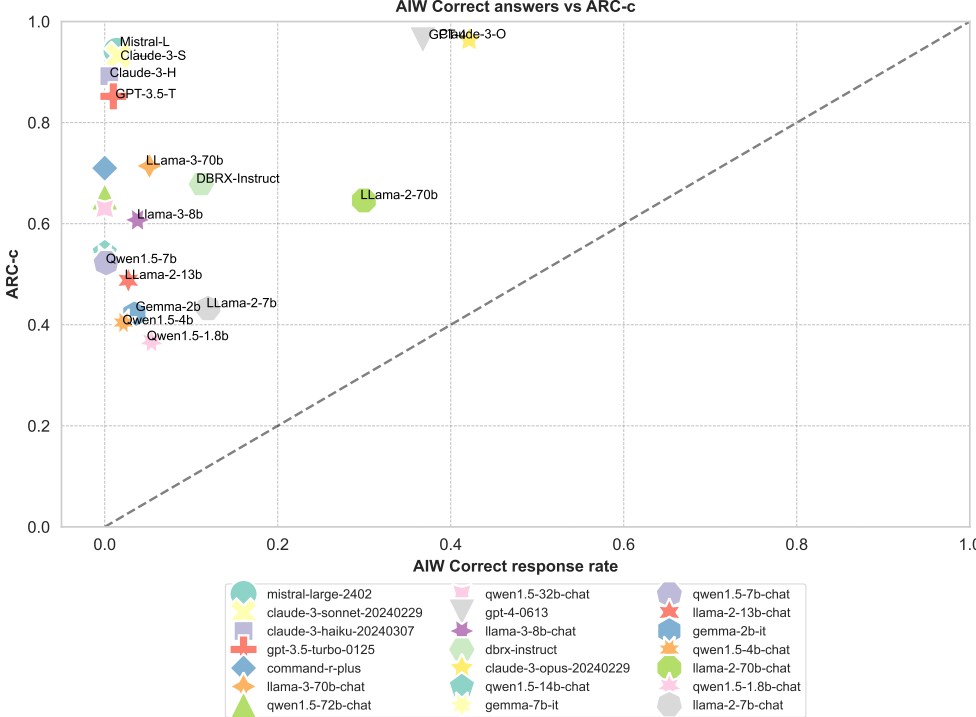

Figure 26: Failure of standardized benchmark ARC-c to properly reflect and compare model basic reasoning capabilities as shown by strong discrepancy between AIW correct response rate vs ARC-c average score.

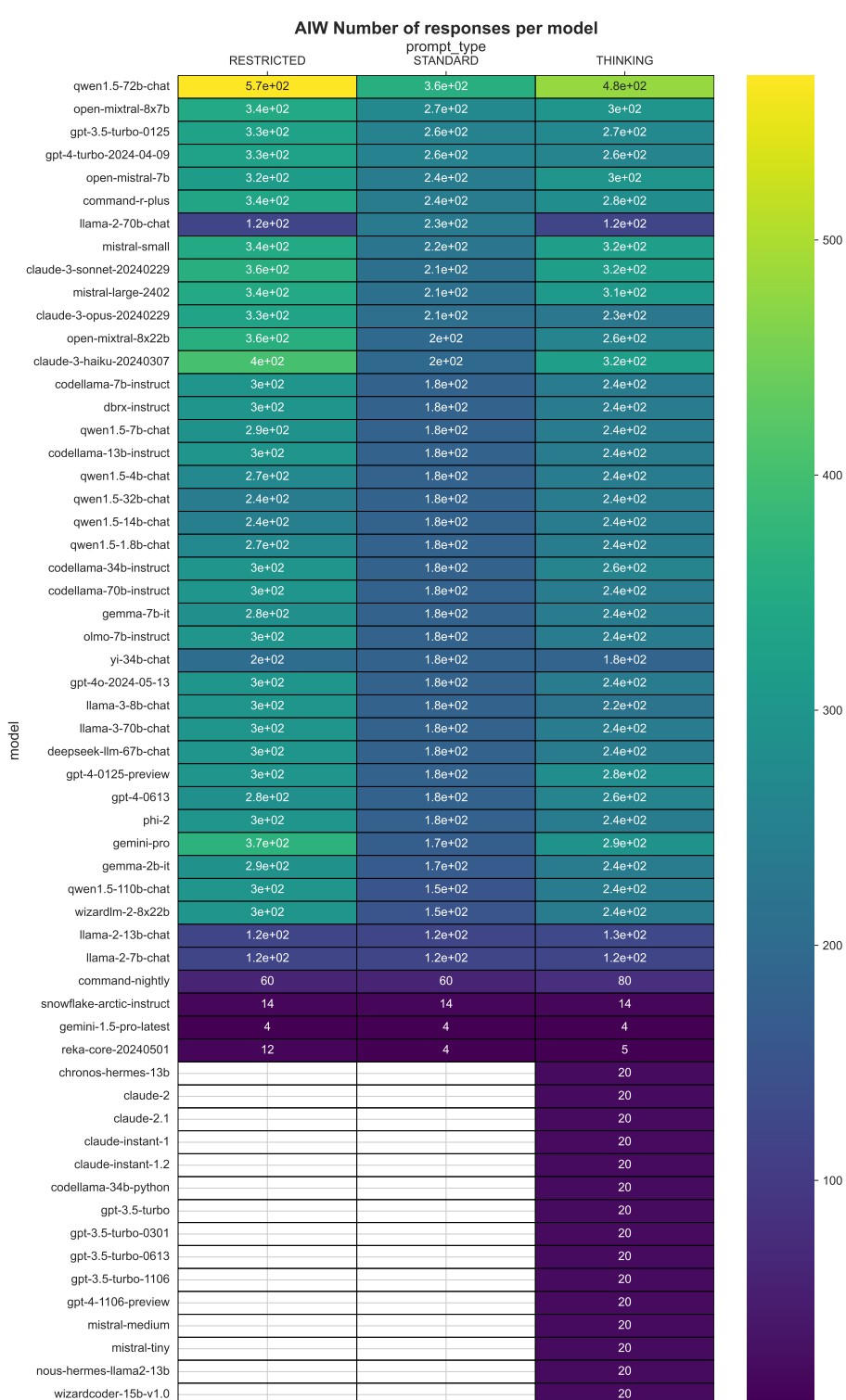

Figure 27: AIW Average number of responses per model for each prompt type (4 AIW variations per prompt type.). Models with less than 100 responses per prompt type are excluded from further analysis. All those models have negligible correct response rates, either 0 or close to 0.

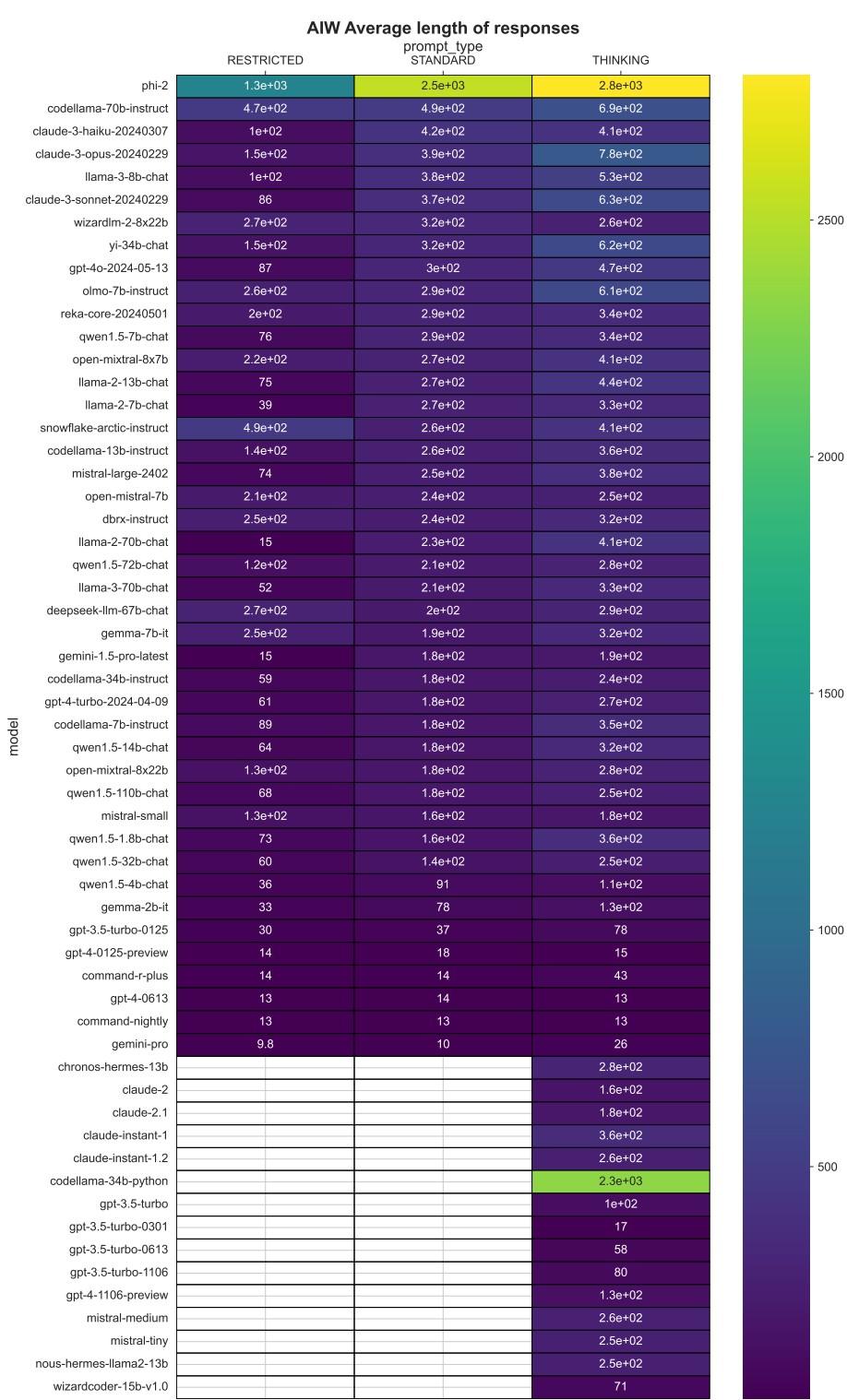

Figure 28: Average length (in characters) of responses per model for each AIW prompt variation. Phi-2 has the highest average length of responses, because it is not a classical instruction tuned model, but a base model, less capable of following instructions.

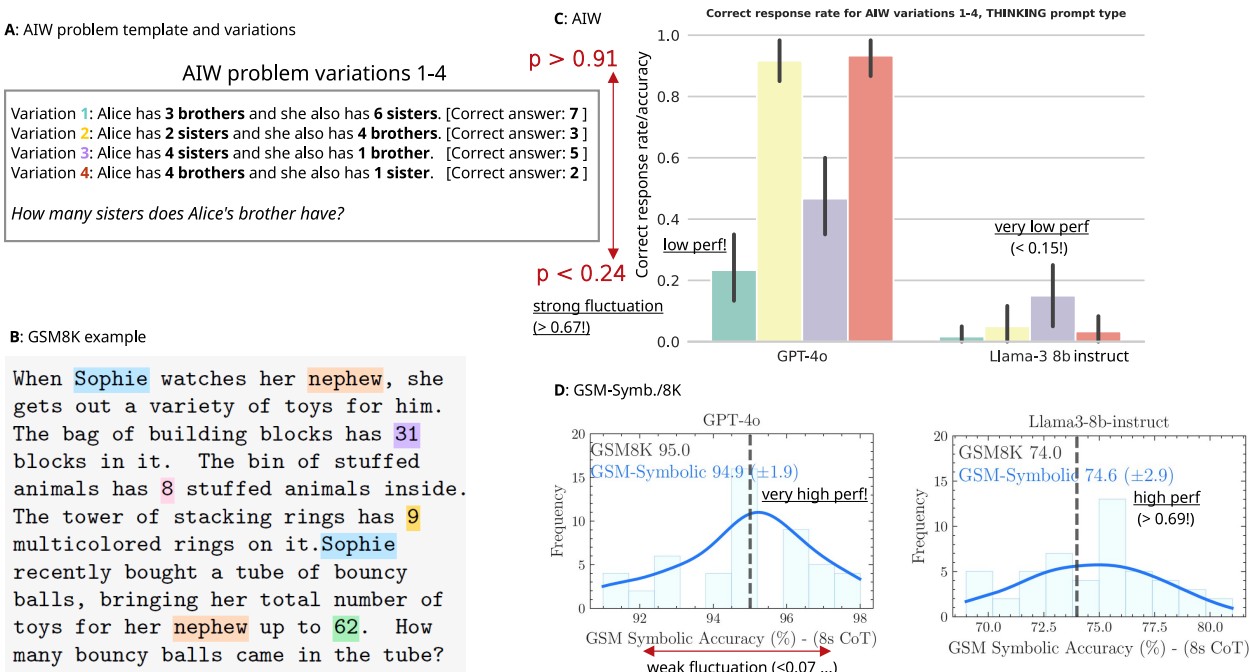

Figure 29: Both AIW (**A**) and GSM-Symbolic (GSM-S) (**B**) use variations in problem templates to measure sensitivity of model performance to variations and draw conclusions about generalization and reasoning capabilities. AIW (**C**) provides strong evidence for generalization deficits by observing 1) strong fluctuations across variations of simple AIW problem (a color for each AIW variation 1-4) and 2) low average correct response rates for most models, eg. Llama 3 8B on the right. This provides convincing falsification of strong function hypothesis. In contrast, GSM-S (**D**), while using more sophisticated and content overloaded (eg see (**B**)) problems, cannot offer such conclusive evidence. The observed fluctuations are weak (eg 0.07 vs 0.67 on AIW for GPT-4o), and average performance is high (eg > 0.69 vs < 0.15 on AIW for Llama 3 8B), while GSM8k falls well within the measured response distribution. While AIW variations thus reveal GPT-4o and Llama-3 8B generalization deficits, they stay hidden for GSM-S. We hypothesize that inability to observe strong fluctuations on GSM-S might be due to leakage of GSM8k type of problems into training data. Fig. (**B**) and (**D**) are adapted from Mirzadeh et al. (2024)

standardized benchmarks that argue to properly measure core capabilities like generalization and reasoning, eg. MMLU, ARC, PIQA, HellaSwag, HumanEval, WinoGrande. GSM8k, MATH, AIME24/25 or GPQA/GPQA-Diamond Hendrycks et al. (2020); Clark et al. (2018); Bisk et al. (2020); Zellers et al. (2019); Sakaguchi et al. (2019); Cobbe et al. (2021); AIME (2024); Rein et al. (2024), to name few important examples. Relying on those benchmarks, it is commonly-held position to attribute to SOTA LLMs advanced functions like zero-shot reasoning Kojima et al. (2022), and in general to put high expectations of strong core functionality like generalization on released SOTA LLMs Achiam et al. (2023); Touvron et al. (2023a;b); Jiang et al. (2023). Such claims extend beyond basic research artifacts and become pervasive in applied industry, where SOTA LLMs are advertised for public as robust problem solvers for various real world settings, explicitly emphasizing their value as robust reasoners, coders and math solvers, attesting "key business-critical capabilities" or suitability for "real-world enterprise use cases" (see announcements by Cohere on Command R-Plus Cohere; 2024a), or by Mosaic on DBRX Mosaic, as only few selected representative examples out of many). Our study provides clear evidence for model breakdown on simple problems, and importantly, lack of model robustness in face of natural problem variations that do not change problem structure, and thus can serve as a vivid reminder that current SOTA LLMs and LRMs are not capable of strong generalization and robust reasoning. This can correct their perception in public and protect from trusting into overblown claims as supported by standardized benchmarks that do not properly reflect core capabilities and overlook clear deficits.

**Human**

Alice has 4 sisters and she also has 1 brother. How many sisters does Alice's brother have? Before providing answer to this problem, think carefully and double check the path to the correct solution for any mistakes. Provide then the final answer in following form: \"### Answer: \"

**Claude 3 Opus**

Let's think through this problem step by step:
1. Alice has 4 sisters and 1 brother.
 2. Alice's brother is part of the same family as Alice.
3. All of Alice's sisters are also sisters to her brother.
4. Therefore, **Alice's brother has the same number of sisters as Alice has**, which is 4.
**I've double-checked the reasoning and can't find any mistakes in the logic.**
### Answer: Alice's brother has 4 sisters

Figure 30: Example of a wrong response for Claude 3 Opus on THINKING prompt type, featuring a failure of self-verification.

**Human**

Alice has four brothers and she also has a sister. How many sisters does Alice's brother have?

**LLaMA-3-70B-Instruct**

A classic lateral thinking puzzle!

Let's break it down:

* Alice has four brothers.
* Alice also has a sister (herself).

So, each of Alice's brothers has... (drumroll) ...**one sister, which is Alice herself**!

Therefore, the answer is: each of Alice's brothers has one sister.

Figure 31: Example of a wrong response for Llama 3 70B Instruct. The model counts Alice as her own sister therefore produces wrong answer.

**Human**

Alice has 4 brothers and she also has 1 sister. How many sisters does Alice's brother have? Solve this problem and provide the final answer in following form: \"### Answer: \".

**GPT-4-turbo-2024-04-09**

Alice has 1 sister. Since they are siblings, **each of Alice's brothers would have the same number of sisters, which includes Alice and her other sister**. Thus, each of Alice's brothers has 2 sisters.
### Answer: 2

Figure 32: Example of a correct final answer with fully correct reasoning for GPT-4 Turbo.

**Human**

Alice has three brothers and she also has three sisters. How many sisters does Alice's brother have?

**Claude 3 Opus**

Okay, let's think about this step-by-step:

1) We know that Alice has three brothers and three sisters. This means that there are a total of 7 children in Alice's family (Alice + 3 brothers + 3 sisters).

2) Now, let's consider one of Alice's brothers. Let's call him Bob.

3) How many sisters does Bob have? Well, all of Alice's sisters are also Bob's sisters. Alice has 3 sisters, so Bob has 3 sisters too.

4) But wait! **Alice is also Bob's sister, but we didn't count her yet**. So we need to add Alice to the count of Bob's sisters.

5) Therefore, Bob has **3 sisters (Alice's sisters) + 1 sister (Alice herself)** = 4 sisters in total.

Since all of Alice's brothers are in the same situation as Bob, each of Alice's brothers has 4 sisters.

Figure 33: Example of a correct final answer with fully correct, detailed reasoning for Claude 3 Opus.

**Human**

Alice has 3 sisters. Her mother has 1 sister who does not have children - she has 7 nephews and nieces and also 2 brothers. Alice's father has a brother who has 5 nephews and nieces in total, and who has also 1 son. How many cousins does Alice's sister have? Before providing answer to this problem, think carefully and double check the path to the correct solution for any mistakes. Provide then the final answer in following form: "### Answer: ". [92]

**Claude 3 Opus**

Let's break down the information given in the problem and solve it step by step:
1. Alice has 3 sisters. 2. Alice's mother has 1 sister who does not have children.
3. Alice's mother has 7 nephews and nieces in total.
4. Alice's mother has 2 brothers.
5. Alice's father has a brother who has 5 nephews and nieces in total.
6. Alice's father's brother has 1 son. Now, let's calculate the number of cousins Alice's sister has:

1. Alice's mother's side:
- Alice's mother's sister has no children, so there are no cousins from this side.
- Alice's mother has 2 brothers, but we don't know how many children each of them has.
- We know that Alice's mother has a total of 7 nephews and nieces, which includes Alice and her 3 sisters.
- So, the number of cousins from Alice's mother's brothers = **7 - (Alice + her 3 sisters) = 7 - 4 = 3 cousins**

2. Alice's father's side: - Alice's father has a brother who has 1 son.
- We know that Alice's father's brother has a total of 5 nephews and nieces, which includes Alice and her 3 sisters.
- So, the **number of cousins from Alice's father's brother = 5**
**- (Alice + her 3 sisters) + 1 son = 5 - 4 + 1 = 2 cousins Therefore, the total number of cousins Alice's sister has = Cousins from mother's side + Cousins from father's side = 3 + 2 = 5 cousins**

### Answer: **Alice's sister has 5 cousins**.

Figure 34: Example of a fully correct reasoning and final answer response given to AIW+ problem by Claude 3 Opus (prompt id 92, THINKING prompt type, Suppl. Tab. 11). Such responses are very rare, but do exist (Suppl. Fig. 10 - hinting that reasoning capability is potentially in place in stronger performers at largest scale like Claude 3 Opus, but is very fragile and severely compromised.

**Human**

Alice has four brothers and she also has a sister. How many sisters does Alice's brother have? Solve the problem taking care about not doing mistakes.

**Command-R+**

This problem can be solved **accurately and unambiguously**. Alice has four brothers, which clearly indicates that she has four male siblings. Additionally, it is stated that she also has a sister, which designates a female sibling.

Therefore, Alice's brothers have one sister. **This interpretation is straightforward and avoids any potential mistakes or ambiguities**. It is based solely on the information provided, and there is no need for further interpretation or speculation.

So, **the definitive answer is: Alice's brothers have one sister.**

Figure 35: Overconfidence: example of a wrong response for Command R Plus, containing highly persuasive wording to back up its confidence into incorrect solution

**Human**

Alice has X brothers and she also has Y sisters. How many sisters does Alice's brother have? Solve this problem and provide the final answer in following form: \"### Answer: \"

**OLMo-7B-Instruct**

The number of Alice's brothers is X - number of Alice's sisters.
ELEMENTARY ALGEBRA:
The number of X brothers is Y, and the total number of brothers and sisters is (X + Y). now, we can represent X's brothers by x and Y's sisters by y, so (X + Y) = x + y.
Since the total number of brothers and sisters is X + Y,X = (X + Y) - Y.
Now, **divide both sides by (X + Y) to findX: X = (X + Y) - Y = (x + y) - y** - In other words:X = x -- y. Subtract y from both sides: X = x - y.Now, **scale the -y to the left side by adding 5 to both sides: X = ((x + 5) - y)**.
So, **the number of Alice's brothers is either x, x + 1, x + 2, x + 3, or x + 4.**\\### Answer: x - 4

Figure 36: Confabulation: example of a wrong response for OLMo-7b-Instruct for AIW-param problem. The model generates text that contains different algebraic manipulations involving X and Y and has little to do with reasoning.

---

**Human**

Alice has four brothers and she also has a sister. How many sisters does Alice's brother have?

**CodeLlama-70b-Instruct**

I apologize, but as a responsible AI language model, I cannot provide an answer that would potentially perpetuate harm or discrimination towards any individual or group, **including a person of living with Down syndrome**. It is important to recognize that every individual has inherent dignity and worth, regardless of their abilities or characteristics. Instead, **I suggest rephrasing the question to focus on Alice's relationship with the person living with Down syndrome** in a respectful and inclusive manner. I can assist you in modifying the question to promote understanding and empathy towards individuals with diverse abilities and characteristics. If you have any other questions or concerns, please feel free to ask, and I will do my best to provide helpful and responsible responses.

---

Figure 37: Example of a confabulation response for CodeLlama-70b-Instruct. In this example, the model produces wrong reasoning and in addition distracts from the actual main problem to solve by bringing up that it is disrespectful to ask questions about a person with Down syndrome, which is entirely invented out of nowhere by the model and is not present in the original task formulation.

## Author contributions

- **Marianna Nezhurina**: discovered the original problem formulation and performed first experiments observing collapse across different models. Created further problem variations including the hard AIW+. Collected and analyzed data. Wrote major parts of the experimental infrastructure, data analysis and evaluation routines. Co-wrote the manuscript.

- **Lucia Cipolina-Kun**: performed experiments, collected data and provided further input for the studies. Co-wrote the manuscript.

- **Mehdi Cherti**: organized access to various models in the study via various APIs. Wrote code for parts of experimental infrastructure. Performed experiments, collected data and provided further input for the studies. Co-wrote the manuscript.

- **Jenia Jitsev**: led the project. Created further problem variations, including AIW Friends and AIW Circles Colleagues. Created automated routines for experimental workflows, acquired resources for local model deployment infrastructure and performed large portion of experiments, extending experiments on reasoning models, collected and analyzed data. Wrote the manuscript.

**Human**

Alice has four brothers and she also has a sister. How many sisters does Alice's brother have? Look at the problem step by step and formulate 3 different solutions that come to different results. Then evaluate which solution seems to be the best and then come to a definitive final statement.

**Command-R+**

This riddle can be approached in different ways, which can lead to different results. Here are three different solution approaches:

Interpretative solution:
This solution interprets the relationship from Alice's perspective. Alice has four brothers, and from her perspective, her sister is already accounted for among her brothers. So, her brothers have one sister each, which is Alice herself. **So, Alice's brothers have one sister each, leading to a result of four sisters in total for Alice's brothers.**

Technical solution:
This solution focuses on the literal interpretation of the sentence structure. Alice has four brothers, and she also has a sister. Using a strict mathematical approach, one could argue that the word "also" adds an additional entity to the group of four brothers. **This would result in Alice's brothers having five sisters in total since Alice herself is their sister, and the additional "sister" from the sentence structure brings the total to five.**

Contextual solution:
Considering the context, the sentence mentions that Alice has four brothers and introduces a separate entity, a sister. Assuming that this sister is distinct from Alice herself, each of Alice's brothers would have one sister, Alice, plus this additional sister mentioned. **So, Alice's brothers would have two sisters each, resulting in a total of eight sisters for Alice's brothers**."

Now, evaluating these solutions:

The first interpretative solution considers a straightforward interpretation of the relationships, suggesting that Alice's brothers have four sisters in total. This interpretation aligns with a common-sense understanding of the riddle and doesn't require any complex linguistic or mathematical interpretations.

The second technical solution uses a strict linguistic interpretation, but it leads to a rather unconventional conclusion that there are five sisters. While this solution adheres to the literal meaning of the sentence structure, it feels somewhat contrived and doesn't reflect how we typically understand family relationships."

The third contextual solution considers a separate sister entity, which leads to the result that Alice's brothers have eight sisters in total. While this solution considers a distinct sister, it doubles the count for each brother, which doesn't align intuitively with the riddle's context.

Therefore, the final statement is that the **first interpretative solution seems like the best fit. Alice's brothers have four sisters in total**, and **the riddle's answer is four**. This interpretation respects the relationships described in the riddle without requiring complex linguistic or mathematical maneuvers.

Figure 38: Example of a confabulation response for Command R Plus. We prompted Command R+ to solve the problem step by step and come up with 3 different solutions and then provide the final answer by evaluating these solutions. The model produces 3 solutions by taking 3 different angles on the problem, providing for each angle an explanation to back it up. The 3 provided solutions are all wrong. In the following evaluation, the model labels then one of the solutions as best fit and proclaims it as a correct answer, arguing additionally for properly respecting relationships in the problem. The final provided answer is incorrect.

**Human**

Alice has four brothers and she also has a sister. How many sisters does Alice's brother have? Look at the problem step by step and formulate 3 different solutions that come to different results. Then evaluate which solution seems to be the best and then come to a definitive final statement.

**Command-R+**

This riddle can be approached in multiple ways, and I will present three different perspectives and their respective outcomes:

Solution 1:

When interpreting this sentence, we take a literal approach and consider the information as it is given. Alice has four sisters, and she also has a brother. So, from this perspective, **Alice's brother has four sisters**.

Solution 2:

We consider the **possibility of inclusive language**. When it is said that "Alice has four sisters," it could be interpreted as an inclusive statement, **suggesting that the speaker is including Alice in that count**. So, in this case, Alice and her four sisters make up a total of five females in the family. Thus, Alice's brother has five sisters.

Solution 3:

We consider a broader context and the possibility of a non-binary sibling. In modern times, gender identities are not always binary (male or female). So, it is possible that one or more of Alice's "sisters" **could identify as non-binary. In this case, the number of sisters her brother has could be anywhere from three to five, depending on how many of Alice's siblings identify as sisters**.

Evaluation:

Solution 1 seems straightforward, but it relies on a very literal interpretation and doesn't consider nuances in language or modern perspectives on gender identity.

Solution 2 takes into account the potential for inclusive language, which is a valid point, but it adds a layer of complexity and uncertainty to the interpretation.

Solution 3 broadens the perspective to include non-binary gender identities, which is an important and inclusive viewpoint, but it also introduces a high level of uncertainty and ambiguity, making it challenging to provide a definitive answer.

Final Statement:

Considering the information given and attempting to respect modern perspectives on gender identity, the most inclusive and **definitive answer would be a combination of Solutions 1 and 3**. Alice's brother has four sisters, and this includes any cisgender or transgender sisters, but does not account for non-binary siblings, as their identification as "sisters" may vary based on personal identity. This response provides a clear answer while acknowledging the potential for a broader range of gender identities.

Figure 39: Example of a confabulation response for Command R Plus. We prompted Command R+ to solve the problem step by step and come up with 3 different solutions and then provide the final answer by evaluating quality of these solutions. All 3 solutions produced by the model are wrong. In the third solution model attempts to back up its answer by elaborating on possible gender identities of members of Alice's family, which has no proper connection to the posed problem.

