# OpenReview forum: "Detecting generalization deficits in large language and reasoning models by using natural variations in simple problems"
_TMLR — Accepted by TMLR_

### Review · Reviewer_Xyxi · 2025-09-11

**Summary Of Contributions:**

The paper assesses the performance of a variety of reasoning and non-reasoning LLMs on prompt variations in a simple reasoning task. The paper finds that even SOTA models are highly sensitive to prompt variation despite their strong performance on standard benchmarks.

Strengths: the paper provides a thorough investigation of prompt sensitivity in AIW task and is clearly written.

Weaknesses: There is prior work with similar findings/claims based on different reasoning tasks and thus the novelty of the contribution is unclear to me.

**Audience:**

No

**Audience Explanation:**

The paper points out flaws in reasoning generalization in SOTA LLMs and LRMs. While this type of finding would generally be of interest to the TMLR readership, I'm concerned about its novelty as there's existing work that makes similar claims.

**Broader Impact Concerns:**

No concerns.

**Claims And Evidence:**

Yes

**Claims Explanation:**

I have some comments and suggestions on the analyses but I’m generally convinced that the models tested don’t do well on the variations of AIW prompts. However, I’m wondering what I have learned from this work that I didn’t know before. I provide the details below under Requested Changes.

**Requested Changes:**

a) Critical:
1. Novelty. There’s existing work asking very similar questions. For instance, Shojaee et al. ’25 present a comparison of thinking vs. non-thinking LLMs on several non-standard reasoning tasks and arrive at similar conclusions. The approach of employing variations of the prompt isn’t novel either and neither are the claims that the seemingly strong performance of modern SOTA LLMs on standard benchmarks is due to the training set contamination. Given the above, I’d encourage the authors to spell out the novelty of their contribution more and to situate it more clearly in the current research landscape.

Shojaee et al. 2025: The Illusion of Thinking: Understanding the Strengths and Limitations of Reasoning Models via the Lens of Problem Complexity.
2. The benchmarks. Throughout the paper, the authors claim that the models presented perform well on the standard benchmarks. However, I’m a bit confused by the choice of the benchmarks — MMLU, Hellaswag, or ARC are typically considered NLP rather than reasoning benchmarks; GSM8k is on the simpler side for many of the SOTA models. I’d consider looking at AIME or Math-500 as the performance gap here would be more informative.
3. Design choices:  I’d like to see the motivation for testing the various aspects in the ‘Further relevant observations’ section as some of the design choices are unclear to me. For example, it’s not quite clear to me what MMLU can tell us about reasoning or how #3 differs from #8.

b) Nice-to-have's:
1. Section 3.2 currently reads as an afterthought but I think it’s the more interesting part of the paper and I’d like to see more analyses there. For example, have you considered testing some of the points in the ‘Further relevant observations’ such as AIW-Param for reasoning models and comparing the performance to the non-reasoning ones? Have you considered analyzing the reasoning traces?

c) Smaller comments:
1. In Figs. 1 and 2, GPT-4 is doing very well on prompt variation 4. Do you have an intuition why?
2. What are the error bars in the figures?
3. Consider making the figure labels larger and keeping the style of the figures the same (some have guides while others don’t, the font sizes are different sometimes, the bars aren’t always aligned)-- this will increase readability.

---

> ### Author Response · Authors · 2025-09-14
>
> We thank the reviewer for taking time to deal with our work and appreciate various points the reviewer made, which we would like to address in following:
>
> > Novelty. There’s existing work asking very similar questions. Shojaee et al. ’25
>
> We would like to emphasize that we use a very distinct approach to detect generalization deficits than used in Shojaee et al. Main difference is that we use variations in common problem template that do NOT change either problem structure or its difficulty (see Fig. 1 and Fig. A http://tiny.cc/0xjs001). This allows us to clearly disentangle problem complexity/difficulty/execution length and generalization issues when observing failure to solve a problem instance – which could be due to either of both. Eg Shojaee et al uses as variation number of discs in Tower of Hanoi, where models can fail with increasing disc numbers not due to inability to grasp same underlying problem structure (generalization deficit), but simply due to committing errors in handling very long execution necessary for large disc numbers. We create variations that keep both problem structure and difficulty/execution length the same. This allows us to draw conclusions about generalization issues when observing strong performance fluctuations across structure and difficulty preserving instances of the same problem, as other factors like varying problem difficulty can be ruled out. Moreover, we observe collapse on problems that are simple. AIW original can be arguably solved by average elementary school children, which is not the case for problems in Shojaee et al. Another remark with regard to novelty -  Shojaee et al is ArXiv pre-print, which cites our work released as ArXiv pre-print (we kindly ask NOT to actively search for the ArXiv and the citation).
>
> We would also like to clarify a possible misunderstanding – we do not measure sensitivity to prompt variations. We use different prompts types that are fixed, and measure sensitivity to problem variations for a fixed prompt (Fig A, Fig 3-5). Various prompt types are there to ensure that the observed phenomenon is independent of prompt types (Fig 1, 2).
>
> > Benchmark comparison
>
> Main purpose of our study is to show a method for detecting and measuring generalization deficits. Comparison to standardized benchmarks should make aware that high scores on those do not reveal generalization deficits – many models scoring 0 or close to 0 on AIW score high on standard benchs. MMLU, GSM8k and others contain natural language math problems, so that enough affinity to AIW is given, which is a simple common sense natural language math problem. Still, scores diverge strongly, making wonder what standard benchs actually measure if the scores cannot predict clear breakdown on a simple problem like AIW.  The situation with newer reasoning benchmarks like AIME24 and MATH-500 is similar – models scoring high there can have very low AIW scores. We show this for AIW Friends - a problem that is much simpler that e.g. problems contained in AIME:
>
> | Model                         | AIME24 | MATH-500 | GPQA-D | AIW Friends |
> | ----------------------------- | --------: | -------: | -----------: | ----------: |
> | **s1.1-32B**                  |     0.647 |    0.890 |        0.601 |       0.698 |
> | **DeepSeek-R1**        |     0.798 |    0.973 |        0.715 |       0.588 |
> | **o1-mini**                   |     0.636 |    0.900 |        0.600 |       0.529 |
> | **DS-R1-Distilled-Llama-70B** |     0.700 |    0.945 |        0.652 |       0.537 |
> | **DS-R1-Distilled-Llama-8B**  |     0.417 |    0.891 |        0.490 |       0.150 |
> | **DS-R1-Distilled-Qwen-7B**   |     0.555 |    0.928 |        0.491 |       0.107 |
> | **OpenThoughts3-7B**          |     0.690 |    0.900 |        0.537 |       0.070 |
> | **DS-R1-Distilled-Qwen-32B**  |     0.726 |    0.943 |        0.621 |       0.003 |
>
> > … not clear what MMLU can tell about reasoning …
>
> Here, a look into eg elementary mathematics MMLU section can provide clarification how MMLU suppose to measure reasoning https://huggingface.co/datasets/cais/mmlu/viewer/elementary_mathematics
>
> > Further tests on reasoning models and non-reasoning models
>
> Focus of current work when comparing reasoning and non-reasoning models was to show that 1) SFT on reasoning traces can already substantially improve the observed situation in conventional LLMs (Fig. 5b and Fig. E http://tiny.cc/z2ks001) 2) the generalization deficit still exists in reasoning models, as evident by remaining strong performance fluctuations (Fig. 5a). This gives ground for further experiments in follow up work.
>
> > GPT-4 is doing very well on prompt variation 4, why?
>
> As all variations are the same problem, the best guess we have is presence of more variation 4 like samples in the training data (see also Fig. 3 for control experiments, Fig. 4 for leakage experiments)
>
> Error bars in the figures are standard deviation per variation, see Fig. A  http://tiny.cc/0xjs001 and Suppl. Sec. A.2

---

> ### Author Response · Authors · 2025-09-14
>
> > ‘Further relevant observations’ section ... how #3 differs from #8.
>
> "3. Inability to revise wrong responses" (Suppl. Sec. G) deals with ability of the models to revise an incorrect response either via self-verification or by an explicit request from an user pointing out wrong answer in multi-turn interaction.  "8. In-context-learning (ICL)" (Suppl. Sec. I) deals with scenarios where examples of correctly solved problems are placed in the context of the models before posing the test problem to solve.

---

> ### Author Response · Authors · 2025-10-05
>
> We thank again the reviewers for time spent on our work. Following the comments, we provide a revised paper version. The revision re-focuses text on how natural variations of simple problems can be used to detect severe generalization deficits overlooked by standardized benchmarks in both LLMs and LRMs, with AIW problems as examples demonstrating the approach. Following changes were made:
>
> - adapting title, abstract
> - re-writing introduction, parts of discussion and conclusion
> - improving description of control experiments
> - providing new table to show discrepancy between reasoning benchmarks MATH-500, AIME24 and GPQA-Diamond
> - Shortening long sentences to improve readability and clarity
>
> We hope that the revised version brings the main message of the work now clearer to the readers.

---

> ### Author Response · Authors · 2025-10-15
>
> We thank the reviewer for timely feedback and comments. We are glad to hear that reviewer values the addition of reasoning benchmarks. With regard to Shojaee et al., 2025, we would like again to note that our work preceeds Shojaee et al, which is moreover not yet peer reviewed, which made us not to include it in our original manuscript. As it is now indeed relevant per discussion, we include it in the revised version in the introduction as it also helps to clarify the novelty, and in the relevant work section to emphasize differences to our work. We hope the revised version has now all the information that we have provided in the rebuttal.

---

### Review · Reviewer_neNL · 2025-09-13

**Summary Of Contributions:**

This article shows that modern LLMs and LRMs can struggle with a maths problem. The article describes one such problem and shows that LLMs and LRMs have a wide range of success rates attempting variants of that same problem. The article also suggests that successes of some LLMs and LRMs at tackling earlier variants of the problem could be attributed to the problem leakage. The article calls for more critical, reliable and robust benchmark generation that is algorithmic in nature to reduce the impact of a possible benchmark leakage.

**Additional Comments:**

Overall, I found this article interesting. You have provided some evidence of what could not be the reason of the failure. I was hoping you would also provide a bit more on what could be the reason.

Finally, could you please explain to me what is the point of "Humpty Dumpty sat on a wall" in the name of section 3.1?

**Audience:**

Yes

**Audience Explanation:**

I believe the article would be of interest to a broad audience spanning from those actively working on LLMs and LRMs to those who do not but may simply be interested in the current state of the area.

**Claims And Evidence:**

Yes

**Claims Explanation:**

The article provides sufficient evidence that LLMs and LRMs believed by some to possess generalization and reasoning may not do so in all possible situations.

**Requested Changes:**

As the article is heavily focused on proving generalisation and reasoning breakdown given one class of maths problems, making requests for  changes would probably fall outside of the scope. Just a few examples of changes/additions that I found useful would be:

1) You have never defined what do you mean by generalization and reasoning and how would you measure it. Let's say you have a class of students provided with an exam paper that features a range of maths problems like the one you describe. The probability that one of the students would fail one of these problems is not that low. Does it necessarily mean the lack of generalization and reasoning?

2) You have focused on a single problem albeit one that can be formulated in a myriad of different ways. Although it is interesting that LLMs and LRMs may fail at it, it remains unclear why. I believe they all could solve "if N=Y and  K=1, what is N+K" style problem for at least one choice of Y (K indicates brother/sister). Can they do it for all possible choices of Y? Can you then gradually convert the problem above into the one you have investigated to determine at which point they fail?

3) You could have broadened the scope of maths problems LLMs and LRMs struggle with to provide more insight on failed generalization/reasoning. Are generalization and reasoning capabilities linked with only solving maths problems? Is there a value to look at non-maths problems?

---

> ### Author Response · Authors · 2025-09-14
>
> We thank the reviewer for taking time to deal with our work and appreciate various points the reviewer made, which we would like to address in following:
>
> > Let's say you have a class of students provided with an exam paper that features a range of maths problems like the one you describe. The probability that one of the students would fail one of these problems is not that low. Does it necessarily mean the lack of generalization and reasoning?
>
> The presented analogy seems to stem from misunderstanding of what has been done in our work which we would like to resolve. The right analogy to our study would be following:
>
> a class of students (various models in our study) is presented with same N variations of the same simple problem (AIW problem) - and not with a broad set of unrelated problems. Each of N presented problem instances has **the same structure and difficulty**, being created by instantiating different natural numbers in the same problem template (we have N=4 instances corresponding to variations 1-4 in Fig 1; we show here in explanatory Fig. A http://tiny.cc/0xjs001 N=6 instances adding two more, corresponding to variations 1-6)
>
> We measure correct response rate $p_i$ (and standard deviation $\sigma$, shown as error bars) for each problem variation i. If students can generalize well across problem instances, that is, they are able to infer the common problem structure underlying all the presented instances, the correct response rates $p_i$ should differ only slightly across the variations.
>
> Measuring **sensitivity to problem structure and difficulty preserving variations** - introduced into the same AIW problem template -  allows us thus to test hypothesis whether students / models are able to properly generalize across instances of the **same problem** or not. This would not be possible with a set of various unrelated problems as described in the analogy given by the reviewer. While failing on one problem from the set of unrelated problems does not tell much about generalization, having strong performance fluctuations on problem instances of the **very same problem** does allow to conclude that a given student / model has generalization deficit.
>
> Concretely, imagine we present to a class of students AIW problem variations derived from same template like done in our work (see again Fig. 1 and Fig. A).  As we assume human students are able to correctly infer same underlying structure for this simple problem, we do not expect significant differences in correct response rate for any of those variations and it would be highly surprising if e.g. variation 4 would have correct response rate close to 1, while variation 3  would have correct response rate close to 0. This is though what we observe in our work for tested models. As discussed in our work, measuring generalization of SOTA models is challenging, as it is often not possible to tell whether presented test problems might have leaked into training data for the models due to web-scale character of training data. By using variations of a problem template and measuring  sensitivity of models to those variations, we can still detect generalization collapse and avoid illusion of strong function based on high scores achieved on test problems where leakage cannot be ruled out.
>
> >  Although it is interesting that LLMs and LRMs may fail at it, it remains unclear why. ... Can you then gradually convert the problem above into the one you have investigated to determine at which point they fail?
>
> We do investigate why the failure happens. This is done on the one hand by executing the control experiments (Fig. 2, Fig. B http://tiny.cc/1xjs001). In those experiments, we keep description same as in AIW original, change slightly the question, while leaving correct responses across variations same as in AIW original. As models are able to cope with variations in those control problems without performance fluctuations, we can conclude that models do not suffer from low level issues - for instance, this shows that executing arithmetic operations is not an issue. On the other hand, we show that fine-tuning on AIW original instances removes fluctuations for this problem version, while strong fluctuations remain on variations of highly similar problem version (AIW Ext; Fig. 4 and Fig. C, D http://tiny.cc/2xjs001, http://tiny.cc/pxjs001). Together, this provides strong evidence for the failure stemming from generalization deficit, that is handling problems not in the training data, and not from inability to execute simple operations, eg syntax parsing or elementary arithmetics.
>
> > Is there a value to look at non-maths problems?
>
> Definitely yes, it is outside of the current scope.
>
> > What could be the reason of the failure?
>
> As stated above, we obtain evidence that observed failures are due to generalization being compromised in SOTA LLMs and LRMs. Why is that so is an open research question.
>
> > Humpty Dumpty?
>
> is a metaphor for overclaiming models’ strong function.

---

> ### Author Response · Authors · 2025-10-05
>
> We thank again the reviewers for time spent on our work. Following the comments, we provide a revised paper version. The revision re-focuses text on how natural variations of simple problems can be used to detect severe generalization deficits overlooked by standardized benchmarks in both LLMs and LRMs, with AIW problems as examples demonstrating the approach. Following changes were made:
>
> - adapting title, abstract
> - re-writing introduction, parts of discussion and conclusion
> - improving description of control experiments
> - providing new table to show discrepancy between reasoning benchmarks MATH-500, AIME24 and GPQA-Diamond
> - Shortening long sentences to improve readability and clarity
>
> We hope that the revised version brings the main message of the work now clearer to the readers.

---

### Review · Reviewer_EPht · 2025-09-19

**Summary Of Contributions:**

Contributions:
1. Drawing on the analogy of Alice and Humpty Dumpty, the paper uses a simple “Alice’s relatives counting problem” (AIW) to test mainstream state-of-the-art LLMs and LRMs. It shows that models struggle to correctly answer specific task that requires the integration of numerical reasoning, relational reasoning, and logical inference.
2. The paper raises a call for a reassessment of benchmark leakage. By fine-tuning models and then evaluating across different model generations with AIW questions created at different times, the authors find that later models can solve earlier AIW questions but continue to fail on newer variants. This suggests possible exposure of test items during training.

Pros:
1. The paper provides detailed experimental results on a focused reasoning task across a wide range of LLMs and LRMs.
2. The paper uses the observed failures to motivate a discussion of benchmark leakage, offering a valuable perspective on dataset contamination and evaluation reliability.

Cons:
1. The structure of the paper is unclear: contributions are not highlighted, related work is not separated, and the writing relies on overly long sentences.
2. The range of evaluation is too narrow, relying almost exclusively on AIW-style problems that test only one specific combination of relational, numerical, and logical reasoning.
3. Several claims are overstated and extend beyond what the presented evidence can support.
4. The self-evaluation prompt is overly simplistic and does not follow established best practices, such as chain-of-thought prompting, making the conclusion drawn from it relatively weak.

**Audience:**

Yes

**Audience Explanation:**

Yes, at least some of TMLR’s audience would be interested in the findings. Despite the narrow scope of AIW and the weak evidence for benchmark leakage, the results show that even state-of-the-art LLMs struggle with simple reasoning tasks. This makes the study relevant for researchers focused on reasoning evaluation, benchmark design, and dataset integrity, while opening avenues for further work on broader reasoning problems and root-cause analysis.

**Claims And Evidence:**

No

**Claims Explanation:**

Most claims are only partially supported, as the conclusions are often generalized beyond what the experiments demonstrate.

Claim 1: Failure of LLMs on simple reasoning tasks

The evidence shows that multiple SOTA LLMs and LRMs fail on AIW. However, the authors extend this to claim a general reasoning deficit across models. This is an exaggeration, as the experiments address only one narrow reasoning domain and cannot establish a universal reasoning breakdown.

Claim 2: Attribution to low-level issues vs. untrained problem types

The authors initially suggest that failures may stem from low-level issues such as number recognition or relationship reasoning. Controlled experiments largely rule out these explanations, but instead of resolving the cause, the paper shifts to speculating that models fail because of unfamiliar “untrained genres” of problems. This claim, however, is not supported by further experiments or evidence.

Claim 3: Fake reasoning ability due to benchmark leakage

The analysis of model performance across generations provides reasonable grounds to suspect that AIW examples entered training data. However, since AIW is not an established benchmark, its inclusion in training corpora is unsurprising. From this evidence alone, one cannot conclude that other benchmarks are compromised. The call for reassessment is constructive, but claims of widespread benchmark ineffectiveness are overstated.

**Requested Changes:**

1. Revise the title and framing

The Humpty Dumpty analogy is incorrectly attributed to Alice in Wonderland but actually comes from Through the Looking-Glass. This should be corrected to avoid confusion.

2. Broaden the evaluation

The study should incorporate a wider variety of logical reasoning tasks beyond AIW so that its conclusions can be tested under broader conditions and made more generalizable.

3. Strengthen causal analysis

The authors should design experiments that more directly test whether failures arise from number parsing, relation encoding, or higher-level reasoning. Without this, claims about the causes of failures remain speculative.

4. Moderate claims

The conclusions should be stated more cautiously. Terms such as “reasoning breakdown” suggest an overly broad failure of LLMs and should be avoided, and benchmark leakage should be framed as a possibility indicated by AIW evidence rather than definitive proof of the invalidity of other benchmarks.

5. Revise the design of the self-evaluation prompt

The current prompt is too simple and does not incorporate established best practices such as chain-of-thought prompting. To strengthen the validity of the conclusions, the authors should redesign the self-evaluation setting with more robust prompting strategies and compare results across different prompt types.

6. Improve structure and clarity

The paper should explicitly list its contributions in the introduction, separate related work into its own section, and streamline the writing by shortening long sentences and reducing verbosity.

---

> ### Author Response · Authors · 2025-10-05
>
> We thank the reviewer for taking time to deal with our work. We provide a revised version of our work, where we put stronger focus on detecting generalization deficits overlooked by standardized benchmarks in both LLMs and LRMs by using natural variations of simple problems, like the presented AIW problem. We appreciate various points the reviewer made, which we would like to address in following:
>
> > The range of evaluation is too narrow
>
> We would like to clarify that our work does not aim to evaluate models along broad function spectrum. Main point is to provide convincing falsification of the hypothesis that SOTA LLMs and LRMs possess robust zero-shot generalization, as high scores on standardized benchmarks might suggest (eg reasoning benchs  like AIME2024, GPQA-D, etc) . For the falsification, a strong counterexample is sufficient. We construct such a counterexample by showing that natural variations in a simple problem like AIW that neither change problem structure nor its difficulty lead to strong performance fluctuations across variations. This is incompatible with claim of strong, robust generalization. Using a narrow simple problem and its variations, we can thus show existence of generalization deficit and also point to inability to detect it via standardized benchmarks that contain problems of much higher difficulty. Further, we do use different problem versions beyond AIW original – AIW ext, AIW Friends, AIW Plus, AIW Circles Colleagues (Sec. 2.1.2, 2.1.3; Suppl. Sec. B.2) – to show that same observations also hold across various versions (Fig. 3 A-C, Fig. 5 A, Suppl. Figs. 9-13).
>
> > the authors extend this to claim a general reasoning deficit across models.
>
> We tone down the claim in the revised version and emphasize generalization deficit detection. Arguably, failure to perform robustly on such a simple problem as AIW may though raise concerns about general reasoning capabilities as well. As if those were intact, models should be able to infer common simple problem structure underlying AIW instances and show similar performance across all variations without exhibiting strong fluctuations, which is not the case.
>
> > Attribution to low-level issues vs. untrained problem types. Causal analysis. Data leakage.
>
> Our control experiments (AIW Light problems) rule out that failures arise from natural language/number parsing, or inability to execute other basic operations (arithmetic, basic family relations inference) necessary to solve AIW original. (Fig. 3 E-G, Fig. B http://tiny.cc/1xjs001). Would that be the case, we would have observed same strong fluctuations on AIW Light as we do for AIW original, as both share these requirements. We further observe strikingly different performance level on AIW original and AIW ext for Claude 3.5 Sonnet (Fig. 4 A-C) and Llama 3.1 405B (Fig. 14. A-B). These two problem versions are again highly similar and share same operation requirements (Fig. 3 A-B). It is hard to explain why the performance is then different – except assuming one problem being in the training data while another not. To provide causal evidence backing this up, we perform finetuning (FT) with AIW original examples on Llama 3.1 8B. We observe before FT close to 0 performance across AIW original variations, which goes up close to 1 after FT for AIW original, while showing poor performance on AIW ext – despite both problems being highly similar (Fig. 4 D-G). This demonstrates that presence of examples of one problem type and absence of example of another problem type in the training can lead to high performance on one problem and poor performance on another, even if both problems are highly similar. This provides further strong evidence that issues we see are indeed due to generalization deficit, and data leakage can be responsible for creating observed performance differences for trained  vs untrained problem types.
>
> > Chain of thought prompting, results across different prompt types.
>
> We do employ various prompt types – STANDARD, RESTRICTED, THINKING, THINKING v2 (Sec. 2.2; Fig. 1, 2; Suppl. Fig. 7)  - to show that observed phenomena appears independent of prompt variations. Main experiments are executed with THINKING v2 prompt type which uses chain of though prompting (for the raw data with typical chain of thought responses, see eg https://anonymous.4open.science/r/AITW_anonymous-69A6/collected_responses/raw_data_inspection/o1-set/)
>
> > Revise title and framing, moderate claims, improve structure and clarity
>
> We follow the reviewer recommendation and re-focus the text on how generalization deficits  overlooked by standardized benchmarks in both LLMs and LRMs can be detected by using natural variations of simple problems, with AIW problems being examples demonstrating the approach. We revise title, abstract, introduction, discussion, focus claims on generalization deficits, shorten long sentences to improve clarity. We hope that revised version manages to bring the main message now clearer.

---

### Decision · Action_Editor_5fSR · 2026-01-04

**Recommendation:** Accept with minor revision

**Additional Comments:**

Please make the following small edits/checks before final submission:

- Tone down any broad “reasoning deficit/breakdown” language; scope claims to the tested task family/setting.

- Avoid categorical statements about benchmarks “failing”; use more precise wording (“can miss”, “in this setting”).

- Fix/remove the phrases such as 'Humpty Dumpty' attribution if still present.

- Final pass (a few long sentences, especially in discussion/conclusion).

**Audience:**

Yes

**Audience Explanation:**

The paper highlights a practical evaluation gap: strong benchmark scores can hide brittle generalization under small, controlled variations. That’s relevant to researchers working on evaluation, reasoning, and robustness.

**Claims And Evidence:**

Yes

**Claims Explanation:**

The empirical results are clear and generally support the main claims about sensitivity to structure-preserving variations. A few statements about “reasoning deficits” should be tightened to match the scope of the experiments.

---

> ### Author Response · Authors · 2026-01-31
> **Revision before final submission**
>
> Dear Reviewers & Action Editor,
>
> we thank again for detailed reviews and appreciate the action editor decision to accept the manuscript with minor revisions. We provide a further revision that addresses requested changes :
>
> 1. We ensure "reasoning deficit/breakdown" claims are explicitly scoped to the tested AIW problem setting.
> 2. Statements about standardized benchmarks failures to detect generalization failures are accompanied with precise formulations about the generalization failures being found using AIW problem setting.
> 3. "Humpty Dumpty" references has been removed to keep narration clear.
> 4.  We conducted further pass to shorten long sentences, particularly in the discussion and conclusion sections.
>
> While we make it clear that our empirical observations origin from the specific case of AIW problems, we maintain that these findings carry broader implications. The observed failures are not tied to any particular AIW problem formulation - we observe them across family, friends, and colleague versions alike. Further, our control experiments demonstrate that models can actually handle family relationship reasoning well in isolation. The deficit becomes thus apparent across different setting within common generic, formulation agnostic frame - having structure- and difficulty preserving variations in a simple problem that humans would solve easily, and importantly, without showing fluctuations across instances that are actually the same. This indicates that robust generalization is not functioning as expected more generally; a system with genuinely strong problem-solving capabilities should not exhibit such sensitivity to variations that preserve structure and difficulty. Furthermore, the fact that standardized benchmarks assign high scores to models exhibiting clear AIW deficits reveals a general limitation of standardized benchmarks tested in the study, as ability to detect generalization deficits is falsified by not being able to do so in simple AIW scenario where deficit is clearly expressed. We think that the revised manuscript presents evidence backing up these arguments and their scientific relevance appropriately.
>
> Best,
>
> Authors